# *Yersinia entomophaga* Tc toxin is released by T10SS-dependent lysis of specialized cell subpopulations

Oleg Sitsel [1], Zhexin Wang [1], Petra Janning[2], Lara Kroczek[1], Thorsten Wagner[1] & Stefan Raunser [1] ✉

Disease-causing bacteria secrete numerous toxins to invade and subjugate their hosts. Unlike many smaller toxins, the secretion machinery of most large toxins remains enigmatic. By combining genomic editing, proteomic profiling and cryo-electron tomography of the insect pathogen *Yersinia entomophaga*, we demonstrate that a specialized subset of these cells produces a complex toxin cocktail, including the nearly ribosome-sized Tc toxin YenTc, which is subsequently exported by controlled cell lysis using a transcriptionally coupled, pH-dependent type 10 secretion system (T10SS). Our results dissect the Tc toxin export process by a T10SS, identifying that T10SSs operate via a previously unknown lytic mode of action and establishing them as crucial players in the size-insensitive release of cytoplasmically folded toxins. With T10SSs directly embedded in Tc toxin operons of major pathogens, we anticipate that our findings may model an important aspect of pathogenesis in bacteria with substantial impact on agriculture and healthcare.

Pathogenic bacteria secrete an array of toxic proteins that subvert host defences on a cellular level. Tc toxins from numerous prominent pathogens[1] are a megadalton-scale, architecturally complex example of such proteins[2] for which the secretion mechanism remains highly enigmatic[3]. Their insecticidal function is well established both in cell cultures[4,5] and in vivo[6,7], while their relevance in mammalian infections remains only partially explored[8,9]. Tc toxins consist of three core subunits: TcA, TcB and TcC (and occasionally accessory subunits) in an $A_5BC$ arrangement[10], the structures and roles of which have been extensively studied[2,10–15]. For example, in case of the Tc toxin YenTc, each membrane-translocating TcA protomer is composed of subunits YenA1 (130 kDa) and YenA2 (156 kDa) decorated by the putatively receptor-binding[15] endochitinases[16] Chi1 (74 kDa) and Chi2 (83 kDa). This pentameric assembly is topped by a cocoon consisting of the TcB subunit YenB (167 kDa) and one of three available TcC subunits: YenC1, YenC2 or RHS2 (106–108 kDa, including an autoproteolytically cleaved cytotoxic moiety of 32–34 kDa)[10–15,17], giving rise to a 2.4 MDa complex

almost as large as the 2.5 MDa prokaryotic ribosome. Evidence indicates that once secreted, Tc toxins bind to target cells via glycosylated proteinaceous receptors[5] and surface-exposed glycan moieties[15,18–20] before being endocytosed. Endosomal acidification triggers a conformational change in TcA that forces its central channel into the endocytic membrane[13], causing the membrane-embedded channel tip to open[10] and release the cytotoxic moiety from the TcB-TcC cocoon into the target cell cytoplasm[2] where it modifies substrates such as Rho GTPases and the actin cytoskeleton, and ultimately triggers cell death[2,21,22].

Similar to other Gram-negative bacterial toxins, Tc toxins must traverse the phospholipid inner membrane, peptidoglycan sacculus and lipopolysaccharide outer membrane when leaving the cell[23]. While several competing Tc toxin secretion hypotheses exist, including TcB-TcC-driven autotransport to the bacterial surface with lipase-assisted release[24–26] and T3SS or outer membrane vesicle (OMV)-mediated secretion[7,27], the search for the Tc toxin release pathway has been complicated by the fact that unlike most Gram-negative

[1]Department of Structural Biochemistry, Max Planck Institute of Molecular Physiology, Dortmund, Germany. [2]Department of Chemical Biology, Max Planck Institute of Molecular Physiology, Dortmund, Germany. ✉e-mail: stefan.raunser@mpi-dortmund.mpg.de

toxins, they do not appear to fit the stringent criteria[23,28] for type 1 secretion system (T1SS), Tat/T2SS, Sec/T2SS, Sec/T5SS, OMV, T3SS, T4SS or T6SS pathway-mediated export. They are also too large to be encoded on bacteriophage genomes similar to Shiga toxins, which rely on timed lytic viral release mediated by holin/endolysin/spanin-containing phage lysis cassettes to escape bacterial cells[29–31]. Intriguingly, holin/endolysin/spanin clusters outside bacteriophages were recently identified as type 10 secretion systems (T10SS)[32], which were proposed to function as an avenue for non-lytic protein export on the basis of studies of the archetypal *Serratia marcescens* T10SS[32–35].

In this study, we explored the secretion mechanism of YenTc, the Tc toxin solely responsible for the extreme insect lethality of *Y. entomophaga*[7]. By combining microscopy, proteomic analyses and targeted genomic knockouts, we discovered that a small subset of specialized cells, which we term 'soldier cells', releases YenTc and other virulence factors using a pH-sensitive T10SS in a lytic fashion, and identified the temperature-sensitive regulatory machinery governing soldier cell behaviour and coordinating YenTc production with the T10SS. We then visualized the step-by-step release of YenTc using cryo-electron tomography (cryo-ET). These results resolve the mystery of Tc toxin secretion in pathogens and showcase how specialized cells confer virulence to entire bacterial populations.

## Results

### pH-dependent secretion of Tc toxins

To study Tc toxin secretion, we investigated how growth medium composition affects YenTc production and release. *Y. entomophaga* exhibited very poor secretion into acidifying medium compared with non-acidifying medium (Extended Data Fig. 1a). This led us to hypothesize that *Y. entomophaga* protein secretion is pH dependent. Increasing the pH of the medium from 5.5–6.0 to 7.0–8.0 or reconstituting cells in high pH buffer indeed prompted rapid protein secretion (Extended Data Fig. 1b). Systematic screening revealed that secretion occurs in the pH range of 6.3–10.0, with an optimum between pH 6.9 and 8.9 (Extended Data Fig. 1c,d). Notably, many insect families including Coleoptera which encompasses the natural *Y. entomophaga* host *Costelytra zealandica*, have an acidic anterior midgut and alkaline posterior midgut[36,37]. The pH sensitivity of secretion probably ensures that YenTc is released near the latter, where invading *Y. entomophaga* actively establish themselves[38].

We used mass spectrometry to identify whether other proteins are co-secreted with YenTc. Surprisingly, the secreted fraction contained atypical extracellular proteins (Fig. 1a inset). We then compared their enrichment in the secreted vs non-secreted proteome, revealing that most remained unchanged or decreased (68% and 29%, respectively) and suggesting a very non-specific secretion mechanism. However, 3% were highly enriched, including YenTc and many other toxins/virulence factors, as well as chitin-modifying enzymes

(Fig. 1a). Mysteriously, nearly all lack an established signal sequence for export (Supplementary Fig. 1a), suggesting that *Y. entomophaga* exports this protein cocktail without requiring dedicated secretion signal sequences.

### Specialized cells secrete YenTc using a T10SS

To identify how these bacteria release YenTc and other constituents of the toxic cocktail, we established a scarless edit-capable targeted genomic editing protocol for *Y. entomophaga* (Extended Data Fig. 2a). After knocking out all established *Y. entomophaga* secretion systems (Sec, Tat, T1SS, T2SS, T3SS and T6SS) as well as components of the unconventional secretion pathways hypothesized for the *P. luminescens* Tc toxin[24,25] (lipases, OMV-promoting enzymes, the Tc toxin itself), all resulting strains secreted YenTc and/or other toxins (Extended Data Fig. 2b,c), meaning that in *Y. entomophaga*, neither the previously proposed mechanisms nor other common secretion systems export these toxins.

To resolve this mystery, we fused YenA1 or YenA2 with sfGFP to monitor toxin secretion. Fluorescence microscopy revealed that surprisingly, only a fraction of these isogenic cells expressed the toxin (Fig. 1b), indicating bimodality[39]. Toxin-expressing cells differed morphologically from the rest in the post-log phase (Fig. 1c), being larger and less motile. Of all factors tested (quorum sensing, oxygen levels, genotoxic stress and so on), temperature had by far the strongest influence on the appearance of YenTc producer cells (Extended Data Fig. 3a–n), which makes sense for an insect pathogen. Absence of another host marker, complex organic compounds, also strongly disincentivized differentiation into YenTc-producing cells (Extended Data Fig. 3h,n). Finally, knocking out autoinducer-1 quorum sensing molecule production reduced YenTc-expressing cells by two thirds, emphasizing the role of interbacterial communication in the decision to produce YenTc and correlating with reduced secretion of a corresponding transposon mutagenesis assay hit[40].

We used confocal fluorescence microscopy to visualize pH-induced secretion of YenTc-expressing cells. While YenTc non-expressing cells remained unaltered, the enlarged YenA2-sfGFP-expressing cells underwent a striking metamorphosis within minutes by collapsing into an associated cluster of vesicles and thereby releasing the cytoplasmically localized YenA2-sfGFP (Fig. 1c and Supplementary Fig. 2). This prime example of self-destructive cooperation in bacteria[41] demonstrates that YenTc release is the result of a controlled lysis strictly dedicated to toxin release rather than a typical secretion process, explaining our initially perplexing observation of atypical extracellular proteins (Fig. 1a, inset). We found the remarkable functional, behavioural and morphological differentiation of these YenTc-producing cells compared with their naive counterparts reminiscent of the enlarged, suicide-capable *Globitermes sulphureus* termite soldiers, known to explosively release contents of a specialized hypertrophied gland for colony defence[42].

**Fig. 1 | *Y. entomophaga* soldier cells release YenTc and other toxins using YenLC, a T10SS. a**, Volcano plot of the MS analysis showing significant enrichment (*t*-test *P* < 0.05) of toxic proteins and a T10SS component in the secreted protein fraction of a *Y. entomophaga* culture (comprising 3% of proteins identified) compared to the leftover, non-secreted fraction remaining in the cells post-secretion. After investigating whether any proteins of the secreted fraction are enriched compared with the non-secreted proteome, 68% of identified proteins showed no significant difference in distribution between these fractions, while 29% (primarily membrane proteins) were enriched in the non-secreted fraction. Purple, YenTc components; red, other secreted toxins and virulence factors; green, YenLC components; orange, cytoplasmic proteins enriched in the secreted fraction. Relevant UniProt accession numbers are provided in Methods. *n* = 3 biological replicates. Dotted lines denote significance thresholds (Methods); that is, −log$_2$-transformed fold change > 1.5 on the *x* axis, and *P* < 0.05 on the *y* axis. Full datasets used for this figure are in the Source Data. Inset: the original subcellular localizations of the secreted fraction proteins for

which such information is available (540 of 1,656). **b**, Only a minor fraction of isogenic *Y. entomophaga* cells produces YenA2-sfGFP (green). GFP$^+$ cells: 109 (1%) of 10,373 total. Scale bar, 20 μm. *n* = 3 biological replicates. **c**, Timelapse of YenA2-sfGFP release from soldier cells (green) after acidified growth medium pH was raised to 8.0. Scale bars, 5 μm. *n* = 10 biological replicates. **d**, Knockout of YenLC components blocks secretion of *Y. entomophaga* soldier cells upon raising the pH of acidified growth media. Rz and Rz1 knockouts were largely but not completely defective when we applied shearing forces, which we were later able to attribute to the fragility of cells generated by PepB activity (Extended Data Fig. 8) and shows that these spanins are crucial in more natural contexts where shearing forces are not involved[95] (Fig. 3). *n* = 3 biological replicates. The bands corresponding to YenA1, YenA2 and YenB are boxed for clarity (for a detailed breakdown of all YenTc components visible on such gels, refer to Supplementary Fig. 2a and the Supplementary Data). Inset: the YenLC operon (RoeA, HolA, PepB, Rz, Rz1) and its closest genomic context. Scale bar, 200 bp.

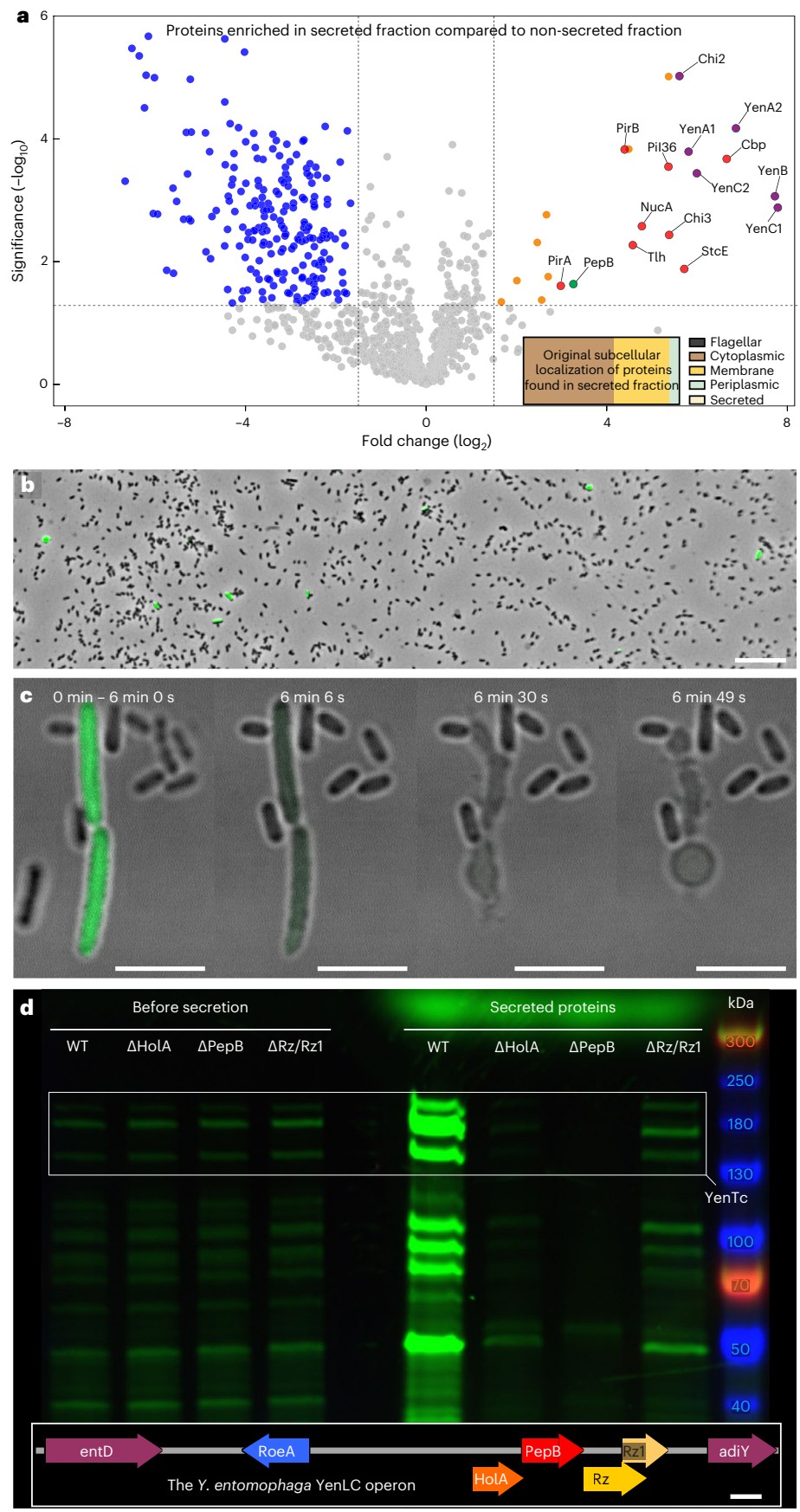

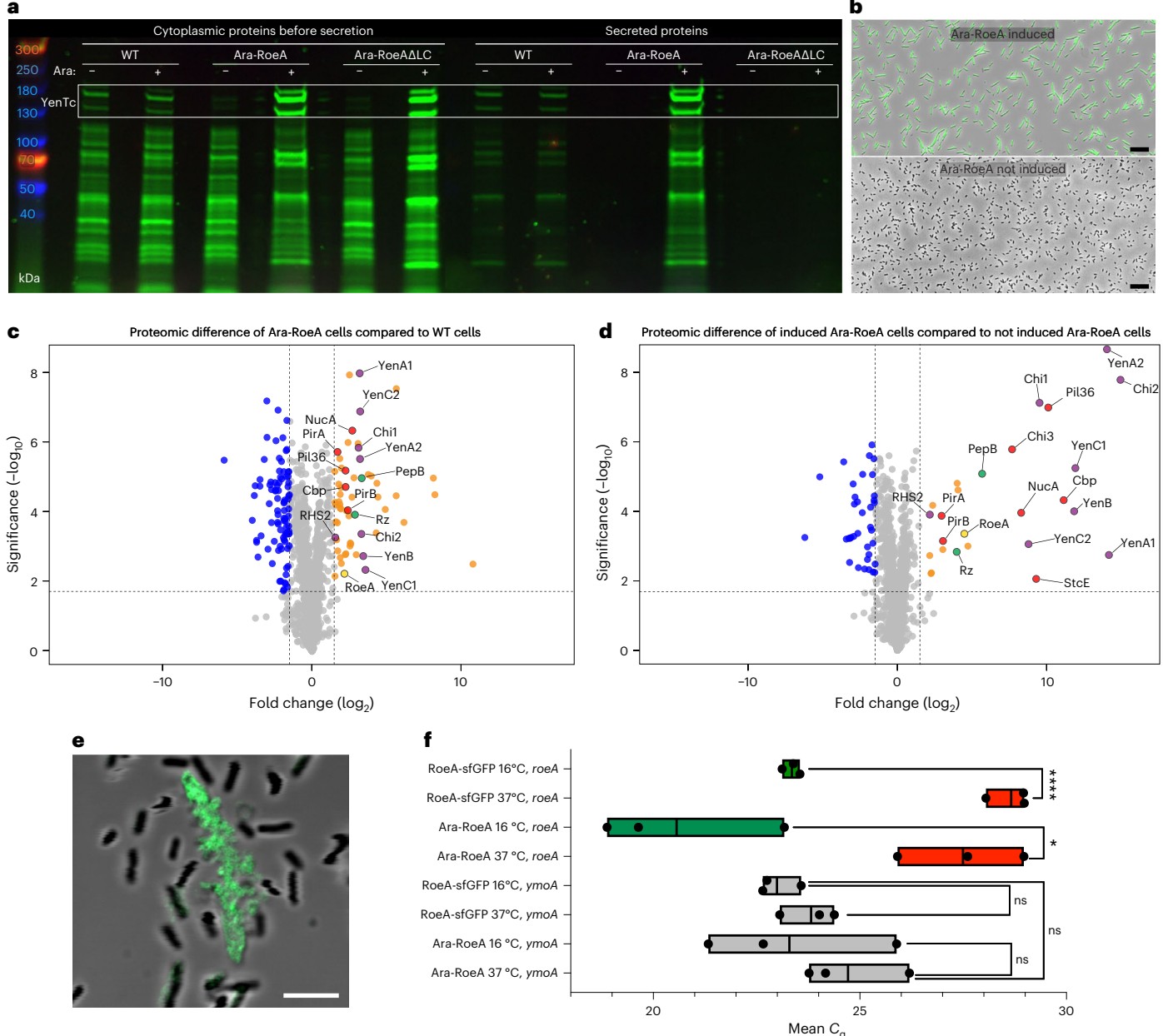

**Fig. 2 | Soldier cells use RoeA to synchronize the expression of YenLC with that of a deadly cocktail of toxins in a temperature-sensitive manner.** **a**, Arabinose induction of the Ara-RoeA strain causes a massive increase in toxin production and secretion compared with wild-type (WT) cells, while absence of induction abolishes these completely. Data are consolidated. $n = 3$ biological replicates. **b**, Induction of RoeA in an Ara-RoeA YenA1-sfGFP strain causes all cells to convert into a soldier cell phenotype (green), while no soldier cells appear in the absence of induction. GFP+ cells: 414 (94%) of 440 total induced cells analysed and 0 (0%) of 1,016 total non-induced cells analysed. Scale bar, 20 μm. $n = 3$ biological replicates. **c,d**, Volcano plot showing significant enrichment ($t$-test $P < 0.02$) of toxic proteins and YenLC components in the total cellular pre-secretion protein fraction of induced Ara-RoeA cells compared with wild-type cells (**c**) or non-induced Ara-RoeA cells (**d**). Colours match Fig. 1a, with the regulatory protein RoeA additionally marked yellow. $n = 3$ biological replicates each. The full proteomic datasets used to generate these figures are available in

the Source Data. **e**, Only explosive lysis-competent soldier cells of the RoeA-sfGFP strain express sufficient RoeA-sfGFP to be seen by confocal fluorescence microscopy. Unlike the soluble YenA2-sfGFP toxin (Fig. 1c and Supplementary Fig. 2c), DNA-bound RoeA-sfGFP remained associated with cell remnants after YenLC-mediated lysis. Scale bar, 5 μm. $n = 10$ biological replicates. See Extended Data Fig. 5 for quantitative data. **f**, $roeA$ transcript levels as measured by RT–qPCR are strongly dependent on temperature regardless of whether the 5′ or 3′ untranslated region (UTR) is disrupted (in the Ara-RoeA and RoeA-sfGFP strains, respectively), while mRNA levels of the $ymoA$ control are not. Data are shown as mean ± minimal/maximal values, $n = 3$ biological replicates, with $P$ values measured using unpaired two-tailed $t$-test ($P$ values for compared 16 °C vs 37 °C pairs, in strain (gene) format: RoeA-sfGFP ($roeA$), ****$P < 0.0001$; Ara-RoeA ($roeA$), *$P = 0.0121$; RoeA-sfGFP ($ymoA$), NS$P = 0.1682$; Ara-RoeA ($ymoA$), NS$P = 0.4111$; RoeA-sfGFP ($ymoA$) vs Ara-RoeA ($ymoA$), NS$P = 0.1003$).

---

This perceived similarity led us to term these large YenTc-releasing kamikaze cells as 'soldier cells'.

In light of soldier cells using controlled lysis to release YenTc, we were intrigued by the exclusive presence of the endolysin PepB in the

secreted proteome (Fig. 1a), which has striking similarity to an archetypal *S. marcescens* T10SS[32,33] component that was shown to cleave peptidoglycan crosslinks[33,43]. In the *Y. entomophaga* genome, this endolysin is situated between holin HolA and bicomponent spanin Rz and Rz1

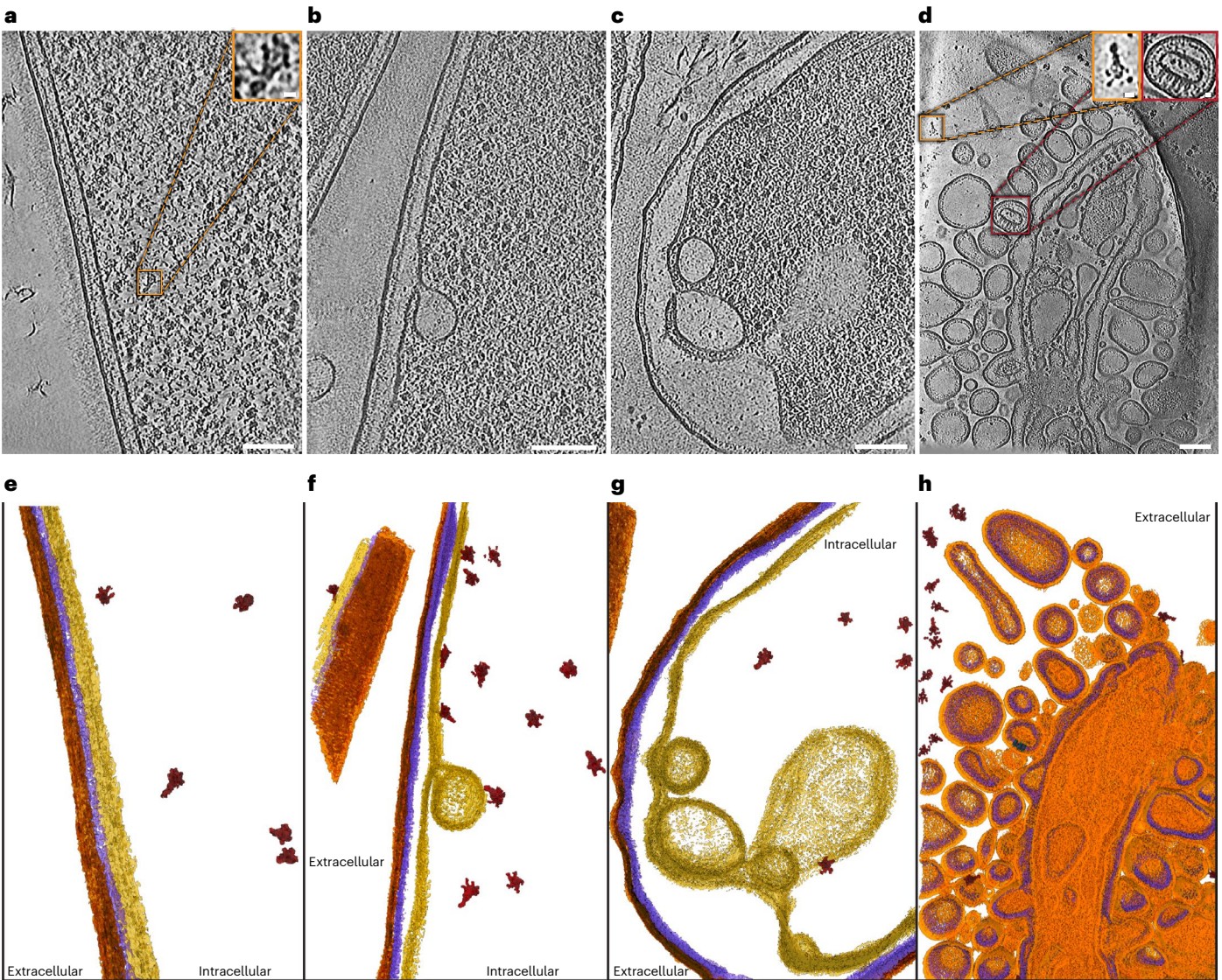

**Fig. 3 | Step-by-step effect of YenLC action and release of YenTc visualized by cryo-ET. a**, A single slice from an Ara-RoeA ΔYenLC cell tomogram, representing the pre-secretion state. Scale bar, 100 nm. *n* = 20 biological replicates. Inset: a fully assembled cytoplasmic YenTc holotoxin. Scale bar, 10 nm. **b**, Ara-RoeA ΔPepB/Rz/Rz1 cell tomogram slice, representing the state of secretion after holin action. Scale bar, 100 nm. *n* = 15 biological replicates. **c**, Ara-RoeA ΔRz/Rz1 cell tomogram slice, representing the state of secretion after endolysin action. Scale bar, 10 nm. *n* = 86 biological replicates. **d**, Ara-RoeA cell tomogram slice, representing the state of secretion after spanin action. Scale bar, 100 nm. *n* = 105 biological replicates. Left inset: a YenTc holotoxin secreted to the external environment. Scale bar, 10 nm. Right inset: an apparently unfused vesicle potentially derived from an area of the cell envelope that contained proteins spanning the entire cell wall. Scale bar, 10 nm. **e**–**g**, Annotated densities from the tomograms shown in **a**–**d**. Dark orange, outer membrane; purple, peptidoglycan; yellow, inner membrane; dark red, YenTc. Only those YenTc densities that could be definitively identified in these 50–100-nm-thick slices were annotated as such. **h**, Annotated densities from the tomogram shown in **d**, presented as a diagonally sectioned view to demonstrate the internal structure that arises after spanin action. Light orange, fused membranes; purple, peptidoglycan; dark red, YenTc; dark blue, potential cell envelope-spanning protein complexes.

(Fig. 1d inset), and is surrounded by transcription factors and essential genes similarly to the *S. marcescens* T10SS, and thus indeed belongs to a T10SS discrete from the YenTc pathogenicity island. Intriguingly, such T10SSs are also found directly within Tc toxin operons (Extended Data Fig. 4). An independent transposon mutagenesis assay[40] had some reduced protein secretion hits associated with an intergenic region close to HolA, suggesting that we were on the right track.

Notably, we completely eliminated *Y. entomophaga* protein export upon deletion of the entire T10SS (Extended Data Fig. 2c), as well as HolA and PepB (Fig. 1d), and severely crippled export by knocking out Rz and Rz1. These results demonstrate that this T10SS is the *Y. entomophaga* death factor responsible for controlled lysis-mediated release of not only YenTc by but also all other toxic proteins we found enriched in the secreted fraction, explaining why nearly all lack a

secretion signal sequence (Supplementary Fig. 1a). The mode of action of this T10SS operon prompted us to name it the *Y. entomophaga* lytic cassette (YenLC).

## Synchronization of YenLC and YenTc production

The OmpR/PhoB-type helix-turn-helix fold protein RoeA encoded directly upstream of HolA drew our attention as a candidate for controlling YenLC expression, analogously to how the LysR-type transcription factor ChiR controls bimodal production of the nearby *S. marcescens* T10SS and several chitinolytic enzymes[34]. To test this, we replaced the native RoeA promoter with arabinose-inducible regulatory elements to create the Ara-RoeA strain. In arabinose-free conditions, this strain stopped secreting and, surprisingly, producing YenTc (Fig. 2a), while adding arabinose massively boosted both compared with wild-type

cells. By contrast, the control Ara-RoeA ΔYenLC strain generated very high levels of intracellular YenTc that it was unable to secrete (Fig. 2a).

We investigated whether RoeA regulates YenLC and YenTc production as the observed phenotypes implied, by analysing the cytoplasmic contents of induced Ara-RoeA cells using mass spectrometry. Comparing these to wild-type and non-induced Ara-RoeA cells (Fig. 2c,d) revealed that, surprisingly, RoeA not only controls YenLC and YenTc production but also the numerous other toxins and virulence factors previously identified as highly enriched in the secreted fraction (Fig. 1a). This demonstrates that YenTc-expressing soldier cells, not another specialized cell subpopulation, are the source of this lethal protein cocktail. Promoter region analysis of RoeA-controlled genes revealed an overrepresented sequence, possibly the RoeA-binding site (Supplementary Fig. 1b). Importantly, another group independently verified the importance of YenLC and RoeA for lytic protein release and YenTc production during review of this study[44]. To avoid potential confusion, we have adopted YenLC component nomenclature from that study.

We imaged an induced Ara-RoeA YenA1-sfGFP strain, revealing conversion of the entire bacterial population into secretion-competent soldier cells (Fig. 2b and Supplementary Video 1), establishing RoeA as the key to both coupling production of YenTc to its secretion system and generating the soldier cell phenotype. To assess whether high intracellular levels of RoeA generate soldier cells also in a native context, we monitored a RoeA-sfGFP strain using confocal microscopy. Only the around 14% of explosive lysis-capable soldier cells (Fig. 2e and Extended Data Fig. 5) exhibited RoeA-sfGFP fluorescence.

Interestingly, inducing Ara-RoeA at 37 °C instead of 20 °C did not generate soldier cells (Supplementary Fig. 3a–e). This aligns with our previous findings that soldier cells are absent at higher temperatures (Extended Data Fig. 3m,n), implying a temperature-sensitive regulatory layer similar to TcaR2 of *Y. enterocolitica*, which is regulated by temperature-sensitive degradation[45]. As increased RoeA levels cause differentiation (Fig. 2e and Extended Data Fig. 5), we hypothesized that elevated temperatures affect RoeA production at mRNA or protein levels. Quantitative reverse transcription PCR (RT–qPCR) analysis of induced Ara-RoeA and RoeA-sfGFP strains (with disrupted 5′ or 3′ untranslated regions) at 16 °C or 37 °C revealed significantly reduced *roeA* transcript levels at 37 °C, unlike the unaffected control gene (Fig. 2f). This suggests a heat-repressible RNA thermostat within the *roeA* coding sequence similar to that of the cold-inducible *cspA* mRNA[46]. Thermosensitivity of *Y. entomophaga* differentiation thus occurs at the *roeA* transcript level, explaining soldier cell suppression in non-insect host conditions.

### The YenTc release mechanism

Using a HolA knockout strain, we demonstrated that PepB requires pH-induced HolA activation to reach the periplasm (Extended Data Fig. 6a,b). This dependence on holins is similar to bacteriophage holin/endolysin/spanin systems[30]. In the latter case, critical holin concentrations cause a collapse of the proton motive force (PMF) that activates holin pore formation[47–49], while here PMF collapse[50] and holin activation occur due to elevated pH. This means that while the initial triggers differ, the ultimate cause of holin activation in phages and YenLC is probably fundamentally similar.

To visualize the individual steps of YenLC-mediated YenTc release from soldier cells following pH triggering, we created Ara-RoeA strain derivatives blockable in the pre-secretion, post-holin, post-endolysin and post-spanin states by knocking out respective YenLC components. We vitrified the cells and prepared lamellae 50–100 nm thin using cryo-focused ion beam (cryo-FIB) milling[51] (Extended Data Fig. 7d–g) for the first three strains due to their thickness (Extended Data Fig. 7a), while the post-spanin state cells were imaged directly (Extended Data Fig. 7c). Notably, the 10–15 kDa YenLC components were too small to visualize in situ, therefore we instead analysed their impact on

cellular ultrastructure. Tomograms showed an intact cell envelope in the pre-secretion state (Fig. 3a,e and Supplementary Video 2). Notably, we found fully assembled YenTc holotoxins in the cytoplasm (Fig. 3a, inset) finally establishing it as the location of Tc holotoxin assembly.

Post-holin-state cells exhibit inner membrane perturbations in the form of invaginations not seen in ΔHolA cells (Fig. 3b,f, Supplementary Video 3 and Extended Data Fig. 7b). Notably, there were no micrometre-scale lesions or cytoplasmic material expulsion into the periplasm as seen for activated phage holin S105[47], even after examining intact cells (Extended Data Fig. 7a,b). Confocal microscopy confirmed the presence of HolA, revealing its distribution throughout the cell surface in nearly ubiquitous small foci (Extended Data Fig. 6c), reminiscent of the oligomeric 'rafts' of S105[52,49]. An alternative holin mechanism, not involving pore formation but instead membrane weakening/flipping for endolysin transport, has been proposed recently[53]. Given the lack of observed lesions, HolA might exemplify such a holin, making *Y. entomophaga* an excellent testbed to investigate this exciting hypothesis.

We examined post-endolysin-state cell tomograms to understand the impact of PepB. These cells display inner membrane detachment from the peptidoglycan layer and outer membrane, causing severe inner membrane bending (Extended Data Fig. 8a, Fig. 3c,g and Supplementary Video 4). In spanin-free conditions, PepB weakens the peptidoglycan sacculus to the extent that cells expand into spheroplasts due to internal osmotic pressure[54] (Extended Data Fig. 8b,c). Our tomograms reveal that spheroplasts have lower protein density due to increased volume, making them ideal for future in situ structural proteomics studies (Fig. 4, Extended Data Fig. 8d and Supplementary Fig. 4).

However, with spanins Rz and Rz1 present, bacteria dramatically transform into loosely bound clusters of unimembrane vesicles (Fig. 3d,h and Supplementary Video 5). Membranes in regions presumably housing cell envelope-spanning protein complexes remain discrete (Fig. 3d, right inset) and vesicle cell walls remain intact, indicating preservation of the lipoprotein Lpp tether of peptidoglycan to the former outer membrane (Fig. 3d). Importantly, YenTc holotoxins are released as a result of this metamorphosis (Figs. 3d (left inset), 4 and Supplementary Fig. 5), demonstrating how spanin-mediated fusion of the inner and outer membranes releases the cytoplasmic contents of soldier cells into the environment.

### T10SSs operate via a hitherto undescribed lytic mechanism

The novel lytic mechanism of T10SS-mediated YenTc release contradicts the non-lytic mode proposed for the archetypal T10SS in *S. marcescens*[32–34]. While our primary focus was to identify and characterize the Tc toxin secretion mechanism, we also wished to investigate the differences between these T10SSs. Using targeted genomic editing (Extended Data Fig. 2a), we created the arabinose-inducible *S. marcescens* Ara-ChiR strain analogous to *Y. entomophaga* Ara-RoeA. *S. marcescens* Ara-ChiR secretion differed from Ara-RoeA in two aspects: it was no longer inhibited by medium acidification and displayed an inverted temperature sensitivity profile (Fig. 5a), showcasing how differences in pathogen lifestyle influence the temperature of T10SS/cargo production and pH sensitivity of secretion. Chitinolytic enzymes notwithstanding, the *S. marcescens* Ara-ChiR secretome was full of typically non-exported proteins (Extended Data Fig. 9a (inset)), the release of which was dependent on the T10SS, as evidenced by their absence upon use of the T10SS knockout strain (Fig. 5a and Extended Data Fig. 9b,c). We controllably triggered the T10SS through anaerobiotic stress-induced PMF collapse (Fig. 5c and Supplementary Video 6) similarly to *Y. entomophaga* soldier cells (Extended Data Fig. 6b and Supplementary Video 1) and acquired cryo-electron tomograms for detailed visualization (Fig. 5d). The tomograms showed the same spanin-driven transformation as in *Y. entomophaga* (Fig. 5e and Supplementary Videos 7 and 8), with bacteria converting into unimembrane vesicles and expelling their

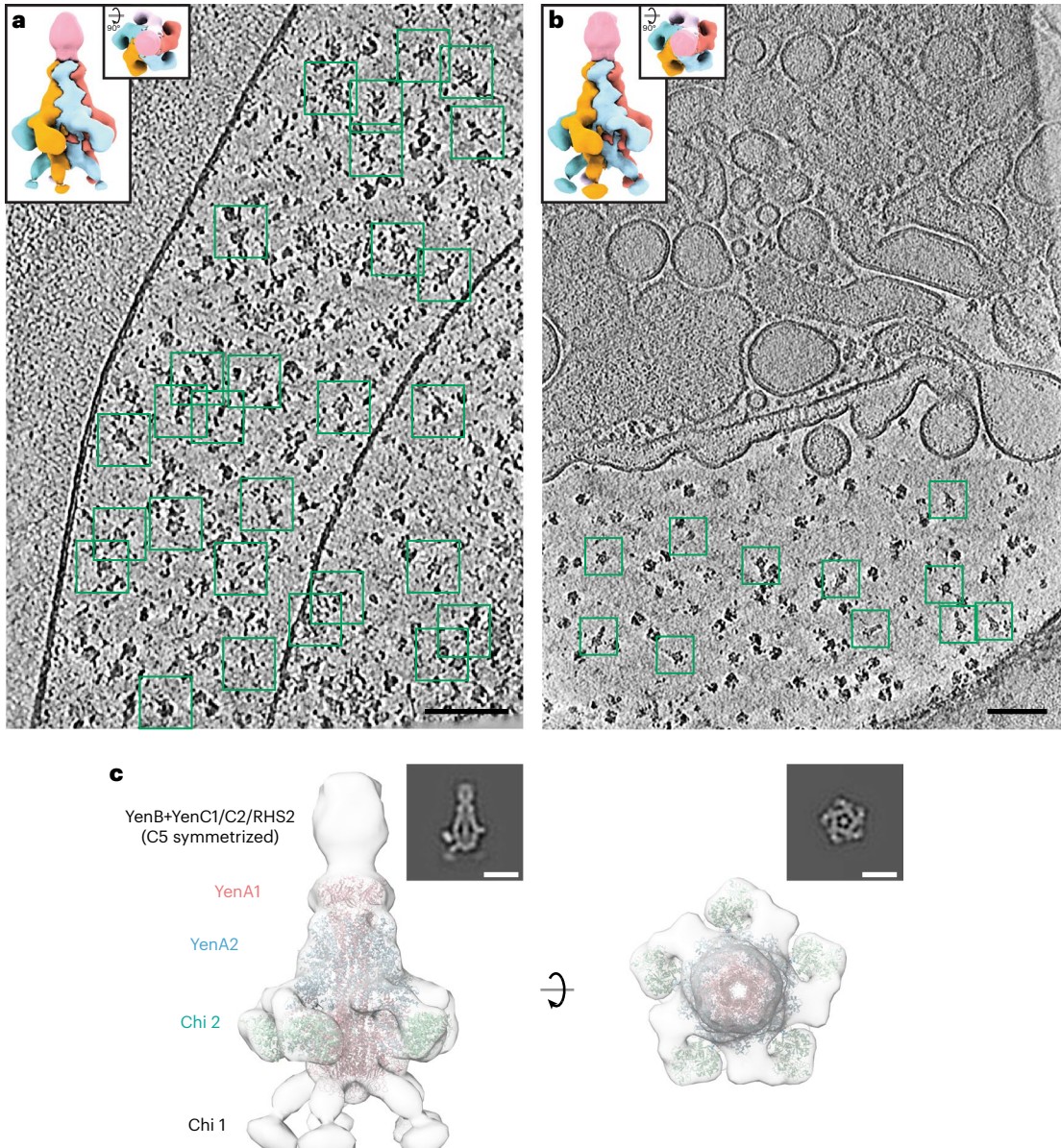

**Fig. 4 | Subtomogram averaging of YenTc before and after release by soldier cells. a**, A 6-nm-thick slice from a representative tomogram of Ara-RoeA ΔRz/Rz1 spheroplast cells. YenTc particles automatically picked to TomoTwin[89] are highlighted by green boxes. Scale bar, 100 nm. $n$ = 7 biological replicates. Inset: the structure of YenTc from these cells with individually coloured protomers of TcA. The TcB/TcC components are coloured in pink. **b**, A 9-nm-thick slice from a representative tomogram of Ara-RoeA cells. YenTc particles manually picked for subtomogram averaging are highlighted by green boxes. Scale bar, 100 nm. $n$ = 12 biological replicates. Inset: the structure of YenTc released into the extracellular space, coloured as in **a**. **c**, Side and top views of the YenTc structure fitted with a structural model of the YenTc TcA component (PDB: 6OGD)[15]. Insets are cross-sections of the structures. Scale bar, 20 nm. $n$ = 167 particles.

contents for rapid release of chitinolytic machinery. This confirms that T10SSs mediate lytic protein release, contrary to the non-lytic fashion previously proposed[33,55], providing a much-needed explanation of how T10SS-exported proteins cross the outer membrane after maturation[32] and why only a minor fraction of *S. marcescens* cells express the T10SS[33,34]. This supports our earlier finding that Tc toxin release in *Y. entomophaga* occurs through the newly described lytic action of the T10SS YenLC. Owing to these parallels, we propose to name the *S. marcescens* T10SS the *S. marcescens* lytic cassette (SmaLC) for consistency.

## Discussion

In this study, we have discovered that the Gram-negative insect pathogen *Y. entomophaga* releases the Tc toxin YenTc and other virulence factors into the environment using the YenLC T10SS. Our observations have led us to propose the model for soldier-cell differentiation and YenLC-mediated protein release shown in Fig. 6.

Furthermore, we show that T10SSs mediate protein release by cell lysis rather than non-lytically as previously hypothesized[32,33] (Fig. 5 and Extended Data Fig. 9), and T10SSs are therefore likely descendants of phage lysis cassettes that bacteria repurposed for their own needs. This makes our cryo-ET data, particularly on the previously unobserved endolysin- and spanin-driven release stages, relevant to the field of bacteriophage biology. The fundamental difference of the mechanism we demonstrate compared with lytic release of, for example, bacteriophage-encoded Shiga toxins is that in our case, toxin production and release by differentiated cells is not a by-product of the phage infection cycle and is therefore SOS response insensitive[29]

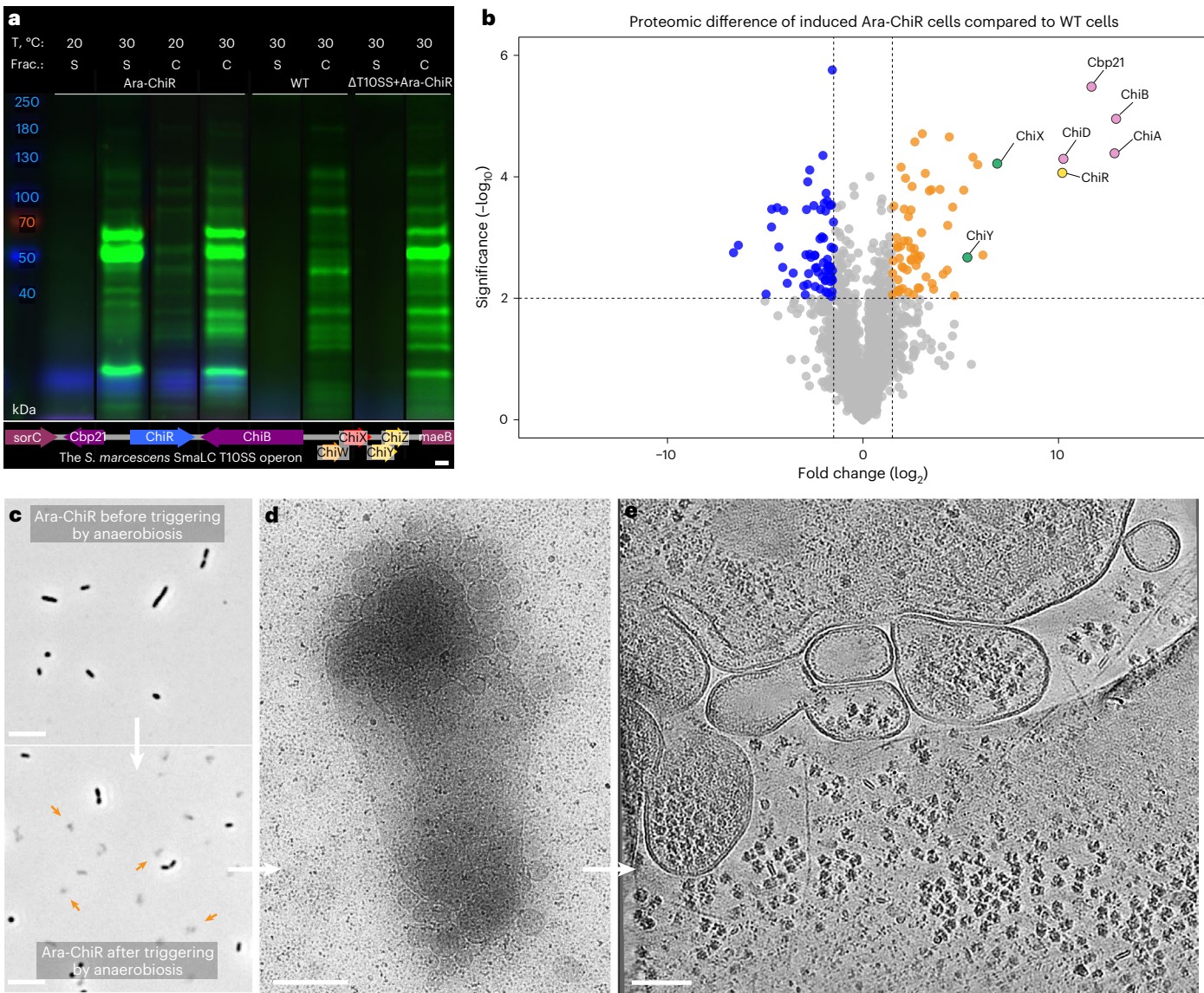

**Fig. 5 | The *S. marcescens* T10SS exports proteins via a lytic mode of action.**
**a**, Arabinose induction of the Ara-ChiR strain (but not wild-type cells) causes temperature-sensitive production of chitinolytic proteins, with secretion only occurring in the presence of the SmaLC T10SS. S, secreted fraction. C, cytoplasmic fraction. Inset: structure of the SmaLC operon, composed of a holin (ChiX), endolysin (ChiY), i-spanin (ChiY) and o-spanin (ChiZ), and the bimodal expression regulator ChiR[33,34]. Data are consolidated. Scale bar, 200 bp. *n* = 3 biological replicates. **b**, Volcano plot showing significant enrichment (*t*-test *P* < 0.01) of chitinolytic and T10SS components in the total protein fraction of induced Ara-ChiR cells compared with ChiR-non-expressing wild-type cells, revealing the previously unreported control of chitinase ChiD expression by ChiR. Pink, chitinolytic components; green, structural components of the SmaLC T10SS; orange, enriched intracellular proteins; yellow, the regulatory protein ChiR. The relevant UniProt accession numbers are provided in Methods. *n* = 3 biological replicates each. The full proteomic datasets used to generate these figures are available in the Source Data. **c**, Aerobic shaking cultures of Ara-ChiR

cells can be stimulated to undergo SmaLC-mediated secretion by immobility-induced anaerobiosis. Orange arrowheads denote examples of cells that already underwent SmaLC-mediated secretion. Lysed cells: 67 (8%) of 798 total cells analysed before triggering by anaerobiosis and 1,413 (59%) of 2,393 total cells analysed after triggering by anaerobiosis. Scale bars, 10 μm. *n* = 3 biological replicates. **d**, A TEM overview image of an Ara-ChiR cell after SmaLC-mediated secretion. Scale bar, 500 nm. *n* = 141 biological replicates. **e**, Tomogram slice of an Ara-ChiR cell after secretion, corresponding to a post-spanin action state. Note the similarity to the phenotype observed for *Y. entomophaga* after YenLC spanin action in Fig. 3d and Extended Data Fig. 7c, including formation of unimembrane vesicle clusters and release of cytoplasmic proteins into the environment. The observed SmaLC-mediated lysis explains the abundance of cytoplasmic and membrane proteins previously observed in the supernatant of *S. marcescens* strain Db10 cells, which was at that time attributed to sensitivity of the mass spectrometry technique[33]. Scale bar, 100 nm. *n* = 141 biological replicates.

(Fig. 4l,n). Since they release transcriptionally coupled cargo proteins via lysis, T10SSs are not a secretion system in the classically used sense of the term. However, we do support the use of the term type 10 secretion system with respect to these specialized lysis cassettes since they function as secretion systems at the population level by releasing beneficial cargo proteins synthesized by the small subset of T10SS-expressing cells.

YenTc appears to be the first example of an anti-eukaryotic toxin using this newly established type of secretion system[32], which enables extremely rapid release of virulence factors of any size including those previously not associated with any known secretion pathway. Soldier cells, bearing the brunt of YenLC-mediated protein release, represent a specialized minority. Factors influencing their differentiation remain unclear, but nutrient-, quorum- and temperature-sensing are important

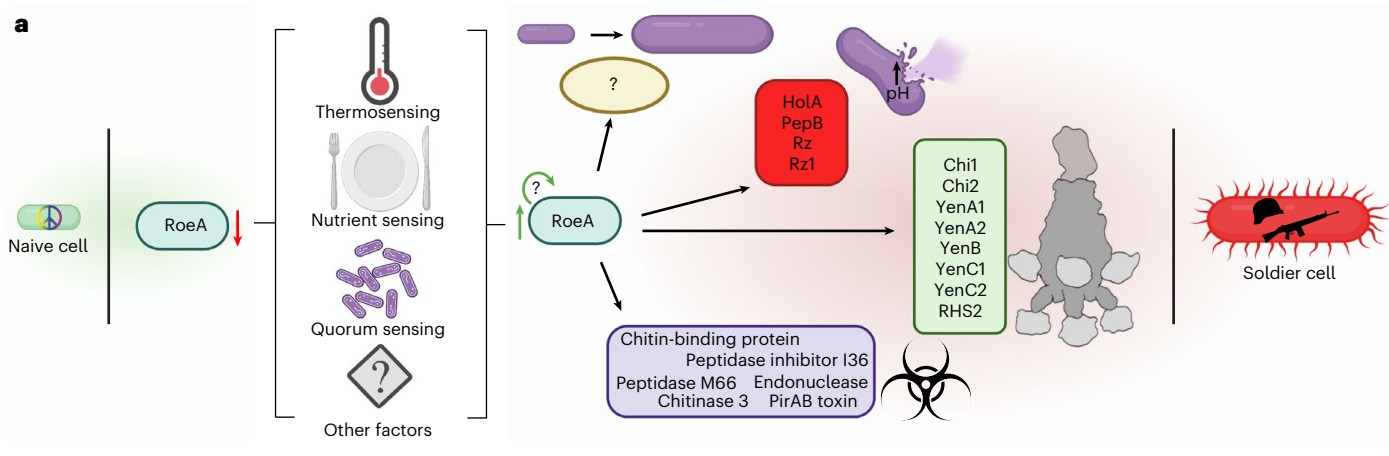

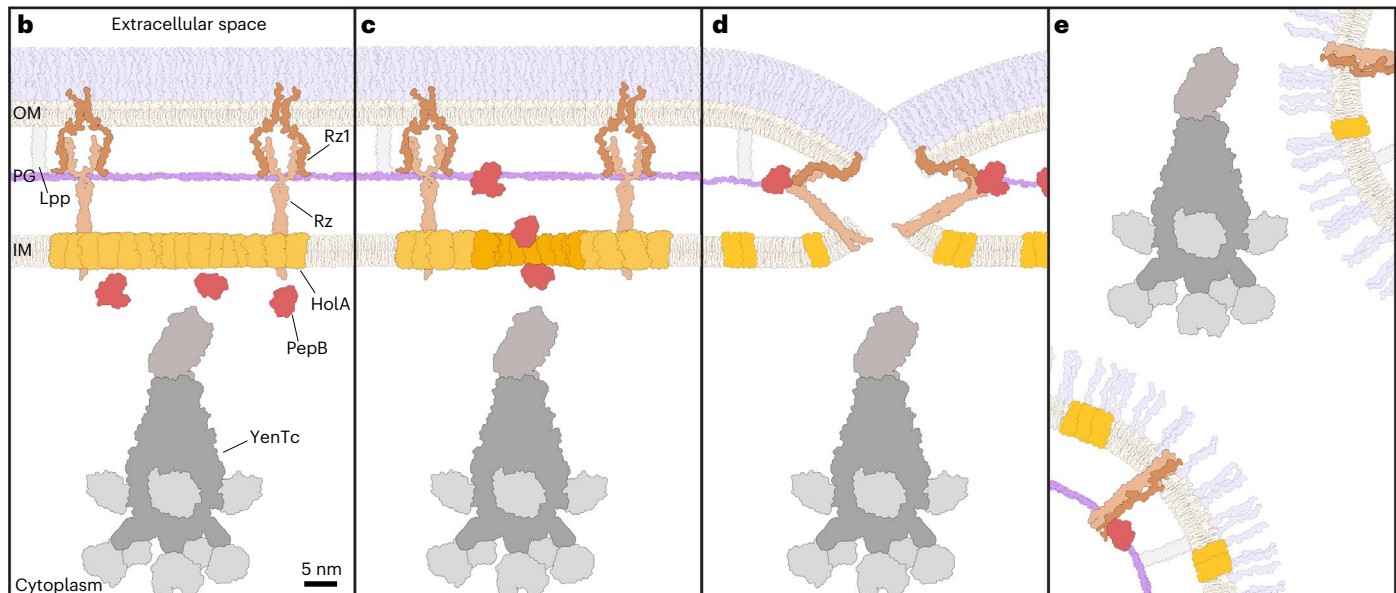

**Fig. 6 | Model for soldier-cell differentiation and subsequent YenLC-mediated YenTc release. a**, At the lower temperatures that are conducive to *roeA* mRNA stability, synergy of several environmental cues causes an increase of RoeA levels in a subset of cells, possibly aided by an autoregulatory positive feedback loop as has been shown for bimodal expression of the *Bacillus subtilis* master spore regulator SpoOA[39]. This leads to differentiation into soldier cells by RoeA-mediated transcription of YenTc components (green box), YenLC components (red box), additional toxins and virulence factors (purple box), and the characteristic enlarged phenotype of soldier cells by affecting a yet undetermined set of genes. **b**, During RoeA-mediated differentiation, soldier cells produce numerous toxins including YenTc, which is fully assembled in the cytoplasm. Components of the T10SS YenLC are produced simultaneously. PepB remains in the cytoplasm, HolA assembles into oligomeric rafts in the inner

membrane (IM) and the inner membrane-embedded Rz forms a complex with the outer membrane (OM)-embedded Rz1. **c**, pH elevation that probably deteriorates the proton motive force either triggers the formation/opening of small pores in the HolA rafts or enables pore-free HolA-mediated transmembrane flipping[53] of PepB, allowing PepB to reach the periplasm and cleave peptidoglycan crosslinks[43]. **d**, Once enough crosslinks have been cleaved[95], the spanin complex is no longer hindered by the peptidoglycan sacculus and can undergo a conformational change that brings the inner and outer membranes into direct contact, leading to the fusion of the outer and inner membranes. **e**, The fused membranes collapse into loosely bound clusters of vesicles still tethered to peptidoglycan via the protein Lpp. The accompanying release of inner osmotic pressure[54] propels YenTc and other soldier cell toxins into the surrounding environment, where they can proceed to locate and act on their cellular targets.

(Extended Data Fig. 3, Supplementary Fig. 3 and Fig. 2f). RoeA acts as the central temperature-sensitive differentiation switch that synchronizes secretion system production with toxic cargoes (Fig. 2), ensuring that the enlarged sacrificial soldier cells have as much carrying capacity and are packed with as many host-damaging factors as possible (Figs. 1c and 2b,e). The net result is a *Y. entomophaga* population demonstrating traits such as differentiation and altruism, reminiscent of eusocial systems, a far cry from the 'bags of enzymes' that bacteria were once considered to be[56].

Differentiation of soldier cells probably activates upon ingestion in response to insect host nutrients (Extended Data Fig. 3). The pH sensitivity of secretion would delay toxin release until the soldier cells reach the alkaline posterior midgut, their major theatre of operations[38].

The chitinases and chitin-binding proteins of the released toxic cocktail could then breach the chitinous peritrophic membrane[57] and enable YenTc and other toxins to access and devastate the underlying midgut epithelium, enabling host colonization by the non-differentiated *Y. entomophaga* population.

Notably, T10SSs can be found in Tc toxin operons of major pathogens such as *Y. pseudotuberculosis*, *Y. enterocolitica* and *Y. pestis*, as well as opportunistic pathogens such as *Salmonella enterica* subsp. *houtenae* (Extended Data Fig. 4), indicating that they very likely employ the same secretion mechanism as *Y. entomophaga*. Whether these and other pathogens make use of specialized lysis-capable cell subpopulations for Tc toxin release in an in vivo setting remains a topic for future research. However, this has already been hinted at

by two very recent studies showing that an intact *Y. enterocolitica* T10SS is as essential for establishing infection in insect and nematode models as is the presence of a functional *Y. enterocolitica* Tc toxin[58,59]. In this light, our finding that an ancient *Y. pestis* Tc toxin operon from a Black Death victim[60] also encodes a T10SS (Extended Data Fig. 4d inset) raises questions about its role in the deadliest pandemic of human history[61].

The T10SS-mediated release of Tc toxins fits into the emerging paradigm of anti-eukaryotic toxin export by phage-derived proteins[53]. Other examples include the holin and/or endolysin-mediated export of toxins such as *Clostridium difficile* toxins A and B[62–64], large clostridial glucosylating toxins[65,66] and typhoid toxin[67], currently generally thought to occur via non-lytic mechanisms[53,62,65–67]. In light of the data presented here, the potential use of suicidal soldier cell subpopulations by these pathogens, however, warrants further investigation. Deletion of YenTc produced by only few cells in a population causes full loss of *Y. entomophaga* virulence[7]. This leads to the startling insight that pathogen virulence can be determined by a small number of specialized soldier cells. If this is indeed found to be a more widespread phenomenon, then medical interventions specifically targeting such specialized subpopulations may be a promising treatment strategy for bacterial diseases.

## Methods

### Bacterial strains and constructs
*Y. entomophaga* strains, *S. marcescens* strains and plasmids used in this study are provided in Supplementary Table 1.

### Cell growth and secretion assay conditions
*Y. entomophaga* type strain MH96[68] and *S. marcescens* type strain BS 303 (also known as strain ATCC 13880) were obtained from the German Collection of Microorganisms and Cell Cultures (DSMZ). To test secretion or *Y. entomophaga* in different growth medium, 10 ml of these medium was inoculated with 500 µl exponential-phase culture grown in LB, then grown for 16 h at 16 °C. The medium was separated from the cells by 5 min of centrifugation at 4,000$g$ and 0.22 µm filtration. Amicon concentrators (Merck) were used for concentrating the medium to 250 µl. All samples were then normalized with PBS to a volume equivalent to a cell optical density ($OD_{600}$) of 4.0 and analysed by Mini-PROTEAN stain-free SDS–PAGE gels (Bio-Rad) on a ChemiDoc MP imaging system (Bio-Rad) using ImageLab 5.2.1 (Bio-Rad). For secretion assays, 20 ml of cells were grown in SOC growth medium for 16 h at 20 °C. Then the pH was elevated either by drop-wise addition of NaOH/a fixed volume of 1 M Tris-HCl pH 8.0, or resuspension of spun-down cells in 100 mM Tris-HCl pH 8.0 and 100 mM NaCl buffer. *S. marcescens* cells were originally grown in M9, LB or SOC medium at 20 or 30 °C, with SOC at 20 or 30 °C used for later experiments. Since the wild-type variant of the *S. marcescens* type strain does not express the ChiR T10SS/chitinolytic machinery regulator under the laboratory conditions tested, to switch on its production we either replaced the native ChiR promotor with arabinose-inducible machinery (in the case of the Ara-ChiR strain) or transformed cells with an arabinose-inducible Ara-ChiR plasmid (in the case of the SmaLC knockout strain), similar to what was earlier done by use of an arabinose-inducible plasmid[34]. T10SS components of the type strain BS 303 used here have 95–99% sequence identity to those of the publicly unavailable strain Db10 investigated previously[33]. SOC medium acidifies to the same extent in both *Y. entomophaga* and *S. marcescens* cultures. Ara-RoeA and Ara-ChiR containing strains and the Ara-PepB and Ara-ChiR plasmids were induced by addition of 0.5% L-arabinose during inoculation. For anaerobiosis-induced secretion, 50 ml induced and acidified cell cultures were left stationary after vigorously shaking during growth for up to 20 min in the case of *Y. entomophaga* Ara-RoeA and up to 60 min in the case of *S. marcescens* Ara-ChiR.

### Targeted genomic editing of *Y. entomophaga* and *S. marcescens*
Cells were initially electroporated with an editing plasmid encoding λ-RED recombineering proteins, an I-SceI endonuclease, as well as a Cas9 protein plus a guide RNA (gRNA) that has minimal off-target specificity in *Y. entomophaga* as determined by Cas-OFFinder[69]. The cells were then transformed with a donor plasmid encoding an I-SceI cleavable fragment flanked by ~300 bp homologous to the 5′ and 3′ sequences of the targeted area, containing an antibiotic resistance marker, a SacB counterselection marker, a 20 bp Cas9-gRNA targeting region and a short repeat of the 5′ sequence. This setup allowed modification of targeted genomic areas with optional subsequent Cas9-mediated excision of the antibiotic resistance marker[70], facilitating multiple sequential targeted genomic edits. The cells were then plated onto LB agar plates containing antibiotics for selection of the editing plasmid antibiotic resistance marker (kanamycin), the donor cassette antibiotic resistance marker (chloramphenicol) and 10% sucrose as a counterselection agent. Surviving colonies were then re-streaked onto an identical plate to eliminate background, and success of the genomic editing was validated by colony PCR and sequencing.

### Non-cryogenic cell imaging
For routine phase contrast and fluorescent imaging, 1 µl of *Y. entomophaga* or *S. marcescens* strain cultures was imaged on a glass slide at ×20 using an EVOS M7000 microscope (Thermo Fisher). Cell counting analyses were carried out in Fiji[71]. Confocal fluorescence microscopy timelapses of YenA2-sfGFP, as well as RoeA-sfGFP and Ara-RoeA HolA-mCherry ΔPepB/Rz/Rz1 cells before and after pH 8.0-induced secretion triggering were done on a glass slide at ×40 using an LSM800 microscope (Zeiss) equipped with an Airyscan detector module (Zeiss).

### Fluorescence spectroscopy
For measuring the effect of different environmental conditions and quorum system knockouts on YenTc production, three biological replicates of YenA1-sfGFP cells with or without genomic knockouts of the acyl-homoserine-lactone synthase (for autoinducer-1 knockout), the *S*-ribosylhomocysteine lyase (for autoinducer-2 knockout) and the L-threonine 3-dehydrogenase (for autoinducer-3 knockout) were grown in the conditions specified in Extended Data Fig. 3. These were collected, resuspended to $OD_{600} = 1.0$ in PBS and used to measure the pre-secretion content of cellular YenA1-sfGFP. After 20 min incubation time, the cells were spun down and the supernatant was used to measure the content of secreted YenA1-sfGFP. Fluorescence emission spectra were recorded on a Spark spectrophotometer (Tecan) using an excitation wavelength of 470 nm and an emission wavelength of 518 nm in a 2 × 2 read mode.

### Proteomic analysis using nanoHPLC–MS/MS
The following sample types normalized to $OD_{600} = 4.0$ were used for mass spectrometric analyses: pre-secretion cytoplasmic contents from cells, secreted protein fractions obtained by incubation of cells in secretion-inducing conditions and subsequent removal of non-secreted material by centrifugation at 4,000$g$ for 5 min with 0.2 µm filtration of the resulting supernatant, or post-secretion protein fractions consisting of said non-secreted material that was washed once before resuspension in the original sample volume. Biological triplicates of these sample types were briefly run on a stain-free SDS–PAGE gel (Bio-Rad). The same was done for YenTc purified by a combination of immobilized metal affinity and size exclusion chromatography techniques. After tryptic digestion and purification, the protein fragments were analysed by nanoHPLC–MS/MS by using an Ultimate 3000 RSLC nanoHPLC system and a Hybrid-Orbitrap mass spectrometer (Q Exactive Plus) equipped with a nanospray source and operated via

Xcalibur 4.0.27.10 (all from Thermo Fisher). In brief, the lyophilised tryptic peptides were suspended in 20 µl 0.1% trifluoroacetic acid) and 3 µl of the samples were injected onto and enriched on a C18 PepMap 100 column (5 µm, 100 Å, 300 µm ID × 5 mm; Dionex) using 0.1% trifluoroacetic acid at a flow rate of 30 µl min⁻¹ for 5 min. Subsequently, the peptides were separated on a C18 PepMap 100 column (3 µm, 100 Å, 75 µm ID × 50 cm) using a linear gradient, starting with 95% solvent A/5% solvent B and increasing to 30% solvent B in 90 min using a flow rate of 300 nl min⁻¹ followed by washing and re-equilibration of the column (solvent A: water containing 0.1% formic acid; solvent B: acetonitrile containing 0.1% formic acid). The nanoHPLC apparatus was coupled online with the mass spectrometer using a standard coated Pico Tip emitter (ID 20 µm, Tip-ID 10 µM; New Objective). Signals in the mass range of 300 to 1,650 $m/z$ were acquired at a resolution of 70,000 for full scan, followed by up to 10 high-energy collision-dissociation MS/MS scans of the most intense at least doubly charged ions at a resolution of 17,500.

Relative protein quantification was performed by using MaxQuant (v.2.0.3.1)[72], including the Andromeda search algorithm and searching the *Y. entomophaga* proteome of the UniProt database (downloaded January 2022). In brief, an MS/MS ion search was performed for enzymatic trypsin cleavage, allowing two missed cleavages. Carbamidomethylation was set as a fixed protein modification, and oxidation of methionine and acetylation of the N terminus were set as variable modifications. The mass accuracy was set to 20 ppm for the first search and to 4.5 ppm for the second search. The false discovery rates for peptide and protein identification were set to 0.01. Only proteins for which at least two peptides were quantified were chosen for further validation. Relative quantification of proteins was performed by using the label-free quantification algorithm implemented in MaxQuant, and the match-between-runs feature was activated.

Statistical data analysis of samples was performed using Perseus (v.1.6.14.0)[73]. Label-free quantification intensities were log transformed ($\log_2$) and replicate samples were grouped together. Proteins had to be quantified at least three times in at least one of the groups of a comparison to be retained for further analysis. Missing values were imputed using small normally distributed values (width 0.3, down shift 1.8 for the datasets involving the *Y. entomophaga* Ara-RoeA and *S. marcescens* strains; width 0.3, down shift 2.0 for the dataset assessing the *Y. entomophaga* secreted vs non-secreted protein fractions), and a two-sided *t*-test (significance threshold: $-\log_2$ fold change > 1.5 for all datasets; $P < 0.02$ for the datasets involving the *Y. entomophaga* Ara-RoeA strain; $P < 0.05$ for the dataset assessing the *Y. entomophaga* secreted vs non-secreted protein fractions; $P < 0.01$ for the datasets assessing the *S. marcescens* strains) was performed. Proteins that were statistically significant outliers were considered as hits. Volcano plots were generated using VolcaNoseR[74].

UniProt accession numbers (in parentheses) of statistically significant hits from *Y. entomophaga* according to the above criteria, which are of major biological importance to this study, are as follows: YenA1 (B6A877), YenA2 (B6A878), Chi1 (B6A876), Chi2 (B6A879), YenB (B6A880), YenC1 (B6A881), YenC2 (B6A882), RHS2 (A0A2D0TC51), Cpb (A0A3S6EXR6), PirA (A0A3S6F007), PirB (A0A3S6F043), Pil36 (A0A3S6F569), NucA (A0A3S6F4M5), Chi3 (A0A3S6F1Q8), StcE (A0A3S6EYX4), Tlh (A0A3S6F052), PepB (A0A3S6F4L4), Rz (A0A3S6F4Q6) and RoeA (A0A3S6F5G2). Such hits from *S. marcescens* are: ChiR (M4SHQ2), ChiX (A0A349ZDQ1), ChiY (A0A379YYR5), ChiA (A0A379Y6D9), ChiB (P11797), ChiD (A0A380ANW3) and Cbp21 (O83009).

## RT–qPCR detection
To quantify the effect of temperature on *roeA* transcript levels, three biological replicates of either *Y. entomophaga* Ara-RoeA or RoeA-sfGFP strain cells were grown at 16 °C or 37 °C in SOC medium overnight with shaking (in the presence of 0.5% l-arabinose for the Ara-RoeA

strain). A volume of culture corresponding to a final $OD_{600}$ of 0.1 was added to 750 µl TRIzol reagent (Thermo Fisher) per sample, and total RNA was purified according to manufacturer protocols. The RNA was then directly treated with the DNA-free DNAse treatment kit (Thermo Fisher) according to manufacturer protocols to remove any contaminating DNA from the total RNA preparations. RT–qPCR was carried out immediately afterwards in a CFX96 system (Bio-Rad) using a Power SYBR Green RNA-to-$C_T$ 1-step kit (Thermo Fisher), which also contains an RNase inhibitor and ROX dye for passive referencing. Primers (100 nM) with sequences CCCTCGCAAAGATTGTAATTCA and CACTGGTTAATCATGCGTCAA were used to target RoeA. Primers (100 nM) with sequences CCTTACATACTTCCAAACACCC and CCAAAACTGACTATCTGATGCG were used to target YmoA in the same samples, which served as a temperature-invariant control. Raw data were analysed using the CFX Manager software (Bio-Rad), with a threshold value of 605 relative fluorescence units (RFU) used to determine the quantification cycle ($C_q$) The acquired data were then assessed for significance using unpaired *t*-tests and further processed using Prism 9 (GraphPad).

## Bioinformatic analyses
Identification of secretion signal sequences for RoeA-controlled toxins and virulence factors was performed using SignalP (6.0)[75]. The sequence logo for the final 30 bp before the start codon of RoeA-controlled genes (see Fig. 2d) was calculated using WebLogo (3.7.12)[76] following multiple sequence alignments via Clustal Omega[77] with combined iterations and maximum guide tree/HMM iterations set to 5. The promoter region of the polycistronic YenLC structural component operon was also included in this analysis. For reconstruction of the YpeTc operon of ancient *Y. pestis* from a Black Death victim, genomic reads from Illumina run SRR341961 of the original study[60] were assembled into contigs using Ray Meta[78], which were in turn assembled into a scaffold using CSAR-Web[79] with the YpeTc operon of *Y. pestis* strain KIM10+ as a reference.

## Bacterial vitrification
Overnight cultures of *Y. entomophaga* strains were spun down for 4 min at 4,000$g$ and resuspended to an $OD_{600}$ of 20 in PBS buffer containing pre-washed 10 nm BSA-NanoGold tracer (Aurion). In the case of the Ara-RoeA strain, 10 µl was directly applied to glow-discharged Quantifoil R1/4 Au-SiO₂ 200 grids and incubated for 10 min in a Vitrobot Mark IV plunger (Thermo Fisher) set to 100% humidity and 22 °C before blotting and plunge freezing. Of the other three strains, 3 µl were applied to identically treated grids in identical plunger conditions following 30 min incubation in the PBS-NanoGold buffer. After a waiting time of 60 s, all grids were blotted from both sides for 32 s using a blot force of 5. After 0.5 s drain time, the grids were vitrified by plunge freezing into liquid ethane. For *Y. entomophaga* Ara-RoeA ΔHolA/Rz/Rz1 cells intended for whole-cell tomography, the $OD_{600}$ was adjusted to 5 and blot time to 10 s, and Quantifoil R2/1 Au-SiO₂ 200 grids were used. Overnight cultures of the *S. marcescens* Ara-ChiR strain at an $OD_{600}$ of 8.5 were transferred to a non-shaking Eppendorf tube for 1 h, 10 nm BSA-NanoGold tracer (Aurion) was added and the sample was then applied to glow-discharged Quantifoil R1/4 Au-SiO₂ 200 grids. Blotting conditions were identical to those used for *Y. entomophaga*. Grids containing *Y. entomophaga* Ara-RoeA strain cells and *S. marcescens* Ara-ChiR strain cells were clipped with standard AutoGrids (Thermo Fisher) and directly used for further data acquisition as the cells were thin enough without additional cryo-FIB milling.

## Lamella preparation by cryo-FIB milling
Grids containing *Y. entomophaga* Ara-RoeA ΔYenLC, Ara-RoeA ΔHolA/Rz/Rz1 or Ara-RoeA ΔRz/Rz1 strain cells were clipped in cryo-FIB-specific AutoGrids (Thermo Fisher) with alignment markers and a cut-out for milling at shallow angles. Clipped grids were transferred to an Aquilos 2 cryo-FIB/SEM dual beam microscope (Thermo Fisher).

Lamella preparation was performed as previously described[80]. In brief, after platinum sputter coating and deposition of metalloorganic platinum, clusters of bacterial cells were targeted for a 4-step milling procedure using decreasing ion beam currents from 0.5 nA to 50 pA. Milling angles of 6–10° relative to the grid were used. Lamellae were milled to a thickness range of 50–100 nm.

## Cryo-ET data acquisition

Once ready for imaging, all grids were transferred into a Titan Krios transmission electron microscope (TEM, Thermo Fisher) operated at 300 kV and equipped with a K3 camera and a BioQuantum energy filter (Gatan). Images were acquired with SerialEM[81]. Overview images were acquired at ×6,500 nominal magnification to identify regions for cryo-ET data acquisition at higher magnification. Images used as references for batch data acquisition were also acquired at this magnification. Tilt series were acquired at ×64,000 (pixel size 1.48 Å) for milled *Y. entomophaga* Ara-RoeA ΔYenLC, Ara-RoeA ΔHolA/Rz/Rz1 and Ara-RoeA ΔRz/Rz1 cells and at ×42,000 (pixel size 2.32 Å) for *Y. entomophaga* Ara-RoeA and *S. marcescens* Ara-ChiR cells using a script based on a dose-symmetric tilt scheme[82] at defoci ranging from −5 to −8 μm. The stage was tilted from −60° to +60° relative to the lamellar plane at 3° increments. Each tilt series was exposed to a total dose of 120–140 $e^-$ Å$^{-2}$. Tilt series for intact *Y. entomophaga* ΔHolA/Rz/Rz1 cells were acquired at ×33,000 magnification (pixel size 2.861 Å) with a Volta phase plate[83]. Before acquiring each tilt series, a new phase plate position was activated for 2 min. A dose-symmetric tilt scheme from −54° to +54° with a 3° increment was used for data acquisition. A defocus of −0.5 μm was applied. Each tilt series was exposed to a total dose of 77 $e^-$ Å$^{-2}$.

## Tomogram reconstruction and subtomogram averaging of YenTc

Acquired movie frames were motion-corrected and combined into stacks of tilt series using Warp[84]. The stacks were aligned and reconstructed using IMOD[85]. During alignment, patch tracking was used in tilt series of milled cells and fiducial-marker tracking was used in tilt series of non-milled cells. The tomograms were 4× binned and low-pass filtered to 60 Å or 100 Å for better visualization using EMAN[86]. Alternatively, denoising by an implementation of cryo-CARE[87] was used for the same purpose. To obtain the structure of YenTc, 167 particles were manually picked from 12 tomograms of Ara-RoeA cells. The subtomograms were extracted from 4× binned tomograms with a box size of 100 pixels (928 Å) using RELION (3.0)[88]. The subtomograms were aligned to a spherical reference and averaged over iterations with C5 symmetry in RELION (Fig. 4 and Supplementary Fig. 5).

Making use of the attractive properties of Ara-RoeA ΔRz/Rz1 cell spheroplasts, intracellular YenTc holotoxin particles were located in these cells using TomoTwin[89] (Supplementary Fig. 4) and used for structure generation via subtomogram averaging. Particles were automatically picked with a pre-release of TomoTwin (0.3)[89] using a clustering workflow. First, all 7 tomograms were rescaled to 15 Å to make the YenTc particles fit into the TomoTwin static box size of 37 × 37 × 37. Next, a single tomogram was embedded with the latest general model (v.052022), which resulted in 4,015,980 embedding vectors. To remove embedding vectors corresponding to background volumes, the median embedding vector was calculated and all embeddings that had a cosine similarity with the median embedding vector higher than 0 were discarded. From the remaining 527,451 embedding vectors, a two-dimensional uniform manifold approximation and projection (2D UMAP) was calculated (Supplementary Fig. 4), with the highlighted cluster corresponding to the YenTc particles. The average of all embedding vectors belonging to this cluster gave the reference embedding that was used for further picking. The six remaining tomograms were then embedded with TomoTwin and the reference used to locate the YenTc particles. Using a confidence threshold of 0.846 gave a total number of 528 particles.

The subtomograms were then extracted from 4× binned tomograms with a box size of 128 pixels (760 Å) using RELION 3.0. The subtomograms were aligned to a spherical reference and averaged over iterations with C5 symmetry in RELION (Fig. 4 and Supplementary Fig. 5).

## Ultrastructural analyses of vitrified bacteria

Several tomograms and cryo-EM projection images were analysed to draw conclusions on the ultrastructural changes of the bacterial cell envelope accompanying the action of various T10SS components, as presented using representative examples in Figs. 3–5 and Extended Data Figs. 7 and 8.

For the Ara-RoeA ΔYenLC strain, 20 cells in total were examined. Of these, 18 had a cell envelope configuration expected of a diderm bacterial species without any abnormalities, while in one case a small cytoplasmic vesicle was observed at the pole, and one had a slightly ruffled outer membrane. For the Ara-RoeA ΔPepB/Rz/Rz1 strain, 15 cells in total were examined. Of these, 11 displayed inner membrane invaginations and/or inner membrane budding into the periplasm, while 4 had an intact cell envelope configuration. For the Ara-RoeA ΔRz/Rz1 strain, 86 cells in total were examined. Of these, 80 displayed areas of large-scale inner membrane bending away from the peptidoglycan and outer membrane components of the cell envelope, and in some cases abnormally sharp-angled curves of the latter two components, while 6 had an intact cell envelope configuration. Of the 80 cells with altered morphology, the inner membrane bending events were exclusively polar in 17 cells, were exclusively localized at the sidewall in 16 cells and were observed at both the polar and sidewall regions in 20 cells, while 20 cells had changed into spheroplasts and 7 had compromised integrity. For the Ara-RoeA strain, 105 cells in total were examined. Of these, 95 collapsed into the characteristic morphology following spanin action with accompanying protein externalization, and 10 had an intact cell envelope configuration. The collapsed cells were predominantly composed of small unimembrane vesicles with an inner peptidoglycan lining. The majority of these vesicles had contact sites to each other, and in some cases, they were tightly associated in bacterial-shaped clusters. For the Ara-ChiR strain of *S. marcescens*, 141 cells in total were examined. Of these, 96 collapsed into a morphology similar to that of Ara-RoeA cells of *Y. entomophaga*, while 45 had an intact cell envelope configuration. *S. marcescens* Ara-ChiR cells generally presented the same ultrastructural changes as did *Y. entomophaga* Ara-RoeA cells, sans the presence of externalized YenTc toxins. At the same time, long thin protein filaments not seen in *Y. entomophaga* were found among the externalized proteins of *S. marcescens*.

## Tomogram annotation and figure production

To visualize the morphological changes accompanying YenLC component action as well as the position and assembly status of YenTc, densities of YenTc, the inner membrane, peptidoglycan layer, outer membrane and fused membranes (in Ara-RoeA tomograms) as well as exemplary cell envelope-spanning proteins (in Ara-RoeA tomograms) were annotated on a slice-to-slice basis in reconstructed tomograms using Dragonfly (Object Research Systems). Weaker densities were masked during this process, which culminated in the generation of 3D segmented volumes.

ChimeraX[90] was used for Fig. 4 and Supplementary Fig. 5. Figure 6a and Extended Data Fig. 2a were created using Biorender.com.

## Statistics and reproducibility

All experiments were carried out with at least 3 independent biological replicates. No statistical methods were used to pre-determine sample sizes but our sample sizes for each type of experiment are similar to those reported in previous publications[91–94]. Data distribution normality and equal variances were formally tested using Shapiro–Wilk and *F*-tests, respectively. Samples were assigned to various experimental groups on the basis of their strain (RT–qPCR, mass spectrometry

samples), growth temperature (RT–qPCR samples) or fraction of origin (mass spectrometry samples). Data collection and analysis were not performed blind to the conditions of the experiments. No data points were excluded from the analyses. Further details of the analyses used for RT–qPCR and proteomics data, including software used, are available in Methods and figure legends.

## Reporting summary

Further information on research design is available in the Nature Portfolio Reporting Summary linked to this article.

## Data availability

Source data (includes unprocessed SDS–PAGE gels, mass spectrometry proteomics data and raw data for graphs) are provided with this paper as Source Data and Supplementary Data. The raw mass spectrometry proteomics data have been deposited to the ProteomeXchange Consortium via the MassIVE partner repository with the dataset identifiers MSV00089961/PXD035561 (secreted fraction vs non-secreted fraction of *Y. entomophaga* cultures), MSV000089964/PXD035573 (induced Ara-RoeA versus wild-type or non-induced Ara-RoeA *Y. entomophaga*) and MSV000091191/PXD039813 (secreted fraction/pre-secretion fraction of induced *S. marcescens* Ara-ChiR versus *S. marcescens* wild-type or induced ΔSmaLC *S. marcescens*). YenTc cryo-ET structures from the post-endolysin and post-spanin states have been deposited in the Electron Microscopy Data Bank (EMDB) under accession numbers EMD-16618 and EMD-15403, respectively. Representative tomograms for *Y. entomophaga* are deposited under accession numbers EMD-15404 (pre-secretion state), EMD-15405 (post-holin state, FIB-milled), EMD-16619 (post-holin state, intact cells), EMD-15406 (post-endolysin state) and EMD-15407 (post-spanin state). Representative tomograms for *S. marcescens* in the post-spanin state are deposited under accession numbers EMD-16538 and EMD-16539. A conversion of our original strain nomenclature used during data deposition (July 2022) to the one adopted in this manuscript during review has been provided in the Supplementary Data for ease of interpretation. No custom code was used in the analysis of the data. Biological materials such as plasmids are available from the corresponding author upon request.

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

## Acknowledgements

We thank K. Vogel-Bachmayr and S. Bergbrede for wet lab technical support; O. Hofnagel and D. Prumbaum for electron microscopy technical support; and A. Brockmeyer and W. Hecker for mass spectrometric technical support. Special thanks go to P. Njenga Ng'Ang'A, W. Oosterheert and G. Rice for fruitful discussions. This study was supported by funding from the Max Planck Society (to S.R.).

## Author contributions

O.S. and S.R. initiated and designed the project. O.S. carried out most of the experiments. Z.W., O.S. and T.W. acquired and analysed tomographic data. P.J. analysed mass spectrometric data. L.K. and O.S. generated bacterial strains. S.R. supervised the project. O.S. and S.R. wrote the manuscript with input from all co-authors.

## Funding

## Competing interests

O.S. and S.R. are inventors on a filed patent application (PCT/EP2023/074766) for producing insecticidal toxins using the described bacterial strains. The remaining authors declare no competing interests.

## Additional information

**Extended data** is available for this paper at https://doi.org/10.1038/s41564-023-01571-z.

**Correspondence and requests for materials** should be addressed to Stefan Raunser.

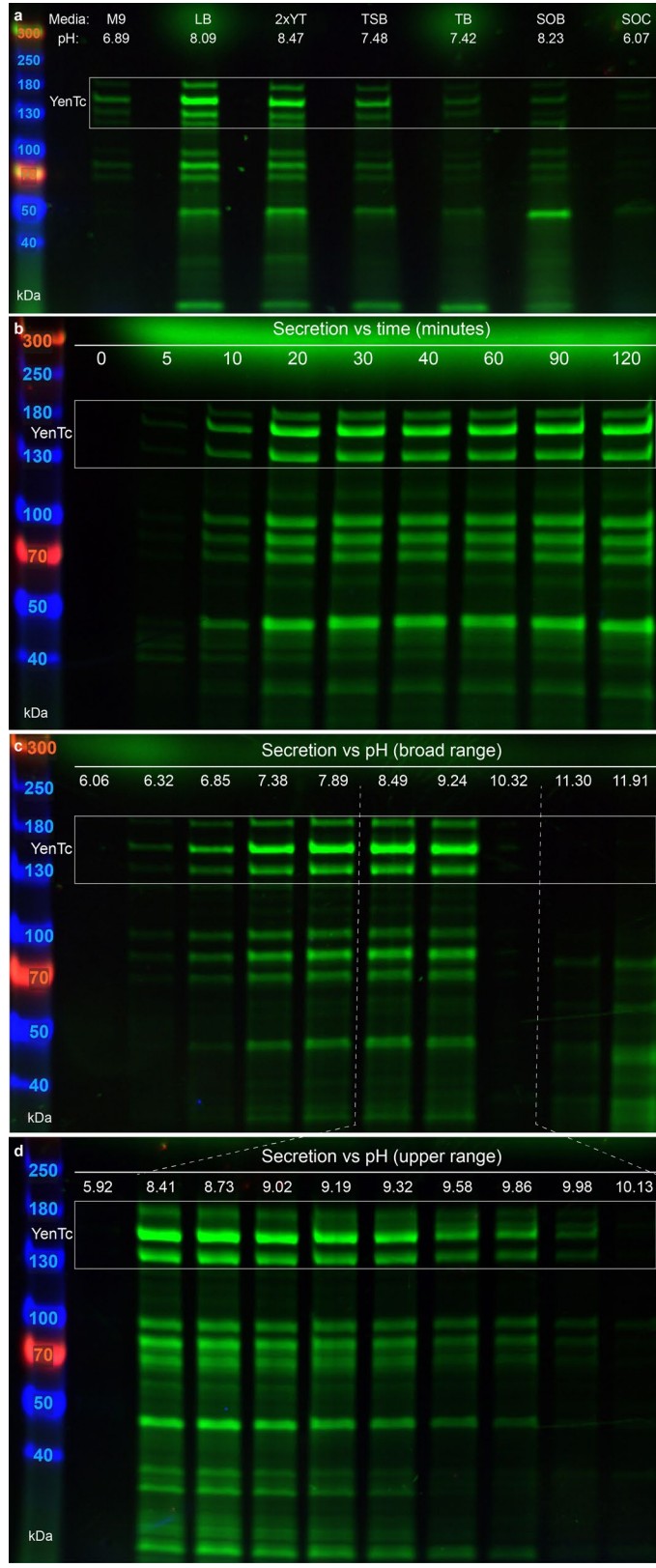

**Extended Data Fig. 1 | *Y. entomophaga* rapidly secretes proteins in a pH-dependent manner. a**, Secreted protein fractions from *Y. entomophaga* cultured in various growth media. Secretion is inhibited in media that acidifies during cell growth. Bands corresponding to YenA1, YenA2 and YenB are boxed for clarity (see Supplementary Fig. 2 for more details). n = 3 biological replicates. **b**, *Y. entomophaga* cells grown in acidified SOC media rapidly secrete when pH is adjusted to 8.0. n = 3 biological replicates. **c**, Broad range screening of the pH dependence of secretion, assessed by adjusting the pH of acidified SOC media to the values indicated. Non-specific cell lysis seen in the two most alkaline samples demonstrates the upper limit of *Y. entomophaga* pH tolerance. n = 3 biological replicates. **d**, Narrow range screening of the pH dependence of secretion to determine the upper pH limit of secretion. n = 3 biological replicates.

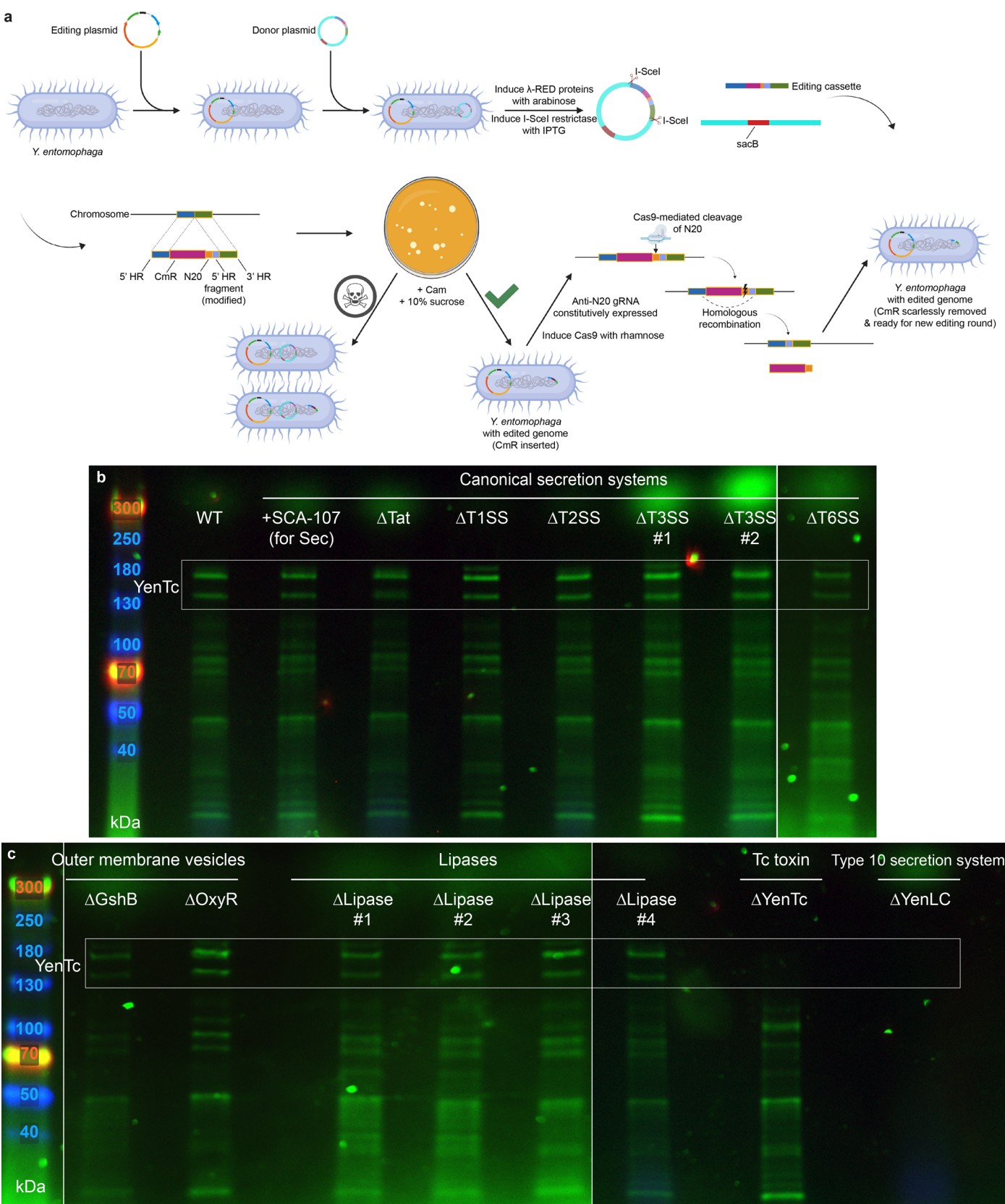

**Extended Data Fig. 2 | See next page for caption.**

**Extended Data Fig. 2 | A T10SS was identified as the major pathway for
*Y. entomophaga* protein secretion. a**, Schematic representation of the
targeted scarless genomic editing protocol established for *Y. entomophaga*.
The same protocol was also used to modify the genome of *S. marcescens*. **b-c**, *Y.
entomophaga* knockout strains of established anti-eukaryotic toxin secretion
pathways (**b**) or previously proposed Tc toxin export pathways (**c**, final lane
notwithstanding) have WT-like secretion levels, while the knockout of YenLC, a
novel T10SS, abolished protein export completely under the conditions tested.
The Sec export pathway is essential and non-removable and was therefore
blocked by the specific chemical inhibitor SCA-107[96]. While no single gene
controlling OMV formation is known, absence of OxyR and GshB very strongly
decreased *E. coli* OMV formation levels[97]. *Y. entomophaga* does not encode a
direct Pdl1 lipase homologue that was proposed to be a *Photorhabdus* Tc toxin
release factor, so four proteins with lipase⁻like domains that might serve as
a proxy were targeted, as was YenTc itself to see if it has autotransporter-like
capabilities. Data is consolidated. Bands corresponding to YenA1, YenA2 and
YenB are boxed for clarity. n = 3 biological replicates.

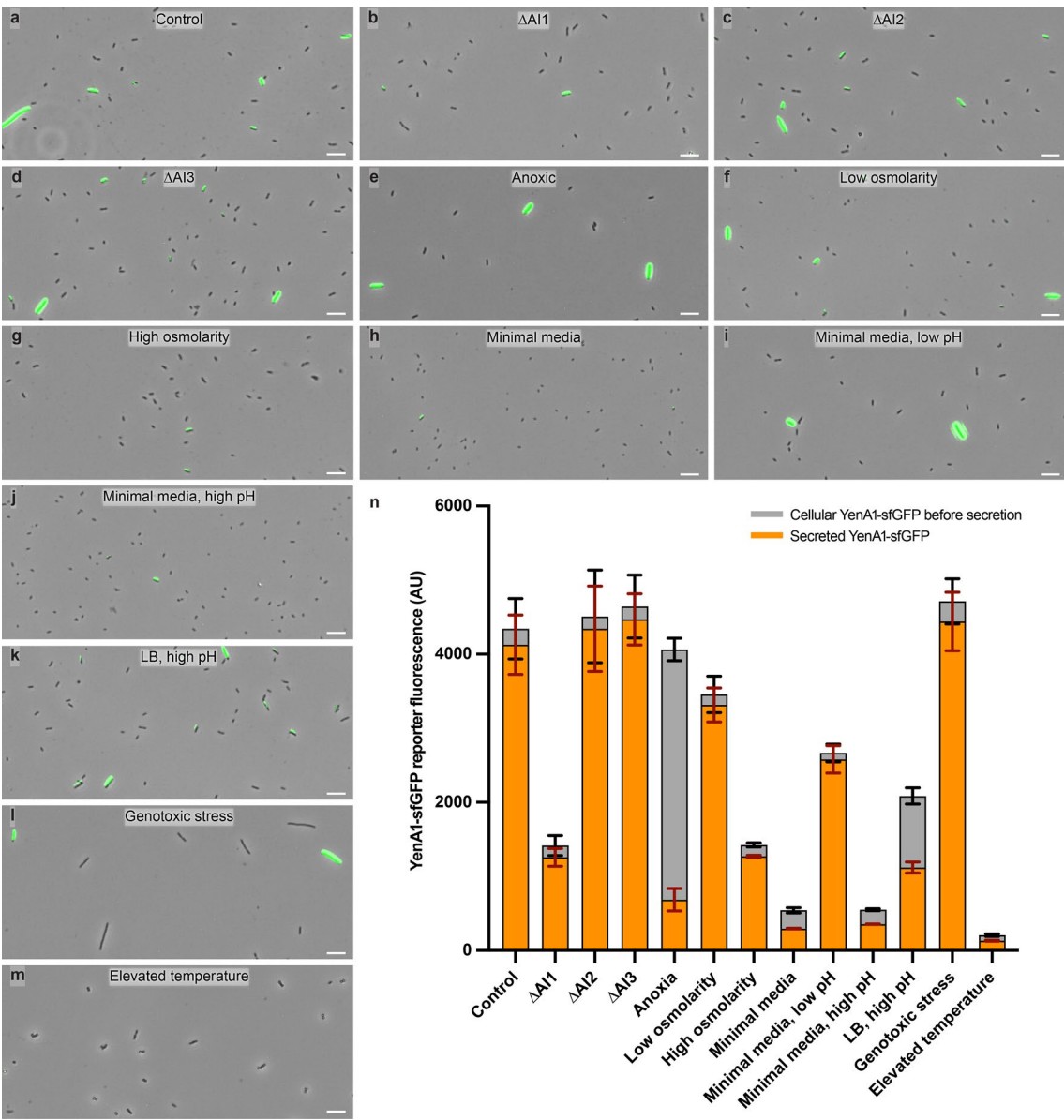

**Extended Data Fig. 3 | Differentiation into YenTc-producing cells is primarily controlled by thermosensing, nutrient availability, and autoinducer-1 quorum sensing molecules. a**, A genomic YenTc reporter fusion (via YenA1-sfGFP) was used as a qualitative marker of differentiation into YenTc-producing cells (green). *Y. entomophaga* was grown in the acidifying growth media SOC at 20 °C unless noted otherwise. **b-d**, Knockout of the acyl-homoserine-lactone synthase (UniProt accession number: A0A3S6EWZ2), the S-ribosylhomocysteine lyase (UniProt accession number: A0A3S6F351), and the L-threonine 3-dehydrogenase (UniProt accession number: A0A3S6EZE1) to test for the essentiality of autoinducer-1, -2 and -3 type quorum sensing molecules in differentiation, respectively. Absence of autoinducer-1 molecules decreases the number of YenTc-producing cells noticeably. **e**, Growth in a pure nitrogen atmosphere to test the effect of anoxia on differentiation. Interesingly, anoxically grown cells demonstrate defective YenTc reporter secretion, but unimpaired YenTc reporter production (**n**). **f-g**, Growth in SOC media prepared with 0 or 570 mM NaCl to test the effect of decreased and increased osmolarity on differentiation. Normal SOC media contains 190 mM NaCl. **h**, Growth in M9 minimal media to test for the essentiality of complex organic molecules for differentiation. In the absence of complex organic molecules, the cells tend to adopt minimal dimensions and strongly reduce differentiation into YenTc-producing cells. **i-j**, Same as in (**e**) but pre-adjusted to pH 6.10 / pH 7.70 to

respectively test for the influence of acidic and alkaline pH on differentiation in absence of complex organic compounds. Acidic, but not alkaline pH, stimulates production of YenTc (**n**). **k**, Growth in LB media, which increases its pH value to 7.50 after cell growth, to test how alkaline pH in presence of complex organic compounds affects differentiation compared to (**a**) and (**j**). **l**, Growth in media supplemented with 200 ng/mL mitomycin C to test the effect of genotoxin stress on differentiation and the relevance of the SOS response pathway, which tends to generate enlarged cells by inhibition of cell division. **m**, Growth at 37 °C to test the effect of elevated temperature on differentiation. Given the lack of observed YenTc expression via the sfGFP reporter, of all factors tested here high temperature affects the propensity of naive cells to differentiate the most. Scale bars: 10 µm. **n**, Differentiation into YenTc-producing cells (grey bars) and their secretory ability (superimposed orange bars) in conditions from (**a-m**) was quantified by measuring fluorescence of the YenA1-fused sfGFP in the pre-secretion and secreted fractions, respectively, of density-normalized cells. Data is shown as mean ± standard deviation, n = 3 biological replicates. For **a-m**, the respective number of total cells/ GFP+ ce-lls /% of GFP+ cells in the original images was: 761/73/10% (**a**), 670/14/2% (**b**), 616/30/5% (**c**), 1033/87/8% (**d**), 267/27/10% (**e**), 465/35/8% (**f**), 509/11/2% (**g**), 1093/14/1% (**h**), 600/26/4% (**i**), 1444/12/1% (**j**), 1071/128/12% (**k**), 139/23/15% (**l**), 351/4/1% (**m**).

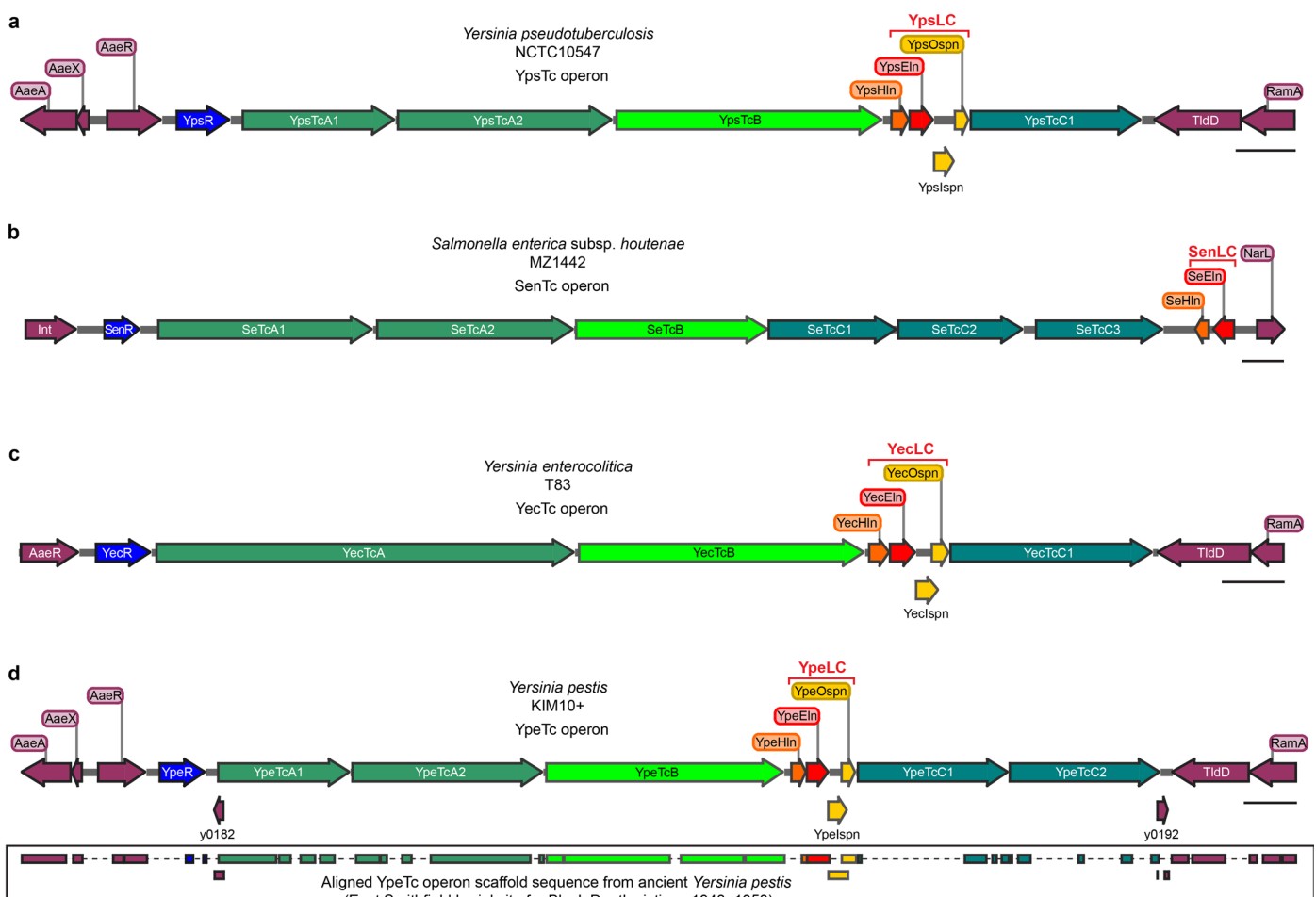

**Extended Data Fig. 4 | Several major human pathogens have type 10 secretion systems associated with Tc toxin genomic regions. a-d**, The Tc toxin genomic regions from *Yersinia pseudotuberculosis* (strain NCTC10547), *Salmonella enterica* subsp. *houtenae* (strain MZ1442), *Yersinia enterocolitica* (strain T83) and *Yersinia pestis* (strain KIM10 + ), with the embedded T10SSs designated as YpsLC, SenLC, YecLC and YpeLC in analogy to YenLC. Of these, all three *Yersinia* species are major human pathogens, while *Salmonella enterica* subsp. *houtenae* is an opportunistic pathogen in humans. The holin component is shown in orange, endolysin in red, spanins in yellow, putative transcriptional regulator of the Tc toxin operon in blue, TcA components in dark green, TcB in light green, TcC in blue-green, and genes not related to Tc toxins in burgundy. Inset of (**d**): the aligned YpeTc operon scaffold sequence from a London victim of the Black Death pandemic, ca. 1348-1350[60]. Regions shown with a dashed line had no available contigs due to degradation of ancient DNA. Scale bars: 1000 bp.

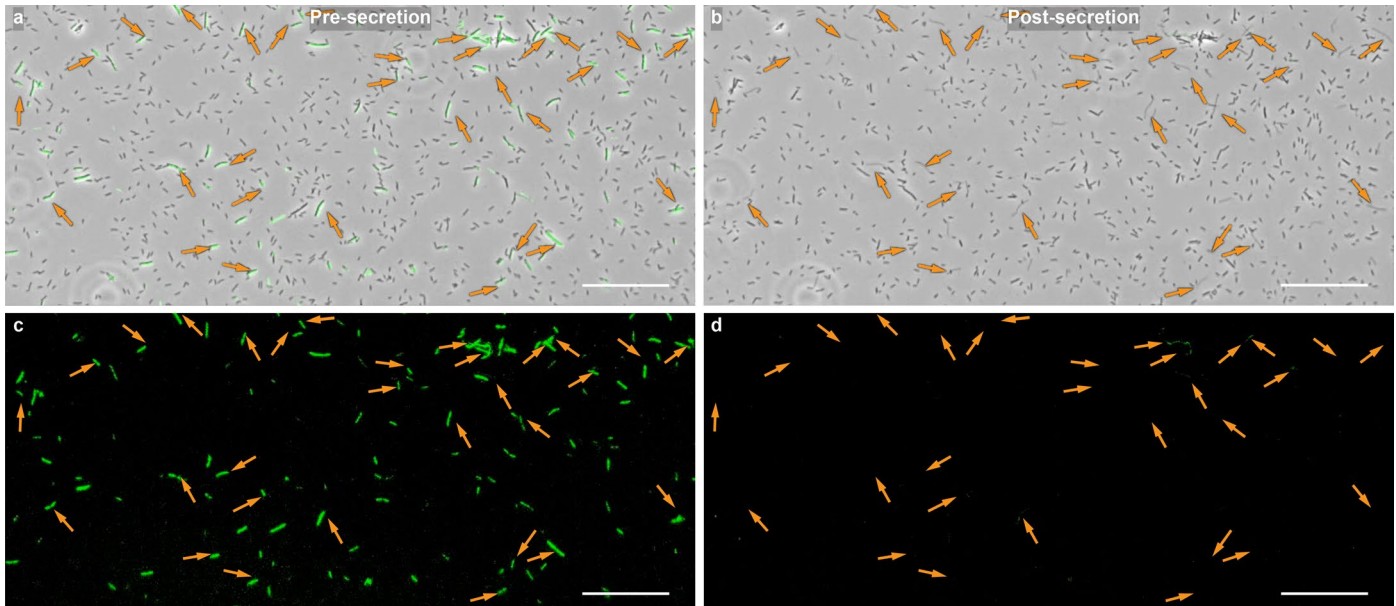

**Extended Data Fig. 5 | Only RoeA expressing soldier cells are capable of YenLC-mediated lysis. a-b**, RoeA-GFP expression serves as a marker for soldier cells, which are capable of YenLC-mediated, pH-dependent lysis. Here, the same area containing RoeA-sfGFP strain cells was imaged before (**a**) and 30 minutes after (**b**) secretion induced by raising the pH to 8.0. n = 3 biological replicates. **c-d**, The GFP channel shows the nearly complete post-incubation disappearance of RoeA-sfGFP signal from the cells' interior that was discernible using our imaging setup. n = 3 biological replicates. GFP⁺ cells: 477 (14%) of 3371 total cells analyzed in (**a, c**) and 12 (0.04%) of 3346 total cells analyzed in (**b, d**). Orange arrowheads denote the 30 RoeA-sfGFP expressing cells that could be definitively determined as having the same identity in both images. After incubation, they exhibit the characteristic YenLC-mediated lytic phenotype in the phase contrast channel. Scale bars: 50 μm.

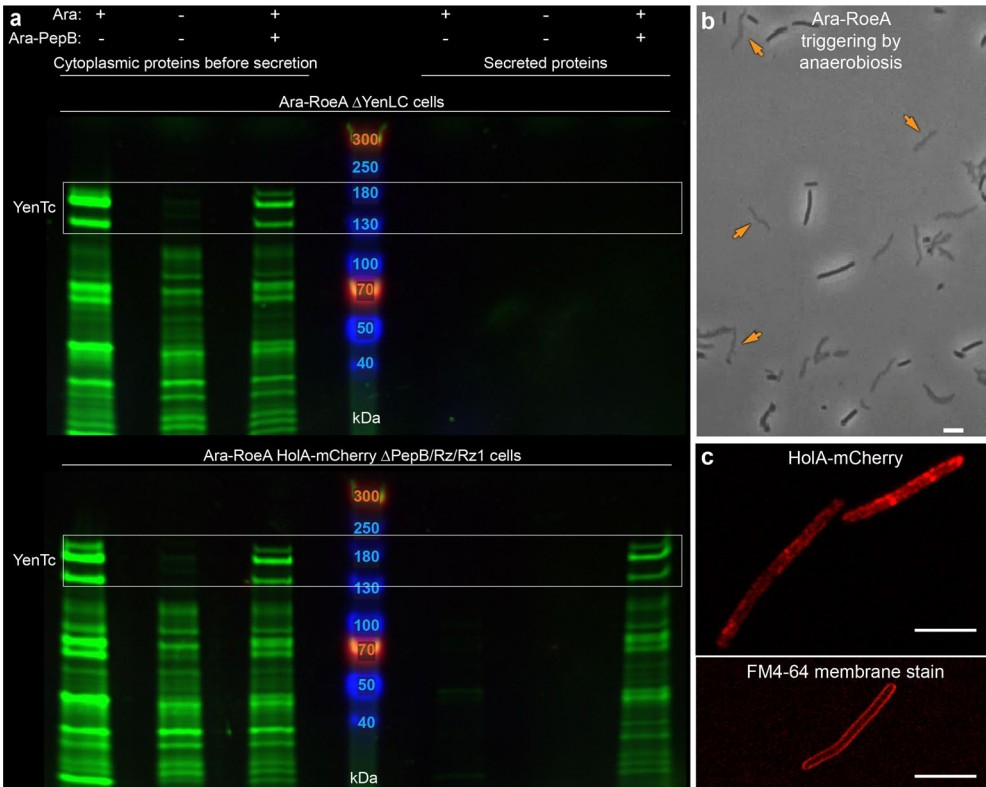

**Extended Data Fig. 6 | pH change triggers HolA, which concentrates into numerous small foci.** a, HolA-mCherry enables protein release in response to elevation of pH in Ara-RoeA HolA-mCherry ΔPepB/Rz/Rz1 cells. In this particular strain, Ara-PepB was produced from a plasmid in lieu of its (deleted) genomic copy to avoid interference of the fused mCherry tag to PepB expression, whereas in the remaining experiments of this study PepB is expressed via its native gene. For similar reasons, Rz and Rz1 were also deleted and given the toxicity we observed for plasmid-borne spanin overexpression, shearing forces applied to the bacteria during sample preparation were used as a replacement for their action, as has been previously documented[95]. Bands corresponding to YenA1, YenA2 and YenB are boxed for clarity. n = 3 biological replicates. b, Aerobic shaking cultures of Ara-RoeA cells can be stimulated to undergo YenLC-mediated secretion by immobility-induced anaerobiosis. Sudden anaerobiotic stress causes a loss in PMF due to decrease of cytoplasmic pH[98] and can lead to similar effects as increasing external pH with respect to PMF reduction between the cytoplasm and periplasm[50]. Orange arrowheads denote examples of cells that already underwent YenLC-mediated secretion. Lysed cells: 44 (66%) of 67 total cells analyzed. See Fig. 2b for an intact reference. Scale bar: 5 μm. n = 3 biological replicates. c, Confocal fluorescence microscopy of Ara-RoeA HolA-mCherry ΔPepB/Rz/Rz1 cells shows that small HolA foci are abundantly distributed across the bacterial membrane which contrasts to the few large lesions observed for triggered phage holin S105[49], and the smooth membrane signal produced by FM4-64 staining of Ara-RoeA ΔYenLC cells. The HolA-mCherry fusion presented here was shown to be fully functional by the experiments depicted in (a). Scale bars: 5 μm. n = 22 biological replicates.

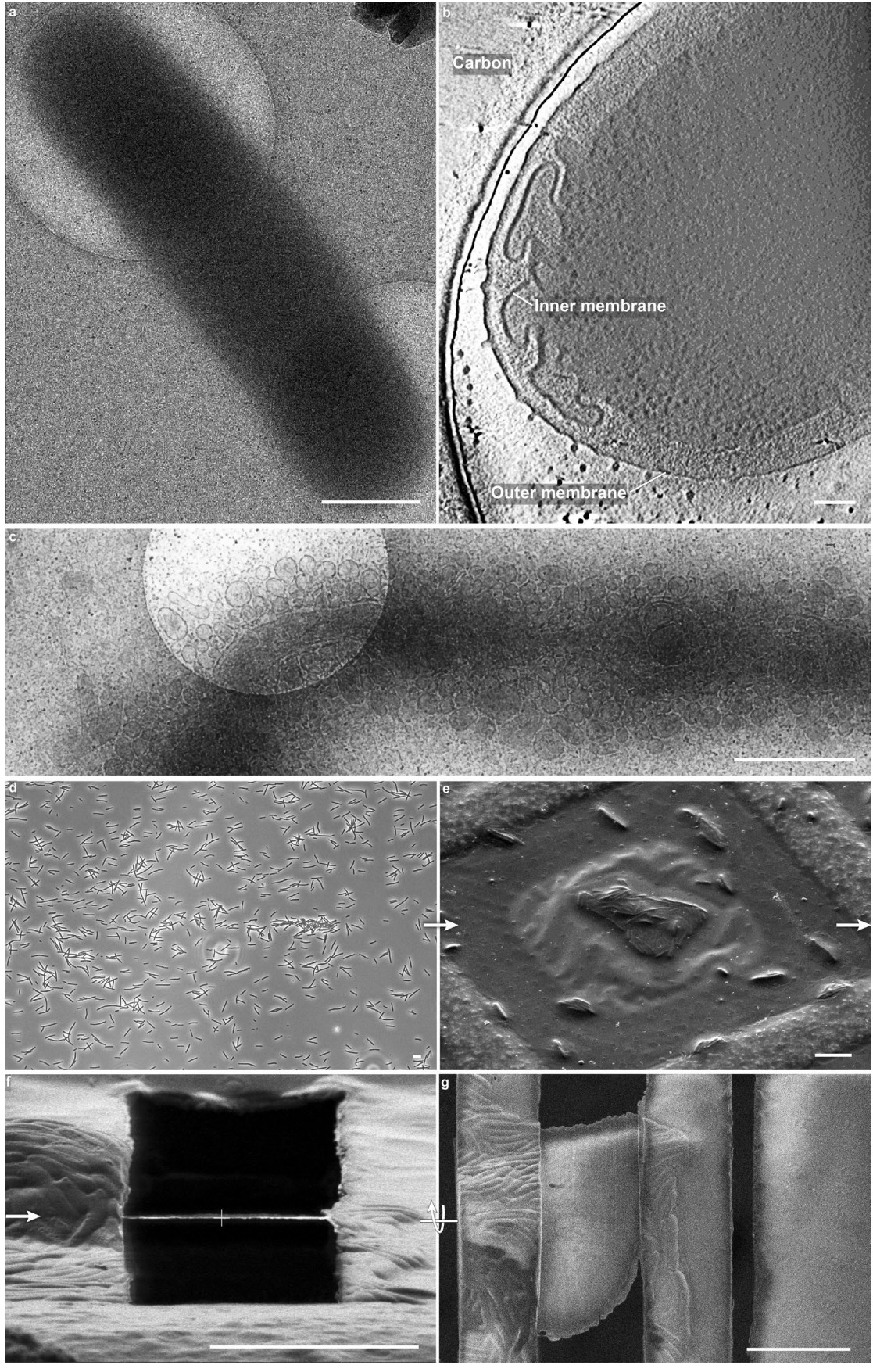

**Extended Data Fig. 7 | See next page for caption.**

**Extended Data Fig. 7 | The importance and process of bacterial sample preparation by cryo-focused ion beam milling. a**, A TEM overview image of an intact, unmilled, pH-induced Ara-RoeA ΔPepB/Rz/Rz1 cell, showing poor contrast due to the large size of *Y. entomophaga* (soldier) cells. Scale bar: 1 μm. n = 36 biological replicates. **b**, A tomogram obtained from an intact pH-induced Ara-RoeA ΔPepB/Rz/Rz1 cell showing numerous minor inner membrane invaginations likely attributable to HolA action. Notice how even with use of a Volta phase plate, only limited features are observable compared to the tomograms acquired on FIB-milled samples shown in other figures. Scale bar: 100 nm. n = 5 biological replicates. **c**, A TEM overview image of an Ara-RoeA cell after YenLC-mediated secretion, showcasing how the electron transparency of cells in this state enables high contrast imaging even without ion beam milling. Scale bar: 500 nm. n = 105 biological replicates. **d**, Induced Ara-RoeA ΔYenLC cells used to investigate the pre-secretion state of soldier cells, prior to vitrification. Scale bar: 10 μm. n = 3 biological replicates. **e**, SEM micrograph of plunge-frozen, pH-triggered Ara-RoeA ΔYenLC cells on an EM grid, prior to lamella generation. Scale bar: 10 μm. n = 3 biological replicates, 4 technical replicates each. **f-g**, SEM micrograph of an Ara-RoeA ΔYenLC cell lamella milled with a focused ion beam, side and top view. Scale bars: 10 μm. n = 3 biological replicates, 4-9 lamella sites each.

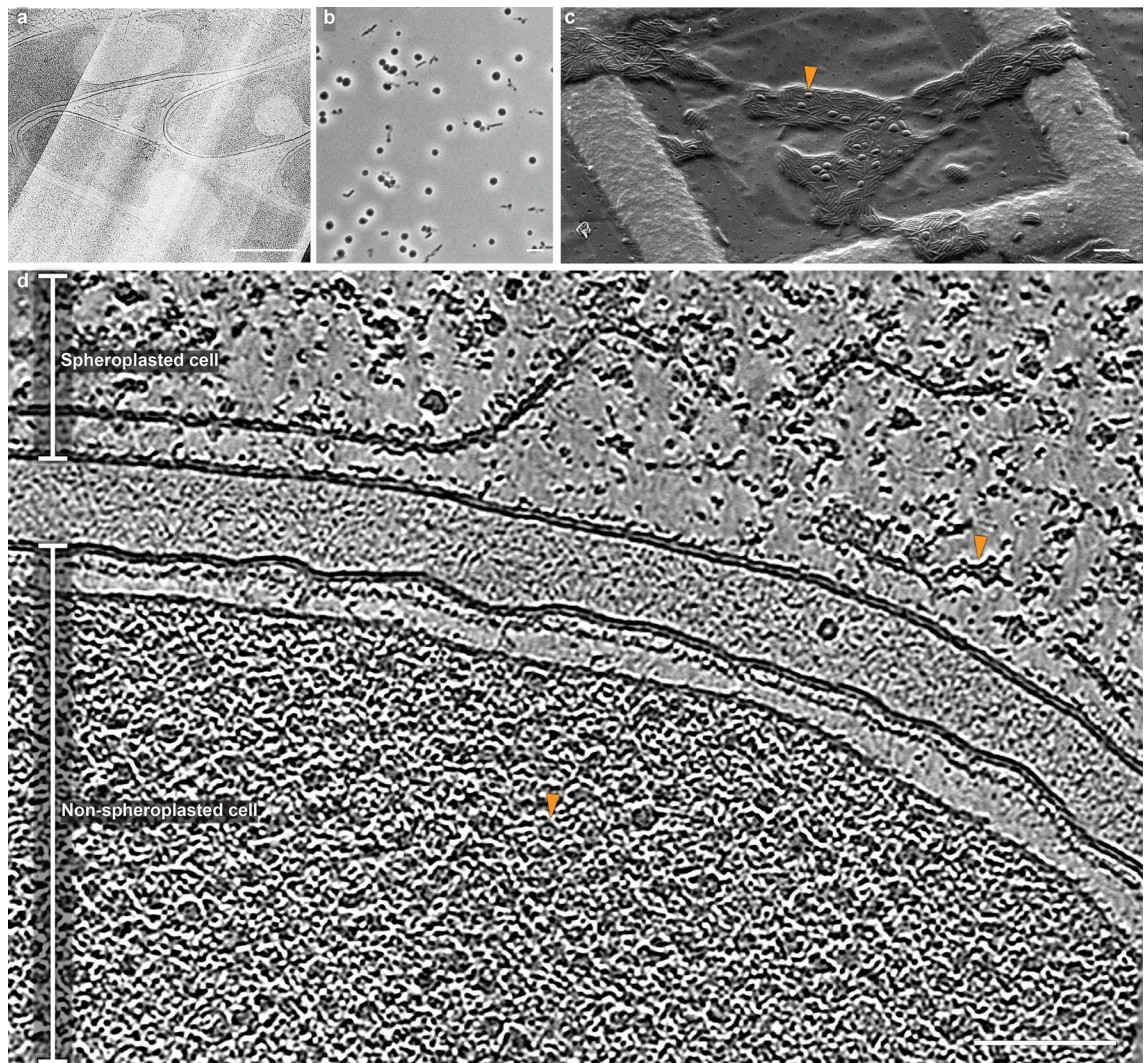

**Extended Data Fig. 8 | Spanin knockout soldier cells form spheroplasts with attractive qualities for *in situ* tomographic protein analysis. a**, During a ~<30 min window after pH-induced activation, enzymatic activity of HolA-translocated PepB causes Ara-RoeA ΔRz/Rz1 strain cells to exhibit severe inward bending of their inner membranes, as seen in this overview image of a cryo-FIB milled lamella. Scale bar: 1 μm. n = 86 biological replicates. **b**, Eventually such cells transform into spheroplasts. Scale bar: 10 μm. n = 3 biological replicates. **c**, SEM micrograph of plunge-frozen Ara-RoeA ΔRz/Rz1 cells that were pH-triggered on an EM grid, prior to lamella generation. Arrowhead: an example spheroplast. Scale bar: 10 μm. n = 3 biological replicates, 4 technical replicates each. **d**, A single slice from an Ara-RoeA ΔRz/Rz1 cell tomogram illustrating the difference in interior protein density of a spheroplasted cell compared to a non-spheroplasted cell. Notice the ultrastructural changes to the spheroplast inner membrane that occur as a result of cell expansion. Arrowheads: an example YenTc holotoxin from a spheroplast (side view) and non-spheroplast (top view). Scale bar: 100 nm. n = 86 biological replicates.

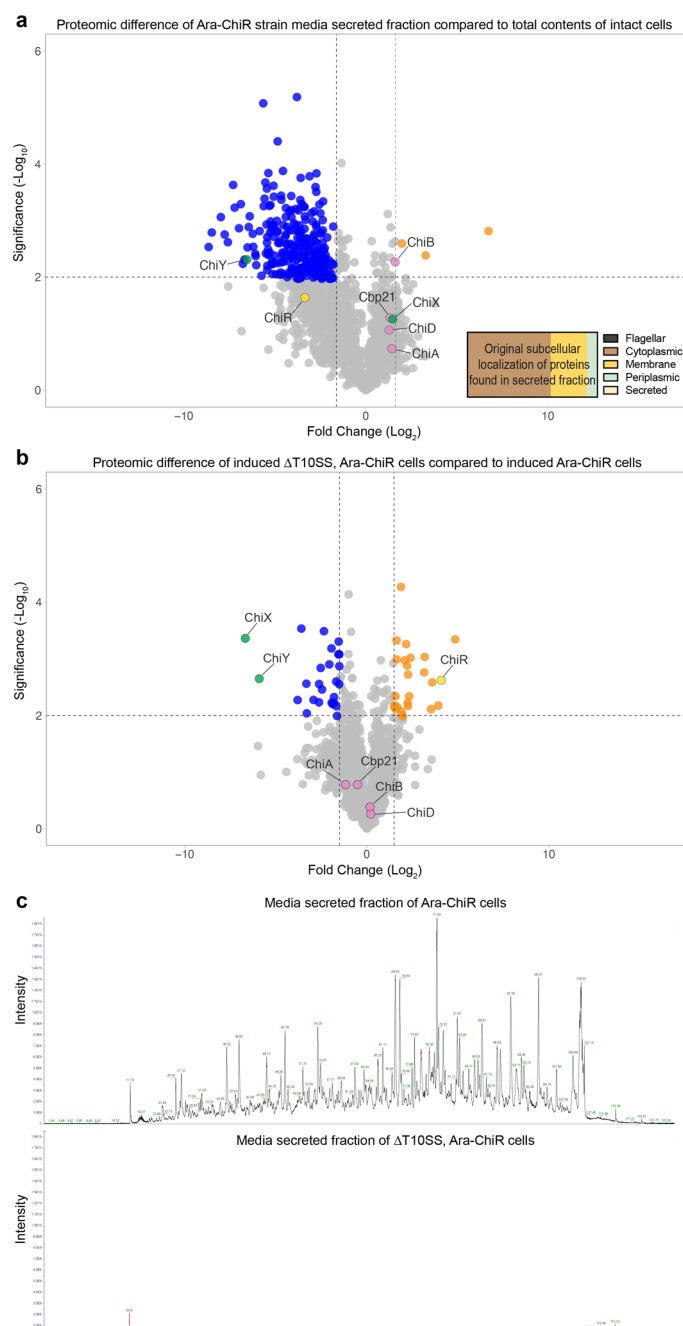

**Extended Data Fig. 9 | The *Serratia marcescens* T10SS enables release of most intracellular proteins into the surrounding environment. a-b**, Volcano plot of induced Ara-ChiR cells showing the (**a**) overall similarity of the protein fraction already secreted into growth media compared to the total protein fraction of still intact cells (inset: original subcellular localizations of those proteins found in the secreted fraction for which such information is available (443 of 1817 total hits), and (**b**) overall similarity of total cell proteomes from Ara-RoeA cells compared to SmaLC T10SS knockout cells expressing an arabinose-inducible version of ChiR from a plasmid. Comparison of the secreted fraction to the pre-secretion fraction shows that they are highly similar (grey circles correspond to most hits), as are the cytoplasmic fractions of the Ara-ChiR cells compared to the SmaLC knockout cells expressing a plasmid-encoded ChiR, which is produced in the latter at higher levels than the genomically encoded version in the former. A t-test p-value < 0.01 was used, and colors match Fig. 5b. n = 3 biological replicates each. The full proteomic datasets used to generate these figures are available in the Source Data. **c**, Total ion chromatograms from the media secreted fractions of Ara-ChiR cells and SmaLC T10SS knockout cells expressing an arabinose-inducible version of ChiR from a plasmid, showing the abundance of proteins released into the media via action of SmaLC and their complete absence therefrom upon deletion of this T10SS.

# Reporting Summary

## Statistics

For all statistical analyses, confirm that the following items are present in the figure legend, table legend, main text, or Methods section.

| n/a | Confirmed | |
|---|---|---|
| ☐ | ☒ | The exact sample size (*n*) for each experimental group/condition, given as a discrete number and unit of measurement |
| ☐ | ☒ | A statement on whether measurements were taken from distinct samples or whether the same sample was measured repeatedly |
| ☐ | ☒ | The statistical test(s) used AND whether they are one- or two-sided *Only common tests should be described solely by name; describe more complex techniques in the Methods section.* |
| ☐ | ☐ | A description of all covariates tested |
| ☐ | ☐ | A description of any assumptions or corrections, such as tests of normality and adjustment for multiple comparisons |
| ☐ | ☒ | A full description of the statistical parameters including central tendency (e.g. means) or other basic estimates (e.g. regression coefficient) AND variation (e.g. standard deviation) or associated estimates of uncertainty (e.g. confidence intervals) |
| ☐ | ☒ | For null hypothesis testing, the test statistic (e.g. *F*, *t*, *r*) with confidence intervals, effect sizes, degrees of freedom and *P* value noted *Give P values as exact values whenever suitable.* |
| ☐ | ☐ | For Bayesian analysis, information on the choice of priors and Markov chain Monte Carlo settings |
| ☐ | ☐ | For hierarchical and complex designs, identification of the appropriate level for tests and full reporting of outcomes |
| ☐ | ☐ | Estimates of effect sizes (e.g. Cohen's *d*, Pearson's *r*), indicating how they were calculated |

*Our web collection on statistics for biologists contains articles on many of the points above.*

## Software and code

Policy information about availability of computer code

| Data collection | Cryo-ET: SerialEM 3.8.5, Biochemistry: Bio-Rad ImageLab 5.2.1, MS: Xcalibur 4.0.27.10 |
|---|---|
| Data analysis | Cryo-ET: Warp 1.0.9, IMOD 4.10.28, RELION 3.0, EMAN2 2.91, cryoCARE, Dragonfly 2022.1, ChimeraX 1.4, TomoTwin 0.3. Biochemistry: GraphPad Prism 9. MS: MaxQuant 2.0.3.2, Perseus 1.6.14.0, VolcaNoseR. Bioinformatics: SignalP 6.0, WebLogo 3.7.12, Clustal Omega, Ray Meta, CSAR-Web. |

For manuscripts utilizing custom algorithms or software that are central to the research but not yet described in published literature, software must be made available to editors and reviewers. We strongly encourage code deposition in a community repository (e.g. GitHub). See the Nature Portfolio guidelines for submitting code & software for further information.

## Data

Policy information about availability of data

All manuscripts must include a data availability statement. This statement should provide the following information, where applicable:
- Accession codes, unique identifiers, or web links for publicly available datasets
- A description of any restrictions on data availability
- For clinical datasets or third party data, please ensure that the statement adheres to our policy

Source data (includes unprocessed SDS-PAGE gels, mass spectrometry proteomics data, and raw data for graphs) are provided with this paper in the Source Data and Supplementary Source Data files. The raw mass spectrometry proteomics data have been deposited to the ProteomeXchange Consortium (https://

# Research involving human participants, their data, or biological material

Policy information about studies with [human participants or human data](). See also policy information about [sex, gender (identity/presentation), and sexual orientation]() and [race, ethnicity and racism]().

| | |
|---|---|
| Reporting on sex and gender | N/A |
| Reporting on race, ethnicity, or other socially relevant groupings | N/A |
| Population characteristics | N/A |
| Recruitment | N/A |
| Ethics oversight | N/A |

Note that full information on the approval of the study protocol must also be provided in the manuscript.

# Field-specific reporting

Please select the one below that is the best fit for your research. If you are not sure, read the appropriate sections before making your selection.

☒ Life sciences          ☐ Behavioural & social sciences          ☐ Ecological, evolutionary & environmental sciences

For a reference copy of the document with all sections, see [nature.com/documents/nr-reporting-summary-flat.pdf]()

# Life sciences study design

All studies must disclose on these points even when the disclosure is negative.

| | |
|---|---|
| Sample size | No statistical methods were used to pre-determine sample sizes but our sample sizes are similar to those reported in previous publications. |
| Data exclusions | No data exclusions were made. |
| Replication | All experiments were carried out at least in biological triplicates. |
| Randomization | For the 3D refinement of cryo-EM/ET structures, particles were randomly split into two half sets. For all other experiments, randomization was not required because all data were used in the analysis. |
| Blinding | This study does not involve any experiments where blinding would be applicable. |

# Reporting for specific materials, systems and methods

We require information from authors about some types of materials, experimental systems and methods used in many studies. Here, indicate whether each material, system or method listed is relevant to your study. If you are not sure if a list item applies to your research, read the appropriate section before selecting a response.

## Materials & experimental systems

| n/a | Involved in the study |
|---|---|
| ☒ ☐ | Antibodies |
| ☒ ☐ | Eukaryotic cell lines |
| ☒ ☐ | Palaeontology and archaeology |
| ☒ ☐ | Animals and other organisms |
| ☒ ☐ | Clinical data |
| ☒ ☐ | Dual use research of concern |
| ☒ ☐ | Plants |

## Methods

| n/a | Involved in the study |
|---|---|
| ☒ ☐ | ChIP-seq |
| ☒ ☐ | Flow cytometry |
| ☒ ☐ | MRI-based neuroimaging |

## Plants

| Seed stocks | N/A |
|---|---|
| Novel plant genotypes | N/A |
| Authentication | N/A |

