## [Peer Review File · Nature Microbiology]

Peer Review Information

Journal: Nature Microbiology

Manuscript Title: Yersinia entomophaga Tc toxin is released by T10SS-dependent lysis of specialized cell subpopulations

Corresponding author name(s): Stefan Raunser

Reviewer Comments & Decisions:

Decision Letter, initial version:

Message: 26th June 2023

Dear Stefan,

First of all, thank you for very much for your patience while your manuscript "Specialized pathogenic cells release Tc toxins using a type 10 secretion system" was under peer-review at Nature Microbiology. It has now been seen by 3 referees, whose expertise and comments you will find at the of this email. You will see from their comments below that while they find your work of interest, some important points are raised. We are very interested in the possibility of publishing your study in Nature Microbiology, but would like to consider your response to these concerns in the form of a revised manuscript before we make a final decision on publication.

In particular, you will see that several of the referees asked for further validation and quantification of imaging data to support the conclusion that a subset of bacteria undergo T10SS-dependent cell lysis. There were also requests for further information on how secreted fractions were analysed to also support this section of the manuscript. Additionally, there were questions over the physiological relevance of conclusions drawn from the use of YenR overexpression or shortened YenTc toxin secretion analysis, and importantly, questions over the in vivo relevance of these findings and to what extent they affect pathogenesis in human pathogens. Although the T10SS and Tc toxin has already been shown to contribute to *Yersinia enterocolitica* virulence in *Galleria* models, we would strongly encourage you to explore whether the division-of-labour strategy in T10SS activity is also occurring in vivo. Moreover, if no evidence can be provided to support the conclusion that this T10SS-dependent cell lysis mechanism of Tc toxin release occurs in any candidate human pathogens, this aspect of the manuscript should also be toned down. Several of the referees suggested careful reediting of the manuscript to increase accessibility of the text, and also to qualify certain language so that it is in keeping with current concepts held within the field. We feel that these are critical points which would need to be addressed for us to further consider a revised manuscript, alongside the remaining issues outlined in the referees' reports, which are clear and should be straightforward to address.

Should further experimental data allow you to address these criticisms, we would be happy to look at a revised manuscript.

2We are committed to providing a fair and constructive peer-review process. Please do not hesitate to contact us if there are specific requests from the reviewers that you believe are technically impossible or unlikely to yield a meaningful outcome.

Please include a data availability statement as a separate section after Methods but before references, under the heading "Data Availability". This section should inform readers about the availability of the data used to support the conclusions of your study. This information includes accession codes to public repositories (data banks for protein, DNA or RNA sequences, microarray, proteomics data etc...), references to source data published alongside the paper, unique identifiers such as URLs to data repository entries, or data set DOIs, and any other statement about data availability. At a minimum, you should include the following statement: "The data that support the findings of this study are available from the corresponding author upon request", mentioning any restrictions on availability. If DOIs are provided, we also strongly encourage including these in the Reference list (authors, title, publisher (repository name), identifier, year). For more guidance on how to write this section please see: <http://www.nature.com/authors/policies/data/data-availability-statements-data-citations.pdf>

* If you have not done so already we suggest that you begin to revise your manuscript so that it conforms to our Article format instructions at <http://www.nature.com/nmicrobiol/info/final-submission>. Refer also to any guidelines provided in this letter.

When submitting the revised version of your manuscript, please pay close attention to our [href="https://www.nature.com/nature-portfolio/editorial-policies/image-integrity">Digital](https://www.nature.com/nature-portfolio/editorial-policies/image-integrity)

Image Integrity Guidelines. and to the following points below:

Note: This url links to your confidential homepage and associated information about manuscripts you may have submitted or be reviewing for us. If you wish to forward this e-mail to co-authors, please delete this link to your homepage first.

Nature Microbiology is committed to improving transparency in authorship. As part of our efforts in this direction, we are now requesting that all authors identified as 'corresponding author' on published papers create and link their Open Researcher and Contributor Identifier (ORCID) with their account on the Manuscript Tracking System (MTS), prior to acceptance. This applies to primary research papers only. ORCID helps the scientific community achieve unambiguous attribution of all scholarly contributions. You can create and link your ORCID from the home page of the MTS by clicking on 'Modify my Springer Nature account'. For more information please visit www.springernature.com/orcid.

If you wish to submit a suitably revised manuscript we would hope to receive it within 6 months. If you cannot send it within this time, please let us know. We will be happy to consider your revision, even if a similar study has been accepted for publication at Nature Microbiology or published elsewhere (up to a maximum of 6 months).

Yours sincerely,

[redacted]

Reviewer Expertise:

Referee #1: Yersinia, Tc toxins, type 10 secretion systems

Referee #2: Tc toxins, insect pathogens

Referee #3: Structural biology, cryo ET, pore-forming toxins, bacterial cell biology

3Reviewer Comments:

Reviewer #1 (Remarks to the Author):

Oleg Sitsel and colleagues sent the manuscript entitled "Specialized pathogenic cells release Tc toxins using a type 10 secretion system" to Nature Microbiology for publication (no. NMICROBIOL-23030741). In this study, the authors state that a subset of *Yersinia entomophaga* cells produces and releases the insecticidal toxin complex Tc into the environment. The study investigates the pH-dependent secretion of the Tc, the a role of a T10SS for Tc release, a regulator that synchronizes Tc production and release in a temperature-dependent and bistable manner, and individual steps of T10SS activity. Finally, the authors include the *Serratia marcescens* T10SS to transfer their findings.

• Conclusion: The study provides very interesting data on the mechanism of Tc release in combination with heterogeneity and a phage-like lysis system. However, there are several major drawbacks. First, there is a lack of novelty regarding several parts of the study. Second, the study does not provide data to support the *in vivo* relevance of the mechanism described. Third, the generalisation that Toxin release via specialised cells takes place in human pathogen is not supported by the experiments performed. Fourth, the experiments do not convincingly demonstrate toxin release by cell lysis. Fifth, the results must be complemented by sufficient quantitative data and by the application of native conditions to verify the conclusions drawn by the authors. Finally, the study does not follow the nomenclature of established concepts and terms, and its background is not state-of-the-art with respect to microbial pathogenesis. This reviewer recommends to address the different aspects of the manuscript by in-depth-analyses for separate publication.

Major points:

1. Data regarding the co-release of proteins like chitinases, the temperature-dependent regulation of the major factors involved in Tc release, the pivotal role of a novel regulatory protein, and the heterogeneity of a cell population have recently been published by the Hurst lab (Schoof et al. (2023; Lysis cassette-mediated exoprotein release in *Yersinia entomophaga* is controlled by a PhoB-like regulator; Microbiology Spectrum DOI: 10.1128/spectrum.00364-23). The authors might have overseen that, and, in particular, the hypothesis on specialised cells was not elaborated by Schoof and coworkers. However, this publication is state-of-the-art with respect to the results listed above.
2. The title is too general and does not match well to the content. I propose a more specific one: "Self-destructive cells of *Yersinia entomophaga* release..."
3. All results were obtained *in vitro*. However, it cannot be excluded that there is a distinct mechanism during infection of hosts, e.g. invertebrates. Therefore, conclusions on a new mechanism that play a role in pathogenesis are not supported by the data.
4. The authors state the release mechanism via cell lysis is valid for all T10SS identified and described so far. This conclusion, however, cannot be drawn from the data that are restricted to *Y. entomophaga* and *Serratia marcescens*. With respect to a number of pathogens, alternative mechanisms have been proposed in several studies cited by the authors and are still not falsified.
5. Line 35-38: This statement is not corrected and not supported by the literature cited. The Tc are not known "as crucial virulence factors in numerous highly prominent...human pathogens". Song et al. performed an *in silico* search for the abundance of TcC, which ADP-ribosylates eukaryotic target molecules, in the genomes of >133,000 bacterial genomes.

The study identified N-termini of TcCs encoding RhsA and RHS core domains to be widespread among bacteria, but those N-termini are far away from a canonical toxin complex. Moreover, this study does not provide any data on the toxicity of newly identified domains.

6. The microbiological background in the abstract and in the introduction is not state-of-the-art and confusing. Abstract, lines 26–29: there is no toxin complex operon in *Salmonella enterica*, this is a misleading statement. In *Salmonella enterica*, TccC-like domains are conserved, but these are far from a toxin complex-encoding operon. The term Tc is restricted to insecticidal toxins with homology to the Tc of *Photorhabdus luminescens*. The typhoid toxin is produced by *Salmonella enterica* serovar Typhi, but a holin is still missing. The endolysin probably involved in typhoidal toxin release is not a T10SS that by definition requires the tandem activity of a holin and an endolysin (Palmer et al. 2021, DOI: 10.1111/mmi.14599).

7. Same lines: *Yersinia pestis* is not an appropriate example for a pathogen that profits from insecticidal activity conferred by Tc. Although *Y. pestis* in vitro produces Tc, a role in the life cycle has not been demonstrated. Rather, it is assumed that the Tc of *Y. pestis* is not expressed (pseudogenes) or produced under in vivo conditions, e.g. in the flea, because it would otherwise harm its insect host. Therefore, it was suggested that a loss of insecticidal Tc activity was required for the evolution of this human pathogen (see papers by the Hinnebusch lab).

8. According to the literature, the Tc confers insecticidal activity. This is to be mentioned in the abstract.

9. Summarizing the major points made above (title, abstract), the reader gets the impression that interesting data on self-destruction-dependent release of insecticidal toxin are overstated to imbed the insecticidal toxin and its release mechanism into the broader context of bacterial pathogenicity towards humans. However, the statement of lines 26–29 is not at all supported by the experimental findings reported in this manuscript.

10. Co-secretion of proteins in Fig. 2: it remains unclear why the secreted fraction is highly enriched with other (virulence) factors in comparison with those remaining in the cell. Essential details on the experimental settings are missing here. Which growth conditions were used to select for lysing cells? The control is MS with the “remaining fraction”, please clarify how this fraction was isolated. What is the difference to the cytoplasmic content? It is misleading to write remaining fraction when lysis occurs. Is there any selection mechanism of the exoproteins? Literature addressing this aspect by alternative concepts are missing here (Toyofuku et al. 2017, Nat. Comm. 8:481; Turnbull et al. 2016 Nat. Comm. 7:11220; Vacheron et al. 2021 Commun Biol 4:87).

11. Overexpression of YenR used in Fig. 2c and 2d to study exoproteins is an artificial condition that might be different to the native condition, in particular with respect to cell lysis or related mechanisms. To summarize, the data provided in Fig. 1–2 cast the lysis hypothesis into severe doubt.

12. Fig. 2e: This finding requires thorough quantitative analysis rather than providing a single image.

13. Use of the fusion YenA1-sfGFP: in this construct, GFP is fused after residue 37 of the YenA1 subunit. Thus, the authors applied a translational fusion of ~275 residues and 30 kD to monitor the release of a high molecular weight toxin subunit. However, there is no YenTc-sfGFP construct mentioned in the supplementary material, whereas YenA1-sfGFP is not found in the main text. Please clarify. More important, such a construct with 37 residues of YenA only does not match the natural conditions, and a full-length fusion or

anti-YenA1-antibodies must be used for monitoring the release mechanism.

14. The introduction of new term such as "soldier cells" and "death factor" (DF) is confusing. Soldier cells is a new term that the authors define as a small subset of specialized cells. What is the reason for this wording? Many cellular activities require specialization. Soldiers do a lot, but rarely undergo suicide. The term "soldier" might be eye-catching, but its use in the context of T10SS is not appropriate and misleading. Please replace with "self-destructive cells", which refers to a well-known concept in bacterial pathogenicity (see publications by the Hardt and Ackermann labs, e. g. Nature 454:987 (2008)).

Same for "death factor": there are numerous publications now that describe or hypothesize the role of lysis cassettes in toxin release, prompting Palmer and coworkers to establish new a type 10 secretion system. Even if it might turn out in the future that toxin release is mediated by cell lysis rather than by controlled secretion, the T10SS concept is very helpful to improve our understanding of toxin release via holin and endolysin. The term "Explosion-competent" should be introduced by citing respective literature, otherwise use "lysis-competent" instead.

15. The nomenclature of the regulatory factor identified in this study, "YenR", is another example for an insufficient microbiological background in the study. YenR is a regulator of *Yersinia enterocolitica* involved in quorum sensing systems (Atkinson et al., Mol Micro 1999), and its sequence is distinct from "YenR" of *Y. entomophaga*. Please replace with "RoeA" (Schoof et al.).

General remarks to be addressed:

16. Introduction: please provide more information on YenTc. Compare subunit nomenclature used in this study with the nomenclature of *P. luminescens* Tc subunits. What are the (putative) functions of the YenTc subunits? Why targeting YenA1? Tc stoichiometry?

17. Supplementary Fig. 1a: The legend and the methods should comprise more details on protein separation including the imaging system. Here and in the main manuscript, data on the molecular weight of YenTc components are missing. Please comment on the bands with higher and lower MW than YenTc. How were YenA1, YenA2, and YenB identified here and elsewhere, just to their lack in the Tc deletion mutant? Other proteins are also missing upon Tc deletion. Antibody-based detection is missing here.

18. Fig. 1 The map of YenTc should be improved; cf. to those provided by Hurst and coworkers in numerous papers.

19. Fig.S4: Deletion of a lipase significantly reduces the amount of YenA1 released, suggesting its role in Tc release similar to a lipase of *P. luminescens*.

20. Fig. 2b: no soldier cells appear in the absence of induction? This contrasts data shown in Fig. 1b, please clarify.

21. Lines 126: Did the authors exclude the presence of secretion system duplications in the genome of *Y. entomophaga*?

22. Fig. S5n, the experiment requires more details to ease reading. Which conditions were applied? How was secreted YenTc-GFP identified and measured?

23. Fig. S6: "raft-like oligomers" are not clearly visible.

Minor points:

1. A consistent reference style should be used

2. Line 36: "non-phage-derived" anticipates that factors involved in T10SS are phage-related, but this is unknown for many readers not familiar with the topic.

3. Line 82: "...that has instead evolved to target insects". This statement suggests that a *Yersinia* ancestor split into two lines evolving towards human or insect pathogenicity and thus misunderstands the yersinial evolution. In fact, *Y. pestis* only recently evolved from *Y. pseudotuberculosis*, and it is assumed that many *Yersinia* species were adapted to invertebrates before the evolution of vertebrates. This sentence should be rewritten.
4. Table S1: "to CmR" means respective gene was exchanged with CmR resistance?

Reviewer #2 (Remarks to the Author):

This is another amazing paper on the Tcs so congratulations on that.

All of my comments are really about 'framing' of the study and not really about the science as such.

1. Soldier cells is an attractive name. But do these cells die prematurely in the secretion process (do they bleb themselves to death)? If so maybe kamikaze cells is better?
2. Why does *Photothabdus* (PI) not use this mechanism? As I understand it the tc encoding genes are found on the pADAP plasmid of *Ye* and are not incorporated into the genome as they are for PI. So if the PI Tcs are able to secrete themselves then perhaps the Type10 secretion is an intermediate/first step specific to plasmid encoded Tcs?
3. l82 'has evolved to target insects'. This is a rather anthropomorphic view of things? Invertebrates form the vast majority of infectable animals on the planet and infection of vertebrates is therefore secondary- which may also relate to argument 2? So perhaps vertebrate pathogenicity is associated with a) plasmid borne TCs and b) type 10 secretion? Either way *Yersinia* started killing insects and perhaps accidentally has infected humans as *Y. pestis*.
4. In PI Tcs can be found either in the supernatant alone or cell associated. Previously we had assumed that this 'release' into the supernatant was associated with a lipase gene encoded nearby. Do the Type10 finding change this interpretation or offer a better one?
5. The links with holin/endolysin/spanin and bacteriophage are fascinating? Does this mean that the Tcs are themselves actually modified bacteriophage and what implications does this have for their evolution and/or secretion as in 4 above (are 'lipases' important in the release of bacteriophage too?).
6. Finally, l351 'cellular targets'. What are the likely targets of these Tc containing vesicles? Are they likely to be taken up by endocytosis? What are there 'receptors' or do they just fuse with target cells in a non-specific manner?

None of these arguments are mission critical but I hope they raise points for interpretation and/or future work.

Reviewer #3 (Remarks to the Author):

7Summary

Sitsel et al. characterize the previously unknown mode of secretion for Tc toxins, using *Y. entomophaga* YenTc as a model. They find that a large number of other virulence factors and cytoplasmic proteins are released into the environment along with YenTc via a controlled lysis mechanism in a subset of cells that they term "soldier cells", and the T10SS, named YenDF, is responsible for this secretion. Temperature, pH, and other factors may drive the mechanism. They use cryo-ET with YenDF component knockout cells to observe the state of the cell and morphology of the cell envelope at different stages of secretion. This allowed them to propose a model for how holins, endolysins and spanins each contribute to membrane changes and cytoplasmic content release. Their proposed mechanism is consistent with observations in *S. marcescens*, suggesting that it may be wide-spread. This study was interesting, with many orthogonal pieces of data, and the data along with the proposed model for T10SS mechanism is likely to be of broad interest to microbiologists. We thank the authors for posting their work on BioRxiv, which makes it available to the community, and accelerates new discoveries based on their work.

Major Comments

Fig 1A: We did not follow why the "secreted fraction" contains such few hits. If the mechanism is from controlled lysis, and the "secreted" fraction is then actually just released proteins as a result of cell lysis, why aren't all proteins enriched as compared to non-secreted fraction? Perhaps we misunderstood this experiment, and a clearer explanation of how this is carried out, what the "secreted" fraction is, and how this relates to soldier/non-soldier cells would help.

Fig. S4 suggests that all secreted factors are dependent on T10SS, yet other secretion systems are encoded. Is it known why the bacteria encodes all the other secretion systems if they are not used - are they expected to be used only under certain conditions? Are the smaller virulence factors unable to be secreted by any of the typical export pathways, only cell lysis? The lack of signal sequence for all the virulence factors is peculiar. Is this typical for *Yersinia* species? Some explanation of this to contextualize the use of T10SS in relation to others would be helpful.

For all light microscopy, could the authors please provide quantification, and clearly indicate how many cells are analyzed

For tomography data, please indicate how many cells were inspected in each case that leads to a conclusion, and some estimation of the variability observed. We completely understand that cryo-ET is a low-throughput technique at this time, but, given the general heterogeneity of cells when observed at these resolutions, it would be very helpful to gain an understanding of the variability, and the numbers used.

In some cases, we suggest that conclusions from cryo-ET be qualified. For example, the right inset in Figure 3d - may be an unfused vesicle, or may be something else. Since it is very difficult currently to use cryo-ET tags to unambiguously define what one is looking at (unless sufficiently high resolution is achieved using STA for protein complexes), qualifying the identification of any object in the tomograms (with less certainty) would be reasonable. The overall narrative, and the order of figures is somewhat scattered, which makes reading through the whole manuscript quite challenging. For example, Fig. S2, which contains soldier cells, is referenced at the beginning of the results, but the concept of soldier cells has not yet been introduced. It therefore took substantial effort to go back and forth through the text and figures, and keep all the results straight. We would strongly

encourage the authors to ensure that the flow and logic of the manuscript is linear, and that the figures are arranged accordingly.

Minor comments

Line 137-138/Fig S5: This figure does not seem to show that low temperature has the strongest influence on YenTC production, as stated in the text?

Please label which bands correspond to YenA1, YenA2 and YenB on boxed YenTc for all gels
Please explicitly explain how YenA1/A2 and YenB correspond to TcA and TcB

Is there speculation on why an anoxic environment inhibits YenTc secretion but not production? This is just a curiosity question

The figure legend for Fig S2h seems to be missing

Fig 4C show Chi 1 and Chi 2, which were never introduced

Fig 4C: Where is YenB?

Fig 5A: What are M and T fractions?

S2F,G,H - what is the status of GFP+ cells for 37C? It was not clear to us how the morphology correlated to being GFP+/-

Author Rebuttal to Initial comments

Point-to-point response to the reviewers' comments

We thank the reviewers for their positive feedback and insightful comments, which aided us to further improve the manuscript. Below is a point-by-point response to all comments and a detailed description of all changes we have made to our manuscript after considering their suggestions. The changes are highlighted in yellow in the revised text.

Reviewer #1

Oleg Sitsel and colleagues sent the manuscript entitled “Specialized pathogenic cells release Tc toxins using a type 10 secretion system” to Nature Microbiology for publication (no. NMICROBIOL-23030741). In this study, the authors state that a subset of *Yersinia entomophaga* cells produces and releases the insecticidal toxin complex Tc into the environment. The study investigates the pH-dependent secretion of the Tc, the a role of a T10SS for Tc release, a regulator that synchronizes Tc production and

9release in a temperature-dependent and bistable manner, and individual steps of T10SS activity. Finally, the authors include the *Serratia marcescens* T10SS to transfer their findings.

- **Conclusion:** The study provides very interesting data on the mechanism of Tc release in combination with heterogeneity and a phage-like lysis system. However, there are several major drawbacks. First, there is a lack of novelty regarding several parts of the study. Second, the study does not provide data to support the *in vivo* relevance of the mechanism described. Third, the generalisation that Toxin release via specialised cells takes place in human pathogen is not supported by the experiments performed. Fourth, the experiments do not convincingly demonstrate toxin release by cell lysis. Fifth, the results must be complemented by sufficient quantitative data and by the application of native conditions to verify the conclusions drawn by the authors. Finally, the study does not follow the nomenclature of established concepts and terms, and its background is not state-of-the-art with respect to microbial pathogenesis. This reviewer recommends to address the different aspects of the manuscript by in-depth-analyses for separate publication.

We thank the reviewer for providing a thorough analysis of our work. However, we disagree with the critical points in the summary above, which attempts to portray our work as non-novel and unconvincing. We have addressed each of these in our responses to the list of 27 comments by the reviewer (below). In addition, we strongly disagree with the reviewer's suggestion to "address the different aspects of the manuscript by in-depth-analyses for separate publication" – that is, "salami slice" our manuscript, which is broadly considered to be an unethical practice. Rather, we are convinced that here we have presented a multifaceted story that brings together microbiology, genetics, proteomics, live-cell imaging and structural biology to demonstrate an exciting aspect of bacterial pathogenesis, and intend to keep our study as integrated and complete as our available resources allow us to.

Major points:

1. Data regarding the co-release of proteins like chitinases, the temperature-dependent regulation of the major factors involved in Tc release, the pivotal role of a novel regulatory protein, and the heterogeneity of a cell population have recently been published by the Hurst lab (Schoof et al. (2023; Lysis cassette-mediated exoprotein release in *Yersinia entomophaga* is controlled by a PhoB-like regulator; *Microbiology Spectrum* DOI: 10.1128/spectrum.00364-23). The authors might have overseen that, and, in particular, the hypothesis on specialised cells was not elaborated by Schoof and coworkers. However, this publication is state-of the-art with respect to the results listed above.

10It was not possible for us to know of the study by Schoof et al. and hence overlook it, given that it appeared in the 13 April 2023 issue of *Microbiology Spectrum* – well after our manuscript was written, published on bioRxiv, submitted to this journal, and sent out for review. Having said that, we of course now happily acknowledge it by including a relevant sentence and citation in our manuscript in the section on synchronization of YenDF and YenTc production: the Schoof et al. study provides additional independent support to the story we present here, as well as interesting experimental findings we do not cover in our manuscript, namely the discovery of a regulatory ncRNA in the YenDF region and a transcriptomic analysis.

A careful comparative reading of both manuscripts does however reveal that while both deal with the same organism and overall topic, their overlap is largely limited to the observation that deletion of the YenDF operon results in cessation of protein release by *Y. entomophaga* and that mutation of its YenR regulator additionally prevents production of the Tc toxin YenTc. Not only was characterization of the specialized pathogenic subset of *Y. entomophaga* cells absent in Schoof et al., as the reviewer noted, but it also lacked many crucial components of our study. These include – but are not limited to:

- Full proteomic differential analyses of the proteins produced and released by soldier cells and their mutants (only four select non-YenTc bands were excised and identified in Schoof et al.)
- The observation and characterization of pH-dependence for YenDF-mediated secretion
- Characterizing the influence of various environmental conditions apart from temperature on soldier cell differentiation and secretion
- Observation and characterization of the stepwise YenDF-mediated protein release process by cryo electron tomography, as well as identification of the cytoplasm as the location of Tc holotoxin assembly (which precludes the secretion of Tc toxins by means other than lytic release).
- Investigation of the archetypal T10SS from *Serratia marcescens* using a combination of genomic editing, differential proteomic analyses, and cryo electron tomography.

Furthermore, we have our own set of concerns with several elements of the Schoof et al. study:

- The use of an overexpression plasmid containing an artificially gene structure-optimized version of YenDF to come to the conclusion of its lytic capability, while using the original version of YenDF in such a system resulted in both a lack of lysis and a corresponding inability to rescue protein release in the Δ YenDF mutant strain. In contrast, we use the original genomically encoded system to study YenDF function and show that it functions in a lytic manner in its native setting.
- Observations of heterogeneous holin reporter expression simultaneously with non-heterogeneous spanin reporter expression, although they are located in the same operon. This data is at odds with the bimodal, YenR-dependent expression of the entire YenDF operon that we present here.

- Absence of an explanation for the temperature sensitivity of the YenR regulator. We clearly show that this is due to the instability of the YenR mRNA at elevated temperatures and determine its role as the master regulator of pathogenic cell production in *Y. entomophaga*.
- Lack of a proper deletion mutant for the YenR regulator, with a partial deletion mutant having no effect on protein production and secretion. We have solved this issue by inserting a binary on/off promoter switch for YenR and use this to present very clear data on the proteomic, phenotypic and secretory role of YenR.
- Ignoring the mechanistic role of the two spanins of YenDF in protein release. We, on the other hand, provide cutting edge cryo-electron tomographic evidence at 42000x magnification of the crucial role and effect of spanins during Tc toxin release.
- Presenting the type 10 secretion systems as if they were already known to be lytic. This is simply not true – reading the original reference the Schoof et al. study cites (Hamilton et al., 2013) as well as the follow-up review paper on type 10 secretion systems (Palmer et al., 2021) will confirm that the authors of these expressly considered type 10 secretion systems to be non-lytic. We however show that they are in fact lytic in both *Y. entomophaga* (without resorting to an overexpression plasmid containing an artificially gene structure-optimized version thereof) and in the original organism that the archetypal type 10 secretion system was studied in – *Serratia marcescens*, a well-known human, animal and plant pathogen.

In conclusion, we are happy to acknowledge the concurrent Schoof et al. study in our manuscript, yet we strongly object to the reviewer’s suggestion that our work lacks sufficient novelty with respect to it.

2. The title is too general and does not match well to the content. I propose a more specific one: “Self-destructive cells of *Yersinia entomophaga* release...”

We thank the reviewer for this point and agree that mentioning *Yersinia entomophaga* in the title is a good idea. It has been modified accordingly.

3. All results were obtained in vitro. However, it cannot be excluded that there is a distinct mechanism during infection of hosts, e.g. invertebrates. Therefore, conclusions on a new mechanism that play a role in pathogenesis are not supported by the data.

The conclusions of this work have been contemporarily confirmed by *in vivo* studies of the Fuchs group, where *Yersinia enterocolitica* Tc toxin and/or T10SS mutants were shown to be incapable of establishing an infection compared to wild type bacteria. This is clearly stated and cited in the third-to-final paragraph of the discussion section. These studies, combined with our data, present overwhelming evidence that T10SSs are indeed necessary for Tc toxin release – also *in vivo*.

4. The authors state the release mechanism via cell lysis is valid for all T10SS identified and described so far. This conclusion, however, cannot be drawn from the data that are restricted to *Y. entomophaga* and *Serratia marcescens*. With respect to a number of pathogens, alternative mechanisms have been proposed in several studies cited by the authors and are still not falsified.

Only three T10SSs have been described so far experimentally. The first is the archetypal T10SS from *S. marcescens* that was originally proposed to function non-lytically by Palmer and colleagues, which we disprove here. The second is the T10SS from *Y. enterocolitica*, which shows lytic action upon heterologous expression as demonstrated by the Fuchs group, and the third is YenDF from *Y. entomophaga*, which is thoroughly studied here and demonstrated to be lytic (also when heterologously overexpressed by plasmid as shown in a concurrent study by the Hurst group, albeit only when a gene-optimized variant is used in their case). Lytic release has now been consequently shown for all three experimentally described T10SSs, which fits well with the general similarity of T10SSs to bacteriophage lysis cassettes. We therefore consider it quite reasonable that the burden of trying to provide experimental evidence for the non-lytic action of T10SSs now lies on the proponents of this concept.

As for alternative mechanisms of toxin export by pathogens employing phage-derived proteins other than T10SS clusters, we clearly state in the discussion that “the current consensus is that the holin and/or endolysin-mediated export occurs via a non-lytic mechanism” however we do suggest that “in light of the data presented here the potential use of suicidal soldier cell subpopulations also by these pathogens is a fascinating possibility that is worth investigating” - is it not?

5. Line 35-38: This statement is not corrected and not supported by the literature cited. The Tc are not known “as crucial virulence factors in numerous highly prominent...human pathogens”. Song et al. performed an *in silico* search for the abundance of TcC, which ADP-ribosylates eukaryotic target

13molecules, in the genomes of >133,000 bacterial genomes. The study identified N-termini of TcCs encoding RhsA and RHS core domains to be widespread among bacteria, but those N-termini are far away from a canonical toxin complex. Moreover, this study does not provide any data on the toxicity of newly identified domains.

According to the study we cite:

- “Moreover, the findings of this study are of wider significance because Tc toxin homologues have been shown to be encoded by a range of human pathogens. These include *Salmonella* and *Yersinia*, suggesting their potential roles in human infectious diseases”
- “TcC proteins are particularly prevalent in the class γ -proteobacteria (82.8%) and β -proteobacteria (13.3%), which include pathogens such as *Pseudomonas*, *Yersinia*, *Salmonella* and *Burkholderia*. This indicates that Tc toxins are widely distributed among bacteria, suggesting they could play an important role in the pathogenesis of infections”
- “Many *Yersinia* species, including important human pathogens *Y. pestis* and *Y. pseudotuberculosis*, are also adapted to infect insects, and Tc toxins are known to play a role in such a process” and
- “Some *Yersinia* species that have not been associated with insects, such as *Y. enterocolitica* which is a food-borne pathogen primarily found in mammals [51], and *Y. ruckeri* which is mostly isolated from fish [52], still encode Tc toxins.”

All of the above quotes from the cited paper fit the now-modified statement that “these non-phage-derived protein complexes are now known to be abundant virulence factors in numerous highly prominent insect and human pathogens”.

6. The microbiological background in the abstract and in the introduction is not state-of-the art and confusing. Abstract, lines 26-29: there is no toxin complex operon in *Salmonella enterica*, this is a misleading statement. In *Salmonella enterica*, TccC-like domains are conserved, but these are far from a toxin complex-encoding operon. The term Tc is restricted to insecticidal toxins with homology to the Tc of *Photobacterium luminescens*. The typhoid toxin is produced by *Salmonella enterica* serovar Typhi, but a holin is still missing. The endolysin probably involved in typhoidal toxin release is not a T10SS that by definition requires the tandem activity of a holin and an endolysin (Palmer et al. 2021, DOI: 10.1111/mmi.14599).

First, regarding the specific points raised here:

- There are in fact many subspecies and strains of *Salmonella enterica* that house complete Tc toxin operons, and not just TcC subunits. Please see an exhaustive list at <http://www.mgc.ac.cn/cgi-bin/dbTC/listtc.cgi?tax=Salmonella%20enterica> and the Song et al. paper referred to in point 5.
- We are not entirely sure what the reviewer means by the objection that “the term Tc is restricted to insecticidal toxins with homology to the Tc of *Photobacterium luminescens*”. All Tc toxins referred to in this paper have the general sequence hallmarks present in the originally described *Photobacterium luminescens* Tc toxin. Since their original description as insecticidal toxins, Tc toxins have also been shown to be active at least in vitro against various mammalian cell cultures (see e.g. Hares et al. 2008 or Leidreiter et al. 2019) and nematodes (see e.g. Sanger et al. 2023), which casts doubt onto their exclusively insecticidal nature also outside a laboratory setting – especially considering that intact Tc toxin operons are found in species not associated with insects (see Song et al. 2021). The Hinnebusch lab papers on the Tc toxin of *Yersinia pestis* (see e.g. Spinner et al. 2013) also point towards this.
- The typhoid toxin is not referred to in the abstract or introduction, as the reviewer suggests, but rather in the second-to-final paragraph of the discussion section, which is used to place T10SSs into the larger emerging paradigm of anti-eukaryotic toxin export by phage-derived proteins. Our statement that “notable examples [of anti-eukaryotic toxin export by phage-derived proteins] include ... endolysin-dependent secretion of typhoid toxin” neither claims that typhoid toxin export is mediated by a T10SS, nor contradicts the absence of a known holin involved in its export.

As for the generic statement that “the microbiological background in the abstract and in the introduction is not state-of-the art and confusing”, we find this to be unsubstantiated criticism. Our microbiological background is neither “not state-of-the art” nor “confusing”.

7. Same lines: *Yersinia pestis* is not an appropriate example for a pathogen that profits from insecticidal activity conferred by Tc. Although *Y. pestis* in vitro produces Tc, a role in the life cycle has not been demonstrated. Rather, it is assumed that the Tc of *Y. pestis* is not expressed (pseudogenes) or produced under in vivo conditions, e.g. in the flea, because it would otherwise harm its insect host. Therefore, it was suggested that a loss of insecticidal Tc activity was required for the evolution of this human pathogen (see papers by the Hinnebusch lab).

We have never claimed that *Yersinia pestis* profits from specifically insecticidal activity of its Tc toxin. In fact, the Hinnebusch papers which this reviewer refers to show that production of the Tc toxin is highly upregulated while within the flea (Spinner et al. 2012) yet are not toxic to the flea, in contrast to what the reviewer assumes. Follow-up experiments by the Hinnebusch lab furthermore show that the activity of the *Y. pestis* Tc toxin is crucial in modulating the innate immune response of the mammalian host immediately following transmission from the flea by inhibiting the phagocytic capability of polymorphonuclear leukocytes (Spinner et al. 2013), once again demonstrating that the concept of Tc toxins being exclusively insecticidal might be outdated. This is further suggested by independent experiments demonstrating that *Y. pestis* Tc toxin is also not active against e.g. *Manduca sexta* or cultured insect cells, but efficiently kills mammalian cells (Hares et al. 2008).

8. According to the literature, the Tc confers insecticidal activity. This is to be mentioned in the abstract.

As stated in our response to points 5, 6 and 7, there is evidence that Tc toxins may play a role in the bacterial life- and pathogenicity cycle also outside of an insect context. Therefore, we refrain from using the restrictive expression “insecticidal Tc toxins” in the abstract but rather use the more inclusive phrase “Tc toxins from human and insect pathogens” instead.

9. Summarizing the major points made above (title, abstract), the reader gets the impression that interesting data on self-destruction-dependent release of insecticidal toxin are overstated to imbed the insecticidal toxin and its release mechanism into the broader context of bacterial pathogenicity towards humans. However, the statement of lines 26-29 is not at all supported by the experimental findings reported in this manuscript.

The final statement of the abstract refers primarily to the three final paragraphs of the discussion section. As is apt for a discussion section, these consider the possibility that the T10SSs embedded in Tc toxin operons of human pathogens may play an important role in the life- and pathogenicity cycle of these bacteria. This is a reasonable possibility since Tc toxins from such bacteria have also been suggested to play an important role in their life-/pathogenicity cycle by the Hinnebusch, French-Constant, Waterfield and Fuchs groups. Nonetheless, we have now modified the final statement of the abstract to be less human-centric, reserving this primarily for the discussion section.

1610. Co-secretion of proteins in Fig. 2: it remains unclear why the secreted fraction is highly enriched with other (virulence) factors in comparison with those remaining in the cell. Essential details on the experimental settings are missing here. Which growth conditions were used to select for lysing cells? The control is MS with the “remaining fraction”, please clarify how this fraction was isolated. What is the difference to the cytoplasmic content? It is misleading to write remaining fraction when lysis occurs. Is there any selection mechanism of the exoproteins? Literature addressing this aspect by alternative concepts are missing here (Toyofuku et al. 2017, Nat. Comm. 8:481; Turnbull et al. 2016 Nat. Comm. 7:11220; Vacheron et al. 2021 Commun Biol 4:87).

We presume that by “**Fig. 2**” the reviewer is referring to **Fig. 2c-d**. Here, as per the figure legend, the samples are of the total cellular protein fractions before secretion (as controlled by the methodology described in the section “pH-dependent secretion of Tc toxins”), and are not separated into a “secreted” and “remaining” fraction. To eliminate any possible confusion about the origin of the samples, we have made a modification to the figure legend, as well as to the “**Proteomic analysis using NanoHPLC-MS/MS**” section of **Materials and methods**.

11. Overexpression of YenR used in Fig. 2c and 2d to study exoproteins is an artificial condition that might be different to the native condition, in particular with respect to cell lysis or related mechanisms. To summarize, the data provided in Fig. 1-2 cast the lysis hypothesis into severe doubt.

Our system does not overproduce YenR by multicopy plasmid overexpression, which is a well-established method but might lead to potential artifacts in some cases. Rather, we replaced the single genomic copy of the YenR promoter by an arabinose-inducible binary ON/OFF-switch. Both the behavior, pre- and post-explosive lysis phenotype, and identity of proteins released into the secreted fraction of the resulting strain have been validated here to match those found in YenTc cells with the native YenR promoter (see e.g. **Fig. 1a**, **Fig. 1c**, **Fig. 2e**, **Supplementary Fig. 8** and the corresponding lanes of **Fig. 2a** for WT YenR cells versus e.g. **Fig. 1d**, **Supplementary Fig. 9c-d** and the corresponding lanes of **Fig. 2a** for Ara-YenR cells, as well as the **Source data** file). Therefore, neither the data provided in **Fig. 1**, **Fig. 2** nor elsewhere in this study casts doubt onto our observations of YenDF-mediated lysis of soldier cells.

12. Fig. 2e: This finding requires thorough quantitative analysis rather than providing a single image.

This is a fair point, and has been addressed by adding **Supplementary Fig. 6**.

13. Use of the fusion YenA1-sfGFP: in this construct, GFP is fused after residue 37 of the YenA1 subunit. Thus, the authors applied a translational fusion of ~275 residues and 30 kD to monitor the release of a high molecular weight toxin subunit. However, there is no YenTc-sfGFP construct mentioned in the supplementary material, whereas YenA1-sfGFP is not found in the main text. Please clarify. More important, such a construct with 37 residues of YenA only does not match the natural conditions, and a full-length fusion or anti-YenA1-antibodies must be used for monitoring the release mechanism.

We thank the reviewer for noticing this issue and bringing it to our attention. The sfGFP reporter strains used either had an sfGFP fused to YenA1 after residue 37 (this strain was subsequently used to generate several sub-strains) or had an sfGFP fused to the C-terminus of YenA2. We did not observe any difference in the behavior of the two strains. We have now ensured that all strains and derivatives are correctly labelled either YenA1-sfGFP (for the truncated YenA1 fusion) or YenTc-sfGFP (for the YenA2 fusion).

We have now also appended **Supplementary Fig. 5**, which validates that YenTc-sfGFP released from the soldier cells by YenDF-mediated lysis assembles into full YenTc complexes in the same manner as the sfGFP-free version, and we confirm the expected composition of secreted YenTc by mass spectrometric analysis as now shown in the **Source Data** file. The text has been modified correspondingly. Furthermore, **Fig. 3**, **Fig. 4**, **Supplementary Fig. 10** and **Supplementary Fig. 12** directly visualize and validate the YenDF-mediated release mechanism of YenTc at 33000x – 64000x magnification.

14. The introduction of new term such as “soldier cells” and “death factor” (DF) is confusing. Soldier cells is a new term that the authors define as a small subset of specialized cells. What is the reason for this wording? Many cellular activities require specialization. Soldiers do a lot, but rarely undergo suicide. The term “soldier” might be eye-catching, but its use in the context of T10SS is not appropriate and misleading. Please replace with “self-destructive cells”, which refers to a well-known concept in bacterial pathogenicity (see publications by the Hardt and Ackermann labs, e. g. Nature 454:987 (2008)).

We appreciate the reviewer’s reference to the Ackermann et al. 2008 study modelling self-destructive cooperation in bacteria, which we have now included in our text. While expressions such as self-destructive cooperation are well-known in the community, the term “self-destructive cells” suggested by the reviewer is to the best of our knowledge not used in publications referring to bacterial pathogenicity and is therefore also novel. As we show here, YenTc-expressing cells are noticeably larger than their isogenic non-producing brethren, and differ significantly in both their proteomic composition and behavior – in other words, represent a specialized cell type in this bacterial species. Cell types, in turn, are traditionally given unique names. In our case, “soldier cells”. At the same time, we realize that we have not explained the logic for our decision to choose the term soldier cells in the text, which we have now amended to do so. Hopefully the reasoning is now clearer.

Same for “death factor”: there are numerous publications now that describe or hypothesize the role of lysis cassettes in toxin release, prompting Palmer and coworkers to establish new a type 10 secretion system. Even if it might turn out in the future that toxin release is mediated by cell lysis rather than by controlled secretion, the T10SS concept is very helpful to improve our understanding of toxin release via holin and endolysin. The term “Explosion-competent” should be introduced by citing respective literature, otherwise use “lysis-competent” instead.

The name “*Yersinia entomophaga* death factor” (“YenDF”) is not a replacement for the well-accepted terms “lysis cassette” or “T10SS”. Rather, it is the name we have given to this *particular* T10SS based on the *Yersinia entomophaga* cell death phenotype that manifests as a result of its lytic action. Giving names to genetic elements in this manner is standard scientific practice (see e.g. the naming of the eds system and BAS1664 locus by Laut et al. 2020). Therefore, the name YenDF in no way contradicts the terms “lysis cassette”, which is generally reserved for holin/endolysin/spanin elements of (cryptic / pro)phage context, nor “T10SS”, which represents similar elements outside of a such a context. As with “soldier

cells”, the text has been expanded to better accommodate the logic underlying the name. Also, the term “explosion-competent” was modified as the reviewer suggested.

15. The nomenclature of the regulatory factor identified in this study, “YenR”, is another example for an insufficient microbiological background in the study. YenR is a regulator of *Yersinia enterocolitica* involved in quorum sensing systems (Atkinson et al., Mol Micro 1999), and its sequence is distinct from “YenR” of *Y. entomophaga*. Please replace with “RoeA” (Schoof et al.).

YenR is the *Yersinia enterocolitica*-specific name of the autoinducer-1 quorum sensing transcriptional regulator LuxR. In nearly every other bacterial genome, including its only homologue/orthologue in *Yersinia entomophaga*, it is annotated as LuxR (or a LuxR family protein). Therefore, we consider YenR to be an available name for the *Yersinia entomophaga* transcriptional regulator described in our study and wish to keep it.

Moreover, the alternative name “RoeA” from the concurrent Schoof et al. study suffers from the same drawback as the one the reviewer tries to use as an example of an “insufficient microbiological background in [our] study”. The name RoeA - regulator of exopolysaccharide A - has already been given to a bacterial diguanylate cyclase (see e.g. Merritt et al. 2010, Dahlstrom et al. 2015), and *Yersinia entomophaga* already possesses several diguanylate cyclases and GGDEF-domain containing proteins that could be considered reasonably homologous to RoeA.

General remarks to be addressed:

16. Introduction: please provide more information on YenTc. Compare subunit nomenclature used in this study with the nomenclature of *P. luminescens* Tc subunits. What are the (putative) functions of the YenTc subunits? Why targeting YenA1? Tc stoichiometry?

The nomenclature of the YenTc subunits is in line with the study by Piper et al., 2019 as cited in the text. As the reviewer suggested, we have added a summary of the YenTc subunits and their putative functions

to the introduction, and agree that it helps a broader audience better appreciate the structural complexity of YenTc.

17. Supplementary Fig. 1a: The legend and the methods should comprise more details on protein separation including the imaging system. Here and in the main manuscript, data on the molecular weight of YenTc components are missing. Please comment on the bands with higher and lower MW than YenTc. How were YenA1, YenA2, and YenB identified here and elsewhere, just to their lack in the Tc deletion mutant? Other proteins are also missing upon Tc deletion. Antibody-based detection is missing here.

We have updated the “**Cell growth and secretion assay conditions**” section of the **Materials and methods** with more precise information on the protein separation and imaging system. We have now also appended **Supplementary Fig. 5**, which together with the mass spectrometric analysis in the **Source Data** file confirms the expected composition of YenTc and the molecular weights of its components. A comparison of **Supplementary Fig. 3** and **Supplementary Fig. 5** shows that the other proteins missing upon Tc deletion are in fact smaller subunits of the YenTc complex.

18. Fig. 1 The map of YenTc should be improved; cf. to those provided by Hurst and coworkers in numerous papers.

We would like to kindly point out that there is no “map of YenTc” in **Fig. 1**.

19. Fig.S4: Deletion of a lipase significantly reduces the amount of YenA1 released, suggesting its role in Tc release similar to a lipase of *P. luminescens*.

We would like to thank the reviewer for drawing our attention to this issue, which was caused by degradation of the Δ Lipase #2 sample due to improper storage overnight. The knockout strains for that gel were re-grown and the secretion experiment re-done, with the updated **Supplementary Fig. 3** now clearly

21showing that deletion of Lipase #2 does not affect the release of YenTc and other proteins from soldier cells.

20. Fig. 2b: no soldier cells appear in the absence of induction? This contrasts data shown in Fig. 1b, please clarify.

No soldier cells appear in the Ara-YenR strain without arabinose – and on the contrary, all cells of said strain turn into soldier cells in the presence of arabinose (**Fig. 2b**), due to the switching of YenR transcription from an OFF to an ON state. This is further elaborated upon by the analyses of proteomics data in **Fig. 2c** and **Fig. 2d** from the corresponding section of the **Source Data** file. These data clearly demonstrate that the YenR gene needs to be transcribed in order for soldier cells to appear, a process that normally occurs in only a small subset of cells, as per **Fig. 1b-c**, **Fig. 2e** and **Supplementary Fig. 6**, where the YenR promoter is not arabinose-sensitive.

21. Lines 126: Did the authors exclude the presence of secretion system duplications in the genome of *Y. entomophaga*?

This is correct. Any (potential) secretion system duplications were targeted, as per **Supplementary Fig. 4**.

22. Fig. S5n, the experiment requires more details to ease reading. Which conditions were applied? How was secreted YenTc-GFP identified and measured?

The experimental conditions applied are provided in detail in the legend of **Supplementary Fig. 4a-m**. Details of the experimental procedure used to generate **Supplementary Fig. 4n** and measure fluorescence of the sfGFP reporter fusion are already described in the corresponding legend and in the “**Fluorescence spectroscopy**” section of the **Materials and methods**.

2223. Fig. S6: “raft-like oligomers” are not clearly visible.

Technically true. The figure legend has been modified accordingly.

Minor points:

1. A consistent reference style should be used

We thank the reviewer for the comment; however, we have not found examples of a non-consistent reference style.

2. Line 36: “non-phage-derived” anticipates that factors involved in T10SS are phage-related, but this is unknown for many readers not familiar with the topic.

The adjective “non-phage-derived” refers not to the T10SS components, but rather to the Tc toxins themselves (see previous sentence; the T10SSs have not been introduced yet). Many toxins of similar scale – e.g. pyocins and PVCs – are thought to be derivatives of phage tails.

3. Line 82: “...that has instead evolved to target insects”. This statement suggests that a *Yersinia* ancestor split into two lines evolving towards human or insect pathogenicity and thus misunderstands the yersinial evolution. In fact, *Y. pestis* only recently evolved from *Y. pseudotuberculosis*, and it is assumed that

23many *Yersinia* species were adapted to invertebrates before the evolution of vertebrates. This sentence should be rewritten.

This is a good point. While the original phrasing did not intend to mean an ancestral split towards human versus insect pathogenicity, it could be misread as such and was therefore modified.

4. Table S1: “to CmR” means respective gene was exchanged with CmR resistance?

This is correct.

Reviewer #2

This is another amazing paper on the Tcs so congratulations on that.

All of my comments are really about 'framing' of the study and not really about the science as such.

We thank the reviewer for their useful points and comments, and are delighted to see that they share our excitement regarding this story.

1. Soldier cells is an attractive name. But do these cells die prematurely in the secretion process (do they bleb themselves to death)? If so maybe kamikaze cells is better?

We appreciate the reviewer's suggestion and agree that "kamikaze cells" is a good alternative name for this specialized cell type in *Y. entomophaga*, which we now also wove into the text. Our source of inspiration for the name "soldier cells" was however not human warfare, but rather the large soldier caste individuals of certain termite species, which are known to self-sacrificially "explode" for the benefit of their colony. We have now added the explanation of the term's origin to the text.

2. Why does *Photorhabdus* (PI) not use this mechanism? As I understand it the tc encoding genes are found on the pADAP plasmid of Ye and are not incorporated into the genome as they are for PI. So if the PI Tcs are able to secrete themselves then perhaps the Type10 secretion is an intermediate/first step specific to plasmid encoded Tcs?

Whether the mechanism of *Photorhabdus* is holin/endolysin/spanin-driven as here or uses an alternative mode of lytic release (e.g. by single-gene lysis proteins) remains an interesting topic for a future study. The example of *Yersinia entomophaga* demonstrates that the secretion system responsible for the release of Tc toxins may be encoded in a genomically distal manner from the Tc operon, which complicates such investigations. Perhaps full mass spectrometric analyses of the secreted *Photorhabdus* proteome could represent a good approach to reveal crucial lysis components, as was the case for YeEln in this study. One thing seems to be certain from the cryo-ET data we show here: Tc holotoxins are fully assembled in the bacterial cytoplasm and, provided their size and lack of known signal sequences, must therefore be released lytically from the producing cells.

The pADAP plasmid referred to by the reviewer comes from *Serratia entomophila*, whereas in *Y. entomophaga*, the Tc toxin genes are genomically encoded. Tellingly, the Tc toxin encoded by the pADAP plasmid is adjacent to a holin and lysozyme, hinting that the reviewer might be right with their last suggestion.

3. 182 'has evolved to target insects'. This is a rather anthropomorphic view of things? Invertebrates form the vast majority of infectable animals on the planet and infection of vertebrates is therefore secondary- which may also relate to argument 2? So perhaps vertebrate pathogenicity is associated with a) plasmid borne TCs and b) type 10 secretion? Either way *Yersinia* started killing insects and perhaps accidentally has infected humans as *Y. pestis*.

25We agree with the reviewer that the original statement could be misinterpreted in the indicated manner, and have therefore modified the sentence.

While the idea of plasmid-borne Tc toxins being a step towards vertebrate-associated Tc pathogenicity is an interesting one, the example human pathogens mentioned in this study that harbor Tc toxins with T10SSs encode them genomically. E.g. *Y. pestis*, the Tc toxin of which seems to be directed towards mammalian pathogenesis as per studies of the French-Constant, Waterfield, and Hinnebusch groups.

4. In PI Tcs can be found either in the supernatant alone or cell associated. Previously we had assumed that this 'release' into the supernatant was associated with a lipase gene encoded nearby. Do the Type10 finding change this interpretation or offer a better one?

Regarding this, we can refer the reviewer to our answer to their point 2. Due to the lack of a published full genomic sequence, we cannot make an informed statement on T10SSs in the context of the *P. luminescens* W14 strain, which contains the Pdl1 lipase in question. However, the closely related, Pdl1-lipase free T101 strain contains a T10SS distal from Tc toxin operons in its genome (in a manner similar to what we found in *Y. entomophaga*), encoded by the genes plu3747-3749. Whether the lipase serves as a replacement for this T10SS in strain W14, complements it, or serves a different function would be a good question to answer in a future study.

5. The links with holin/endolysin/spanin and bacteriophage are fascinating? Does this mean that the Tcs are themselves actually modified bacteriophage and what implications does this have for their evolution and/or secretion as in 4 above (are 'lipases' important in the release of bacteriophage too?).

The Tc toxins themselves do not appear to be a structural and functional modification to bacteriophages in the same way that e.g. pyocins, PVCs and (possibly) T6SSs are. However, this does not exclude the purely speculative possibility of their original Shiga-toxin like transmission by bacteriophages co-

encoding lysis cassettes, with evolutionary pressure decoupling Tc toxin production and secretion from the bacteriophage lifecycle by removing non-essential phage genes.

To the best of our knowledge, lipases are not known to be involved in bacteriophage release. While most bacteriophages use holin/endolysin/spanin lysis cassettes for release, some smaller phages such as coliphage ΦX174 encode single-gene lysis proteins. The 2020 review by Chamakura and Young provides a good overview of these.

6. Finally, 1351 'cellular targets'. What are the likely targets of these Tc containing vesicles? Are they likely to be taken up by endocytosis? What are there 'receptors' or do they just fuse with target cells in a non-specific manner?

Based on our tomographic data shown in **Fig. 3** and **Fig. 4b**, YenTc is released into the surrounding environment after spanin-mediated membrane fusion rather than becoming entrapped in vesicles. Although we cannot exclude the latter may occasionally happen if a Tc toxin is not ejected out of the exploding cell properly, we have so far not observed this. In a 2019 study by Piper et al., the chitinases of YenTc were proposed to function as glycan-sensing receptor binding domains which would mediate cell attachment. This fits well with the observations of glycan-mediated attachment of Tc toxins by our group as well as the Yang and Schmidt/Aktorries groups. To reach their targets in the insect midgut (as determined by the Hurst group), the YenTc toxins would however need to breach the protective chitinous peritrophic matrix. As we speculate in the discussion section, this may be why *Y. entomophaga* soldier cells also produce and release several chitinases and chitin-binding proteins in large amounts along with the Tc toxin.

None of these arguments are mission critical but I hope they raise points for interpretation and/or future work.

We thank the reviewer for the interesting points and useful ensuing discussion.

Reviewer #3:

Summary

Sitsel et al. characterize the previously unknown mode of secretion for Tc toxins, using *Y. entomophaga* YenTc as a model. They find that a large number of other virulence factors and cytoplasmic proteins are released into the environment along with YenTc via a controlled lysis mechanism in a subset of cells that they term “soldier cells”, and the T10SS, named YenDF, is responsible for this secretion. Temperature, pH, and other factors may drive the mechanism. They use cryo-ET with YenDF component knockout cells to observe the state of the cell and morphology of the cell envelope at different stages of secretion. This allowed them to propose a model for how holins, endolysins and spanins each contribute to membrane changes and cytoplasmic content release. Their proposed mechanism is consistent with observations in *S. marcescens*, suggesting that it may be wide-spread. This study was interesting, with many orthogonal pieces of data, and the data along with the proposed model for T10SS mechanism is likely to be of broad interest to microbiologists. We thank the authors for posting their work on BioRxiv, which makes it available to the community, and accelerates new discoveries based on their work.

We would like to thank the reviewer for their positive feedback and interest in our paper. We found the reviewer’s suggestions regarding quantification of several types of data to be particularly helpful in strengthening the conclusions we have reached in this study.

Major Comments

Fig 1A: We did not follow why the “secreted fraction” contains such few hits. If the mechanism is from controlled lysis, and the “secreted” fraction is then actually just released proteins as a result of cell lysis, why aren’t all proteins enriched as compared to non-secreted fraction? Perhaps we misunderstood this experiment, and a clearer explanation of how this is carried out, what the “secreted” fraction is, and how this relates to soldier/non-soldier cells would help.

We thank the reviewer for bringing this up and were glad to expand upon the experiment to make it clearer. This has now been done in both the “**Proteomic analysis using NanoHPLC-MS/MS**” section of **Materials and methods**, **Fig. 1a** legend, as well as the text – when the observation was originally described, as well as once the phenomenon of controlled lysis by soldier cells was identified in order to explain the previously perplexing observations of **Fig. 1a**.

The modifications should now hopefully make it clear what the nature of each fraction is, as well as that the significantly enriched proteins of the secreted fraction highlighted in **Fig. 1a** comprise only a small subset of the total proteins identified in that fraction, and the relationship of this fraction to the observation of lysing soldier cells.

Fig. S4 suggests that all secreted factors are dependent on T10SS, yet other secretion systems are encoded. Is it known why the bacteria encodes all the other secretion systems if they are not used - are they expected to be used only under certain conditions? Are the smaller virulence factors unable to be secreted by any of the typical export pathways, only cell lysis? The lack of signal sequence for all the virulence factors is peculiar. Is this typical for *Yersinia* species? Some explanation of this to contextualize the use of T10SS in relation to others would be helpful.

All proteins secreted by *in vitro* cultures of *Y. entomophaga* are indeed dependent on the T10SS, as **Supplementary Fig. 3** testifies. The general lack of secretion signal sequences as per **Supplementary Fig. 2** appears to be a general characteristic of T10SS-dependent substrates. Of these, only NucA and Chi3 have been predicted to possibly also use the Sec pathway to move to the periplasm, but no other avenue of their further export to the extrabacterial environment is known, apart from release by T10SS.

The other secretion systems *Y. entomophaga* encodes will however find other uses under other environmental conditions, e.g. detoxification of antibacterial compounds, bacterial interspecies competition, or host infection after the insect midgut has been breached during the initial stages of *Y. entomophaga* infection. Apart from the YenR and T10SS-associated virulence factors of *Y. entomophaga* that we explore here, this species encodes multiple others, e.g. hemolysins, MsgA-, SrfB- and MviM homologues, as well as numerous T3SS and T6SS effectors. We have modified the legend of **Supplementary Fig. 3** to indicate that the conclusions shown there apply to the conditions tested.

For all light microscopy, could the authors please provide quantification, and clearly indicate how many cells are analyzed

We thank the reviewer for the excellent idea, and consider it to be a good way of improving the study. These quantitative analyses are now included in the relevant figure legends.

For tomography data, please indicate how many cells were inspected in each case that leads to a conclusion, and some estimation of the variability observed. We completely understand that cryo-ET is a low-throughput technique at this time, but, given the general heterogeneity of cells when observed at these resolutions, it would be very helpful to gain an understanding of the variability, and the numbers used.

As with the previous point, we appreciate the reviewer's proposal and believe it will further clarify the observations in our study. We have therefore added the section "**Ultrastructural analyses of vitrified bacteria**" to the **Materials and methods**, which covers this in detail.

In some cases, we suggest that conclusions from cryo-ET be qualified. For example, the right inset in Figure 3d - may be an unfused vesicle, or may be something else. Since it is very difficult currently to use cryo-ET tags to unambiguously define what one is looking at (unless sufficiently high resolution is achieved using STA for protein complexes), qualifying the identification of any object in the tomograms (with less certainty) would be reasonable.

Another good point. We have now added relevant qualifiers for the elements of cryo-ET data that cannot be definitively identified.

The overall narrative, and the order of figures is somewhat scattered, which makes reading through the whole manuscript quite challenging. For example, Fig. S2, which contains soldier cells, is referenced at

the beginning of the results, but the concept of soldier cells has not yet been introduced. It therefore took substantial effort to go back and forth through the text and figures, and keep all the results straight. We would strongly encourage the authors to ensure that the flow and logic of the manuscript is linear, and that the figures are arranged accordingly.

We thank the reviewer for assessing the flow of the manuscript with a fresh eye. While we partially attribute the complexity of the manuscript to the many orthogonal pieces of data that needed to be pulled together into a single narrative, we agree that the way some data was scattered between the figures did not make the reader's job easier. We have therefore now renumbered and rearranged figures wherever we could do so (without destroying the internal cohesion of multipaned figures) in order to better fit the flow of the text and reduce the amount of back and forth scrolling required.

Minor comments

Line 137-138/Fig S5: This figure does not seem to show that low temperature has the strongest influence on YenTC production, as stated in the text?

According to **Supplementary Fig. 4n**, elevating the temperature reduces the chance of cells becoming YenTc producers the most, meaning that lowering the temperature conversely increases these chances. We have now removed the qualifier "lower" from the text to avoid confusion, since our control conditions were at low rather than elevated temperature.

Please label which bands correspond to YenA1, YenA2 and YenB on boxed YenTc for all gels
Please explicitly explain how YenA1/A2 and YenB correspond to TcA and TcB

To address this in more detail and give a band reference for all gels that have YenA1, YenA2 and YenB boxed together as YenTc for the sake of avoiding visual clutter, we have now appended **Supplementary Fig. 5**, which provides a description of all YenTc bands visible on the SDS-PAGE gels, validates the final integrity of the resulting complex and confirms the expected composition of secreted YenTc by mass spectrometric analysis as now shown in the relevant section of the **Source Data** file. We have now also extended the introduction to introduce the components of YenTc in more detail.

Is there speculation on why an anoxic environment inhibits YenTc secretion but not production? This is just a curiosity question

This is a very good question. Since the production of YenTc and the YenDF T10SS is coupled, we can assume that cells grown in an anoxic environment have levels of both similar to control conditions, however have incurred changes that reduce the efficiency of lysis by YenDF. A possible point of difference between aerobic and anaerobic conditions might be the composition and arrangement of the peptidoglycan layer, which YeEln must cleave in order for YenDF-mediated lysis to occur. While this has been shown to happen in Gram-positive organisms (O'Brien et al., 1971), we were however not able to find relevant work regarding the effect of aerobic vs anaerobic growth on Gram-negative peptidoglycan.

The figure legend for Fig S2h seems to be missing

Thank you for noticing, this has now been amended.

Fig 4C show Chi 1 and Chi 2, which were never introduced

Good point. This is now fixed by introducing the YenTc components early on in the text.

Fig 4C: Where is YenB?

We have modified the figure to replace the more generic TcB/TcC notation we used with the more specific label YenB+YenC1/C2/RHS2.

Fig 5A: What are M and T fractions?

These are the secreted and cytoplasmic protein fractions, which we now relabeled S and C for clarity. We thank the reviewer for pointing this out, as we forgot to explain this in the figure legend, which we have now amended.

S2F,G,H - what is the status of GFP+ cells for 37C? It was not clear to us how the morphology correlated to being GFP+/-

The cells used in **Supplementary Fig. 7c-e** are not from GFP reporter strains. The purpose of these panels was to support the data of **Supplementary Fig. 2a-b**, showing that at 37 °C, protein release is absent and the cells fail to switch to the enlarged, explosive lysis-competent soldier cell phenotype even when YenR induction is attempted. A GFP reporter perspective on this is provided in **Supplementary Fig. 4m-n**.

Decision Letter, first revision:

Message: Dear Professor Fuchs,

Thank you for reviewing the manuscript "Specialized pathogenic cells of Yersinia entomophaga release Tc toxins using a type 10 secretion system" for Nature Microbiology.

Based on your and the other referees' comments (appended below, for your reference), we have invited the authors to revise the manuscript and resubmit the paper after addressing the criticisms raised. If we do receive a revised manuscript, I hope that we shall be able to contact you to review it again.

We realize that your schedule is full and we very much appreciate the time you've taken to

33provide us with a constructive review.

Thank you again for your help and I hope that we can call upon your advice in the future.

Best wishes,

[redacted]

Reviewers Comments:

Reviewer #1:

Remarks to the Author:

Comments to the author: Many of my concerns were adequately addressed. However, some points still require substantial modifications of the manuscript.

Major points:

Additional comment: With respect to T10SS of chitinases in Salmonella, Krone et al. ([/doi.org/10.1371/journal.ppat.1011306](https://doi.org/10.1371/journal.ppat.1011306)) should be cited.

1. Comment considered.

2. Comment considered.

3. In vivo aspect: I did not question that the TC plays a role in vivo. Rather, I stressed that it cannot be excluded that there is a distinct mechanism during infection of invertebrates. Therefore, I still argue that conclusions on a new mechanism that play a role in infection in vivo are not supported by the data.

4. Comment considered.

5. To be honest, the arguments cited from the Song et al. paper do not at all support what the authors state in their abstract. The mere existence of a tc gene in a human pathogenic bacterium does not allow the conclusion that in those pathogens, the TCs play a role in virulence towards humans. However, such an association is made by the authors few times in the manuscript (details see below). According to the literature, there is (yet) no proven example for a TC contribution to bacterial virulence against human beings. This does not exclude the relevance of T3SS for such activities, but the well-documented examples known so far do include other toxins instead of TC.

To conclude, I suggest to rewrite the following sentences: line 28, lines 36-37, lines 434-435.

6. The list of Salmonella genomes harbouring TC operons is a good example for how the authors overstate microbiological data. The database of TCs from Salmonella comprises a list of ~150 Salmonella genomes, many of them grouping into few serovars. To my knowledge, there are more than 10,000 validated Salmonella genome sequences and more than 470,000 unvalidated (shotgun sequencing, metagenome data etc. to which many of the only 150 sequences belong to) Salmonella genomes. Random sample analysis including control of the annotation revealed that a holin/endolysin is missing in many of these sequences. The example given in Supplementary Fig. 14 is *S. houtenae*, an opportunistic pathogen that is commonly found in reptiles and other animals. The adequate conclusion

34from these data is: TC operons are an evolutionary relict in the Salmonella genus, and it is absent in nearly most of the relevant serovars responsible for pathogenicity against humans worldwide. Moreover, many experts in the field state that yet neglected natural hosts of many human pathogens are invertebrates, a fact that very simply explains the presence of Tc operons in the genomes of "human" pathogens. To address here another argument of the authors: the insecticidal toxin complex indeed plays a role in the life cycle of human pathogens, but only as a key factor in an invertebrate-associated stage. 7. Comment considered (Yersinia and Salmonella are not further mentioned in the abstract).

8. The authors are wrong here. Tc confers insecticidal activity according to the literature. This does not exclude adaptation of the TCs to a function towards mammals, but few studies showing slight or specific effects on mice and human cells do not allow general positive conclusions on an in vivo role in mammals. None of the arguments provided by the authors really support their theory that Tc are abundant virulence factors in human pathogens. Such a statement semantically suggests that TC contribute to virulence towards humans. Literature, however, on such a conjunction is largely missing. Therefore, to not overstretch scientific data, TCs are to be termed as insecticidal toxins in the abstract and in the introduction. The sentence in the abstract ("such as TC toxins from human and insect pathogens") is not supported by the literature and therefore misleading.

9. Comment considered.

10. Comment considered.

11. Comment considered.

12. Comment only partially considered. In legend Fig. 2, please write "For quantitative data, see supplementary Fig. 6. The quantitative data (percentages) should be mentioned in the main text.

13. Comment partially considered. For consistency, please use "YenA2-sfGFP" and "YenA2-His" instead of "YenTc-sfGFP" and "YenTc-His" throughout the text.

14. Comment on soldier cells considered.

Regarding the term "death factor", I have major concerns about using such a term to rename identical factors from the same species. Again, genetic nomenclature must be as consistent as possible to not confuse the readers in the field (see also next comment). "YenDF" was already named "holA" and "pepB" in *Y. entomophaga* by Schoof et al., and there is no need to introduce a new abbreviation here. Lines 356-358 should accordingly be removed.

I agree with the modification made by the authors with respect to the term explosion.

15. I strongly disagree. The name YenR is already introduced in the literature for a Yersinia regulator distinct from that mentioned in the manuscript. Moreover, the name "YenR" is highly unspecific as "Yen" might mean *Y. enterocolitica* or *Y. entomophaga*, and "R" for regulator is very unspecific as well. RoeA in contrast was introduced for the TC-related regulator in *Y. entomophaga* by Schoof et al. The term RoeA is indeed used for factors of other bacteria, but not in Yersinia species. To not confuse the community with two distinct names for the same protein, the authors should accept the term RoeA as already established.

General remarks to be addressed:

16. Comment considered.

17. Again, the molecular weights of YenTc components should be mentioned in the main text and in one full figure legend to ease understanding (and not only in the source data

35file). In Fig. S5, there is an obvious discrepancy between the molecular weights indicated in the source data file and the bands representing the toxin subunits. For example, YenA2 has a MW of ~180 according to the marker in contrast to the theoretical value of 156.18. This needs to be clarified

18. At the bottom of Fig. 1, the authors show a draft/a map/a cluster of the YenDF operon that should be improved according to e.g. maps provided by Hurst and coworkers in numerous papers such as doi.org/10.1371/journal.pone.0263019 and shown separately in Fig. 1 as genetic map/cluster.

19. Comment considered.

20. Comment considered.

21. Comment considered.

22. Comment considered.

23. Comment considered.

Minor points:

1. References: Photorhabdus in ref. 3. Use of capital letters in ref. 1, 23 etc.

2. Line 36: "non-phage-derived" is mentioned too early here in the manuscript. Please remove here and use this phrase only when the holing/endolysin system is introduced.

3. Comment considered.

4. Please write: Allelic gene replacement with CamR.

5. Line 81: comma between pestis and that.

Reviewer #2:

Remarks to the Author:

Thanks for taking the time to revise the paper— it now looks great!

Reviewer #3:

Remarks to the Author:

The authors have addressed my comments and added in several quantifications that are helpful. I have no further comments and congratulate the authors on a very interesting body of work.

A quick comment on Major point 1 of Reviewer 1. Since the Sitsel et al manuscript was made available (Feb 22, 2023) prior to the Schoof et al manuscript (March 23, 2023), it would be unreasonable to have expected a discussion of the Schoof et al manuscript in the current version of the Sitsel et al manuscript. At this point, it would be reasonable for the authors to comment on any differences in results, interpretation and opinion, where the authors feel it is applicable.

Author Rebuttal, first revision:

36Point-to-point response to the reviewers' comments

We would again like to thank the reviewers for their feedback and comments, which helped us to deliver the polishing touches to the manuscript. Below is a point-by-point response to all remaining comments and a description of changes we have made to our manuscript when this was necessary. The changes in the revised abstract and text related to our responses to the reviewers have been highlighted in yellow. The abstract and text have now also been shortened to fit the limit of 150 and 3500 words, respectively, to fit the requirements of *Nature Microbiology* articles.

Reviewer #1

Many of my concerns were adequately addressed. However, some points still require substantial modifications of the manuscript.

We are happy to hear that we have addressed most concerns of this reviewer. For the remaining points, we have either made modifications to the manuscript or further substantiated our statements with reference to the available scientific literature.

Major points:

3. In vivo aspect: I did not question that the TC plays a role in vivo. Rather, I stressed that it cannot be excluded that there is a distinct mechanism during infection of invertebrates. Therefore, I still argue that conclusions on a new mechanism that play a role in infection in vivo are not supported by the data.

As we pointed out earlier, the recent *in vivo* studies of the Fuchs group (PMIDs 36399504 and 37184385), which we cite in this manuscript when discussing the biological relevance of our findings, deal specifically with this question. Those studies demonstrated that deletion of the *Yersinia enterocolitica*

37T10SS results in a phenotype as avirulent as the *Yersinia enterocolitica* Tc toxin deletion mutant, providing convincing evidence that the T10SS-mediated release mechanism of Tc toxins that we describe here is in fact crucial for establishing an infection in invertebrates.

5. To be honest, the arguments cited from the Song et al. paper do not at all support what the authors state in their abstract. The mere existence of a tc gene in a human pathogenic bacterium does not allow the conclusion that in those pathogens, the TCs play a role in virulence towards humans. However, such an association is made by the authors few times in the manuscript (details see below). According to the literature, there is (yet) no proven example for a TC contribution to bacterial virulence against human beings. This does not exclude the relevance of T3SS for such activities, but the well-documented examples known so far do include other toxins instead of TC.

To conclude, I suggest to rewrite the following sentences: line 28, lines 36-37, lines 434-435.

The Song et al. paper abstract that the reviewer refers to states that:

- “Tc toxins were *originally* identified in entomopathogenic bacteria”
- “Currently the pathogenic roles and distribution of Tc toxins among different bacterial genera remain unclear” and
- “It furthermore implies that Tc proteins, which are encoded by a wide range of pathogens, represent an important versatile toxin superfamily with diverse pathogenic mechanisms.”

Furthermore, the author summary of that study clearly states:

- “Moreover, the findings of this study are of wider significance because Tc toxin homologues have been shown to be encoded by a range of human pathogens. These include *Salmonella* and *Yersinia*, suggesting their potential roles in human infectious diseases.”

The quotes from the main body of Song et al. that we earlier presented in response to point 5 of the reviewer are fully supported by these statements.

We however do agree with the reviewer’s argument that the presence of an intact Tc toxin operon in the genome of a human pathogen does not automatically equate to its use during human infection. In fact, this remains one of the most important topics for future research in the field and studies that could definitively answer this by investigating the role of Tc toxins in infection of vertebrates on a level more complex than cell cultures have yet to be conducted. At the same time, as we explained earlier, the Hinnebusch lab showed that the *Y. pestis* Tc toxin targets components of the mammalian immune system rather than kills

its intermediate insect host (Spinner et al. 2013), notwithstanding the multiple studies which show that Tc toxins kill human cells very efficiently. We have therefore acknowledged this issue in the introduction.

Our statements the reviewer questions are:

- “With T10SSs directly embedded in Tc toxin operons of major pathogens, we anticipate that our findings may model an important aspect of pathogenesis in bacteria with a significant impact on agriculture and global human health.”
- “Originally discovered in the insecticidal bacterium *Photorhabdus luminescens*, these protein complexes are now known to be abundant virulence factors in numerous highly prominent insect and human pathogens” and
- “*Y. entomophaga* serves as an excellent proxy for dangerous human pathogens that employ Tc toxins”.

Neither the first nor the second statement are incorrect. They do not imply that Tc toxins have been definitively determined to play a role in human infections, but rather in the lifecycle of major human pathogens. In fact, they must: without selective pressure to maintain them in a functional state, the very large Tc toxin operons would easily become pseudogenized. We can however see how the third statement could be misread and it has therefore been removed. The first and second statements were also rephrased.

6. The list of Salmonella genomes harbouring TC operons is a good example for how the authors overstate microbiological data. The database of TCs from Salmonella comprises a list of ~150 Salmonella genomes, many of them grouping into few serovars. To my knowledge, there are more than 10,000 validated Salmonella genome sequences and more than 470,000 unvalidated (shotgun sequencing, metagenome data etc. to which many of the only 150 sequences belong to) Salmonella genomes. Random sample analysis including control of the annotation revealed that a holin/endolysin is missing in many of these sequences. The example given in Supplementary Fig. 14 is *S. houtenae*, an opportunistic pathogen that is commonly found in reptiles and other animals. The adequate conclusion from these data is: TC operons are an evolutionary relict in the Salmonella genus, and it is absent in nearly most of the relevant serovars responsible for pathogenicity against humans worldwide. Moreover, many experts in the field state that yet neglected natural hosts of many human pathogens are invertebrates, a fact that very simply explains the presence of Tc operons in the genomes of “human” pathogens. To address here another argument of the authors: the insecticidal toxin complex indeed plays a role in the life cycle of human pathogens, but only as a key factor in an invertebrate-associated stage.

The reviewer had earlier claimed that “there is no toxin complex operon in *Salmonella enterica*, this is a misleading statement”. Now the 183 examples from various *Salmonella enterica* subspecies we provided to the contrary have become “a good example for how the authors overstate microbiological data”. However, the 187 Tc toxin loci encoded by the 183 *Salmonella enterica* genomes represent 12% of the entire dbTC database, which contains a total of 1608 Tc toxin loci from all Tc toxin containing bacterial species. Furthermore, to quote from the Song et al. study (responsible for the dbTC database) which carefully analyzed the distribution of Tc toxins in *Salmonella* and *Yersinia*: “Almost all genomes in these [*S. enterica*] subspecies [IIIa, IIIb, IV, VII and novel subspecies A] carry at least one Tc locus.”. The presence of Tc toxins in *Salmonella* is therefore certainly more than just an evolutionary relic, as the reviewer has suggested here.

Where we do agree with the reviewer is that *Salmonella enterica* subsp. *houtenae* – as well as the others not belonging to subspecies I (*Salmonella enterica* subsp. *enterica*) – are generalist pathogens with broad host tropism rather than specialist human pathogens like the others mentioned in the same context when introducing **Supplementary Figure 14**. The main text and **Supplementary Figure 14** legend have been modified to reflect this fact.

Regarding the reviewer’s random sample analysis demonstrating that holin/endolysin genes can also be absent from Tc operons: this is exactly the case for *Y. entomophaga*, which serves as a perfect example that T10SS can be positioned in a genomically distal fashion from Tc toxins operons and not only co-encoded like in the examples we show in **Supplementary Figure 14**.

Finally, while it is possible that some of these pathogens have natural reservoirs in currently unknown invertebrate hosts, the potential role of Tc toxins in their infection can of course not be assessed in the absence of data regarding the identities of such hypothetical hosts.

8. The authors are wrong here. Tc confers insecticidal activity according to the literature. This does not exclude adaptation of the TCs to a function towards mammals, but few studies showing slight or specific effects on mice and human cells do not allow general positive conclusions on an in vivo role in mammals. None of the arguments provided by the authors really support their theory that Tc are abundant virulence factors in human pathogens. Such a statement semantically suggests that TC contribute to virulence towards humans. Literature, however, on such a conjunction is largely missing. Therefore, to not over

overstretch scientific data, TCs are to be termed as insecticidal toxins in the abstract and in the introduction. The sentence in the abstract (“such as TC toxins from human and insect pathogens”) is not supported by the literature and therefore misleading.

We have ultimately changed the phrasing of the abstract which was of concern to the reviewer, therefore the argument of whether Tc toxins are exclusively insecticidal (as the reviewer argues) or have more broad-range activity (as data from the Hinnebusch, Fuchs, Waterfield and our laboratory suggest) is rendered moot with respect to that part of the text.

The discussion on the abundance of Tc toxins in bacteria pathogenic to humans once again brings us back to the Song et al. 2021 study, where the authors show that Tc toxins are (opportunistically pathogenic *Salmonella enterica* subspecies notwithstanding) prominently featured in most genomes of *Yersinia* species - 10 of 15 known species complexes - particularly within the human pathogenic *Y. pseudotuberculosis*, *Y. pestis* and *Y. enterocolitica*. Therefore, Tc toxins being abundant in human pathogens is not even our theory – it is an established fact according to the literature.

The role of Tc toxins in these pathogens in turn returns us to the Spinner et al. 2013 study, which shows *Yersinia pestis* Tc toxin adaptation to mammalian hosts as opposed to flea hosts, which is independently supported by the Hares et al. 2008 study. These studies cannot be simply dismissed. However, as mentioned earlier, we agree with the reviewer that studies for the *in vivo* role of Tc toxins in mammals are comprehensive as those for insect hosts are so far still lacking. We have therefore acknowledged this issue in the introduction.

12. Comment only partially considered. In legend Fig. 2, please write “For quantitative data, see supplementary Fig. 6. The quantitative data (percentages) should be mentioned in the main text.

We have now expanded the **Figure 2e** legend and mentioned the exact percentage of YenR-GFP⁺ cells of **Supplementary Figure 6** in the main text.

13. Comment partially considered. For consistency, please use “YenA2-sfGFP” and “YenA2-His” instead of “YenTc-sfGFP” and “YenTc-His” throughout the text.

We have now modified the nomenclature for these two constructs throughout the text.

14. Comment on soldier cells considered.

Regarding the term “death factor”, I have major concerns about using such a term to rename identical factors from the same species. Again, genetic nomenclature must be as consistent as possible to not confuse the readers in the field (see also next comment). “YenDF” was already named “holA” and “pepB” in *Y. entomophaga* by Schoof et al., and there is no need to introduce a new abbreviation here. Lines 356-358 should accordingly be removed.

I agree with the modification made by the authors with respect to the term explosion.

In general, we agree with the reviewer’s argument that genetic nomenclature should be consistent whenever possible to avoid confusion. Although it is unfortunate timing that the Schoof et al. study came out while ours was in review and introduced a different set of terms for the individual components of the *Y. entomophaga* T10SS (RoeA, HolA, PepB, Rz, Rz1), we recognize that it is important to unify the nomenclature of these components for future readers’ sake. We have therefore now adopted the Schoof et al. notation with respect to these.

As for the term “YenDF”, as we have earlier clarified, this is a shorthand name we have given to the particular T10SS of *Yersinia entomophaga* instead of using YenR-YeHln-YeEln-YeIspn-YeOspn (or RoeA-HolA-PepB-Rz-Rz1 in Hurst notation) every time, which would be quite a mouthful. There is no equivalent shorthand for this T10SS, therefore no renaming is occurring here. Similar shorthand naming schemes are well established in secretion systems literature: for example, the complex T4SS from *L. pneumophila* is termed Dot/Icm as shorthand. We would therefore insist that the name “YenDF” be kept when referring to the *Y. entomophaga* T10SS for the convenience of the *Nature Microbiology* readership.

15. I strongly disagree. The name YenR is already introduced in the literature for a *Yersinia* regulator distinct from that mentioned in the manuscript. Moreover, the name “YenR” is highly unspecific as

42“Yen” might mean *Y. enterocolitica* or *Y. entomophaga*, and “R” for regulator is very unspecific as well. RoeA in contrast was introduced for the TC-related regulator in *Y. entomophaga* by Schoof et al. The term RoeA is indeed used for factors of other bacteria, but not in *Yersinia* species. To not confuse the community with two distinct names for the same protein, the authors should accept the term RoeA as already established.

The name YenR is meant to stand for “*Yersinia entomophaga* noxiousness regulator”, which provides an accurate definition of its purpose. The origin and meaning of the name RoeA remain cryptic in contrast – an informed audience will wonder why a name that is known in the literature as an abbreviation for “regulator of exopolysaccharide A” was used. However, as we state in the previous point, we will resort to the Schoof et al. 2023 nomenclature of the individual YenDF components for future readers’ sake regardless.

General remarks to be addressed:

17. Again, the molecular weights of YenTc components should be mentioned in the main text and in one full figure legend to ease understanding (and not only in the source data file). In Fig. S5, there is an obvious discrepancy between the molecular weights indicated in the source data file and the bands representing the toxin subunits. For example, YenA2 has a MW of ~180 according to the marker in contrast to the theoretical value of 156.18. This needs to be clarified

We thank the reviewer for the helpful suggestion. The molecular weights of the individual YenTc subunits have now been added to the section introducing the molecular composition of this toxin complex in the main text.

A close look at **Supplementary Figure 5** shows that YenA2 in fact migrates well below 180 kDa. Perhaps the reviewer’s original impression was due to downshifted MW marker numbering, which we have now amended.

18. At the bottom of Fig. 1, the authors show a draft/a map/a cluster of the YenDF operon that should be improved according to e.g. maps provided by Hurst and coworkers in numerous papers such as

43doi.org/10.1371/journal.pone.0263019 and shown separately in Fig. 1 as genetic map/cluster.

Our purpose for that inset is to provide the reader with a basic idea about the structure of the YenDF operon (YenR – YeOspn) and its closest genomic context, which it does well in its current state. We view the map of “YeRER” from the paper that the reviewer refers to as a useful complementary zoomed-out view of the genomic region containing the YenDF operon, but see no need to copycat that figure into our manuscript.

Minor points:

1. References: Photorhabdus in ref. 3. Use of capital letters in ref. 1, 23 etc.

We thank the reviewer for the clarification. The original reference titles have now been modified to unify their capitalization styles.

2. Line 36: “non-phage-derived” is mentioned too early here in the manuscript. Please remove here and use this phrase only when the holing/endolysin system is introduced.

This adjective has now been removed.

4. Please write: Allelic gene replacement with CamR.

This has now been corrected.

5. Line 81: comma between pestis and that.

44This has now been corrected.

Reviewer #2

Thanks for taking the time to revise the paper— it now looks great!

We would like to thank the reviewer for their positive feedback and assistance with improving the manuscript.

Reviewer #3:

The authors have addressed my comments and added in several quantifications that are helpful. I have no further comments and congratulate the authors on a very interesting body of work.

A quick comment on Major point 1 of Reviewer 1. Since the Sitsel et al manuscript was made available (Feb 22, 2023) prior to the Schoof et al manuscript (March 23, 2023), it would be unreasonable to have expected a discussion of the Schoof et al manuscript in the current version of the Sitsel et al manuscript. At this point, it would be reasonable for the authors to comment on any differences in results, interpretation and opinion, where the authors feel it is applicable.

We would like to thank the reviewer for their positive feedback and assistance with improving the manuscript.

Decision Letter, second revision:

Message: Our ref: NMICROBIOL-23030741B

30th October 2023

Dear Dr. Raunser,

Thank you for your patience as we've prepared the guidelines for final submission of your Nature Microbiology manuscript, "Specialized pathogenic cells of *Yersinia entomophaga* release Tc toxins using a type 10 secretion system" (NMICROBIOL-23030741B). Please carefully follow the step-by-step instructions provided in the attached file, and add a response in each row of the table to indicate the changes that you have made. Please also check and comment on any additional marked-up edits we have proposed within the text. Ensuring that each point is addressed will help to ensure that your revised manuscript can be swiftly handed over to our production team.

In recognition of the time and expertise our reviewers provide to Nature Microbiology's editorial process, we would like to formally acknowledge their contribution to the external peer review of your manuscript entitled "Specialized pathogenic cells of *Yersinia entomophaga* release Tc toxins using a type 10 secretion system". For those reviewers who give their assent, we will be publishing their names alongside the published article.

Nature Microbiology offers a Transparent Peer Review option for new original research manuscripts submitted after December 1st, 2019. As part of this initiative, we encourage our authors to support increased transparency into the peer review process by agreeing to

46have the reviewer comments, author rebuttal letters, and editorial decision letters published as a Supplementary item. When you submit your final files please clearly state in your cover letter whether or not you would like to participate in this initiative. Please note that failure to state your preference will result in delays in accepting your manuscript for publication.

Cover suggestions

COVER ARTWORK: We welcome submissions of artwork for consideration for our cover. For more information, please see our https://www.nature.com/documents/Nature_covers_author_guide.pdf target="new"> guide for cover artwork.

Nature Microbiology has now transitioned to a unified Rights Collection system which will allow our Author Services team to quickly and easily collect the rights and permissions required to publish your work. Approximately 10 days after your paper is formally accepted, you will receive an email in providing you with a link to complete the grant of rights. If your paper is eligible for Open Access, our Author Services team will also be in touch regarding any additional information that may be required to arrange payment for your article.

Please note that *Nature Microbiology* is a Transformative Journal (TJ). Authors may publish their research with us through the traditional subscription access route or make their paper immediately open access through payment of an article-processing charge (APC). Authors will not be required to make a final decision about access to their article until it has been accepted. <https://www.springernature.com/gp/open-research/transformative-journals> Find out more about Transformative Journals

Authors may need to take specific actions to achieve <https://www.springernature.com/gp/open-research/funding/policy-compliance-faqs> compliance with funder and institutional open access mandates. If your research is supported by a funder that requires immediate open access (e.g. according to <https://www.springernature.com/gp/open-research/plan-s-compliance> Plan S principles) then you should select the gold OA route, and we will direct you to the compliant route where possible. For authors selecting the subscription publication route, the journal's standard licensing terms will need to be accepted, including <https://www.nature.com/nature-portfolio/editorial-policies/self-archiving-and-license-to-publish> self-archiving policies. Those licensing terms will supersede any other terms that the author or any third party may assert apply to any version of the manuscript.

For information regarding our different publishing models please see our <https://www.springernature.com/gp/open-research/transformative-journals>

Transformative Journals page. If you have any questions about costs, Open Access requirements, or our legal forms, please contact ASJournals@springernature.com.

[redacted]

Best regards,

[redacted]

Reviewer #1:

Remarks to the Author:

Oleg Sitsel and colleagues submitted a second revised version of their manuscript now entitled "Specialized pathogenic cells of *Yersinia entomophaga* release Tc toxins using a type 10 secretion system" to Nature Microbiology for publication (no. NMICROBIOL-23030741B).

Comments to the author: Many of my concerns were adequately addressed. However, some points still require substantial modifications of the manuscript.

Major points:

3 In vivo aspect: to resolve this point, I propose to add a sentence in the discussion that in vivo, a distinct mechanism cannot be excluded.

5 I appreciate that the text was further modified, although I am not convinced by the comment of the authors to my point. Please delete "global" in the last sentence of the abstract.

6 I do not agree with the authors. However, I propose to add a sentence to lines 300-303 that the insecticidal TC in *Salmonella* strains point out to yet unknown invertebrate hosts of this human pathogen.

14 I strongly disagree with the term "death factor". Using a new (and dubious) term to rename a holin and an endolysin that are so well known as a lysis cassette by all biologist for decades would confuse the readers and is simply not acceptable in terms of consistency in genetic nomenclature. In parenthesis, the authors might also consider that such a term looks ridiculous in the eye of bacteriologists and other readers of the field. The phage-related genes *holA* and *pepB* constitute a lysis cassette introduced in the literature as LC (Schoof et al. 2023), which is an excellent shorthand. Moreover, this will help the authors to more precisely define this term as their definition made in lines 144-146 is very misleading (the "it" in this sentence seems to refer to the cell death phenotype). I therefore insist on replacing "YenDF" with "YenLC" throughout the manuscript.

15 I appreciate that the authors now refer to the Schoof et al. 2023 nomenclature. Please note that YenR is still mentioned in Figure 2.

4818 At the bottom of Fig. 1, the authors show a draft/a map/a cluster of the YenDF operon that should be improved. As the authors do not agree with the map proposed, please design your own one and put it into the Figure.

Final Decision Letter:

Mes 29th November 2023

sag

e: Dear Professor Raunser,

I am really pleased to accept your Article "Yersinia entomophaga Tc toxin is released by T10SS-dependent lysis of specialized cell subpopulations" for publication in Nature Microbiology. Thank you for having chosen to submit your work to us and many congratulations.

Acceptance of your manuscript is conditional on all authors' agreement with our publication policies (see <https://www.nature.com/nmicrobiol/editorial-policies>). In particular your manuscript must not be published elsewhere and there must be no announcement of the work to any media outlet until the publication date (the day on which it is uploaded onto our website).

Please note that *Nature Microbiology* is a Transformative Journal (TJ). Authors may publish their research with us through the traditional subscription access route or make their paper immediately open access through payment of an article-processing charge (APC). Authors will not be required to make a final decision about access to their article until it has been accepted. [Find out more about Transformative Journals](https://www.springernature.com/gp/open-research/transformative-journals)

49Authors may need to take specific actions to achieve [compliance](https://www.springernature.com/gp/open-research/funding/policy-compliance-faqs) with funder and institutional open access mandates. If your research is supported by a funder that requires immediate open access (e.g. according to [Plan S principles](https://www.springernature.com/gp/open-research/plan-s-compliance)) then you should select the gold OA route, and we will direct you to the compliant route where possible. For authors selecting the subscription publication route, the journal's standard licensing terms will need to be accepted, including [self-archiving policies](https://www.nature.com/nature-portfolio/editorial-policies/self-archiving-and-license-to-publish). Those licensing terms will supersede any other terms that the author or any third party may assert apply to any version of the manuscript.

As Emily mentioned in her previous email, we welcome the submission of potential cover material (including a short caption of around 40 words) related to your manuscript; suggestions should be sent to Nature Microbiology as electronic files (the image should be 300 dpi at 210 x 297 mm in either TIFF or JPEG format). We think that some of the images you have in Fig. 3 for example, are really striking, and might work well as potential cover images. Please note that such pictures should be selected more for their aesthetic appeal than for their scientific content, and that colour images work better than black and white or grayscale images. Please do not try to design a cover with the Nature Microbiology logo etc., and please do not submit composites of images related to your work. I am sure you will understand that we cannot make any promise as to whether any of your suggestions might be selected for the cover of the journal.

With kind regards,

50[redacted]

P.S. Click on the following link if you would like to recommend Nature Microbiology to your librarian <http://www.nature.com/subscriptions/recommend.html#forms>

** Visit the Springer Nature Editorial and Publishing website at http://editorial-jobs.springernature.com?utm_source=ejp_NMicro_email&utm_medium=ejp_NMicro_email&utm_campaign=ejp_NMicro for more information about our career opportunities. If you have any questions please click [here](mailto:editorial.publishing.jobs@springernature.com).**